# Attention-enhanced variational learning for physically informed discovery of exceptionally hard multicomponent bulk metallic glasses

Anurag Bajpai ✉, Jaemin Wang, Barak Ratzker, Bilgehan Murat Şeşen, Florian Kark & Dierk Raabe ✉

The discovery of high-performance multicomponent alloys is constrained by the vastness of composition space and the scarcity of experimentally validated data. We develop VIBANN, a variational information bottleneck-augmented attention-based neural network framework, for uncertainty-aware inverse design of exceptionally hard bulk multicomponent metallic glasses. The model learns chemically structured latent representations of alloy composition and indentation load to search the candidate space under constraints of chemical plausibility, novelty, and predictive uncertainty. Guided by this framework, we synthesize five B-Nb-Fe-W-Co/Hf/Ru/Zr-rich bulk multicomponent metallic glasses. All alloys form fully amorphous rods of 2 mm diameter and reach Vickers hardness values of about 2450 HV, among the highest reported for bulk metallic glasses under comparable conditions. Latent space analysis, attribution trends, and molecular dynamics-based atomistic simulations show that exceptional hardness in this compositional space arises from dense atomic packing, boron-enriched short-range environments and refractory-stabilized local rigidity. Together, we show that uncertainty-aware latent space learning can discover bulk metallic glasses that combine amorphous structure with exceptionally high hardness under limited data conditions.

Metallic glasses (MGs), and particularly multicomponent metallic glasses (MMGs), combine exceptional hardness, elastic resilience, and corrosion resistance, making them attractive for wear-resistant coatings, precision mechanical components, and microelectromechanical systems[1–5]. Among these attributes, achieving high hardness is a central objective for these applications because hardness often defines the operating window in which damage accumulates within a shallow subsurface volume[6,7]. In contrast to crystalline materials, where hardness is often linked to dislocation-mediated mechanisms and microstructural length scales, plasticity in MMGs proceeds through the activation and percolation of shear transformation zones (STZs). Consequently, the macroscopic response is controlled by local packing, chemical short-range order, and the distribution of excess free volume[8,9]. As a result, compositional modifications that appear minor on an atomic percent basis can produce disproportionate changes in the ease of STZ initiation and the subsequent propensity for shear localization. Designing high-hardness bulk MMGs, therefore, requires precise control over multiscale atomic interactions, a challenge compounded by the high dimensionality of composition space and the absence of universal descriptors for mechanical behavior.

Classical alloy design strategies for MGs have relied on heuristic criteria, such as Turnbull-type considerations for glass formation, atomic-size mismatch, and enthalpy-of-mixing arguments, that promote dense packing and suppress crystallization[10,11]. These concepts

Max Planck Institute for Sustainable Materials, Düsseldorf, Germany. ✉e-mail: a.bajpai@mpi-susmat.de; d.raabe@mpi-susmat.de

have provided reliable guidance and have enabled the discovery of several glass-forming alloy families. However, they are insufficient for systematically optimizing mechanical properties, such as hardness, across quaternary and higher-order systems, where multiple competing interactions shape both amorphous phase stability and local resistance to shear. Even when high-throughput routes are used, experimental variability and protocol dependence, including indentation-size effects, complicate direct comparisons across the literature and can obscure the narrow compositional windows that produce exceptional hardness[12]. In parallel, the combinatorial growth of candidate chemistries makes purely empirical exploration increasingly inefficient, particularly when the aim is to identify new bulk MMG compositions that simultaneously satisfy glass-forming constraints and deliver exceptional hardness.

Data-driven approaches have begun to address these challenges by learning composition–property mappings directly from existing literature datasets. Supervised models have been reported for glass-forming ability, characteristic temperatures, and mechanical properties of MGs[13–17]. Despite this progress, most data-driven frameworks are limited to forward prediction, require laborious feature engineering, and typically underperform in close extrapolation regimes, especially when targeting extreme property values[18]. When the training data are sparse near the upper tail of the property distribution, models can produce overconfident extrapolations or return candidates that interpolate among well-sampled regions, neither of which provides a reliable route to designing new materials with exceptional performance.

Generative and latent space-based approaches offer an alternative by enabling systematic exploration of the composition space under learned constraints[19,20]. For instance, Li et al. recently demonstrated a variational autoencoder (VAE) framework that jointly learns to predict properties and generate new alloy compositions with improved GFA[21]. Despite these advances, significant challenges remain. Most generative models treat all compositional degrees of freedom equivalently, diluting the influence of chemically dominant interactions, which are precisely the interactions experimentalists seek to control[22]. They often also provide limited means to separate robust trends from dataset-specific correlations, and frequently depend on post hoc analyses to explain what has been learned[23]. Forrest and Greer demonstrated that even relatively simple neural networks required post hoc permutation tests to relate learned representations to known empirical criteria, underscoring the difficulty of achieving inherently explainable models in this domain[24]. For inverse design of new and exceptionally hard bulk MMGs, this limitation is consequential because model explainability and confidence are not secondary attributes. They determine whether a proposed alloy is chemically plausible, whether it reflects a meaningful separation from known alloys, and whether the prediction is reliable enough to justify experimental synthesis.

In this work, we introduce a Variational Information Bottleneck-augmented Attention Neural Network (VIBANN) framework that addresses these gaps by embedding explainability, uncertainty, and latent optimization into an experimentally validated unified inverse-design pipeline for high-hardness bulk MMG design. Unlike conventional surrogate–generative frameworks, our VIBANN architecture explicitly optimizes a trade-off between latent information compression and predictive accuracy, promoting representations that capture the most essential and physically relevant features for hardness prediction. The self-attention mechanism dynamically reweights input features, enabling the model to assign differential importance to the elemental contributions that are critical to hardness. This combination of disentangled latent learning and feature attribution yields an explainable framework that not only predicts hardness but also guides experimental exploration of compositional spaces (Fig. 1). Using this unified framework, we identify a narrow set of high hardness

candidates and realize five bulk MMGs with Vickers hardness exceeding 2050 HV, approaching the upper bounds reported for bulk MGs under comparable testing conditions. Beyond proposing new compositions, the framework yields composition-resolved insight into the elemental contributions and interaction patterns that most strongly promote hardness within the explored chemistry space. Overall, this work provides an experimentally validated inverse design strategy that integrates physics-informed latent representation learning, feature attribution, and uncertainty quantification for reliable materials design with exceptional properties.

## Results

Figure 1 provides an overview of the VIBANN workflow. The input to the framework is the alloy composition expressed on a normalized composition simplex together with the applied indentation load. The composition vector is first processed by an attention module that learns a weighted representation of the elemental degrees of freedom. This representation is then compressed via a variational information bottleneck (VIB) into a low-dimensional latent vector, trained to remain predictive of hardness while limiting redundant or noise-dominated information. A regression head predicts hardness, and a decoder head reconstructs the composition from the latent vector to maintain a chemically consistent representation that can be traversed during inverse design. Model uncertainty is quantified via Monte Carlo dropout at inference, with batch normalization kept frozen, enabling repeated stochastic forward passes to obtain a predictive distribution for each candidate composition and load. The same predictive mean and uncertainty are then used both to assess model reliability and to enforce risk-aware decision rules during inverse design.

### VIBANN model performance

Figure 2 summarizes the predictive and generalization performance of the VIBANN framework. The parity plot in Fig. 2a shows close agreement between the measured and predicted hardness for both the training and held-out test sets. On the test set, the model achieves high accuracy with $R^2 \approx 0.943$ and low error (mean absolute error, MAE $\approx$ 55.6 HV; root mean squared error, RMSE $\approx$ 101.4 HV), corresponding to an average deviation of only ~11% (mean absolute percentage error, MAPE). The predictions follow the parity line across the full hardness range, including the high-hardness tail most relevant to inverse design. Uncertainty intervals are narrow for most points and expand in regions less densely supported by the training distribution. The inset uncertainty histogram shows a unimodal distribution, with most predictions exhibiting modest dispersion, consistent with stable inference across repeated Monte Carlo dropout evaluations. Across cross-validation folds, the training curves remain consistent, and the performance improves systematically with increasing training set size, supporting stable convergence and the absence of fold-specific instabilities as shown in Supplementary Figs. 5, 6. Residual analysis (Supplementary Fig. 7) shows a progressive negative bias with increasing hardness, indicating that the predictive mean becomes increasingly conservative in the upper tail of hardness, even though the model remains well centered across the full test set.

Figure 2b highlights the impact of the information bottleneck on model learning and generalization. Compared with an identical attention backbone in which the VIB is replaced by a deterministic latent projection, the VIBANN model shows lower validation error throughout training and a reduced train-validation gap. This behavior indicates improved regularization and reduced sensitivity to spurious correlations in the limited data regime, enabling the model to capture only the fundamental composition-load factors that govern hardness[25].

Supplementary Fig. 8 quantifies the behavior of the adaptive bottleneck strength during training and the associated balance between reconstruction fidelity and information regularization. The

 

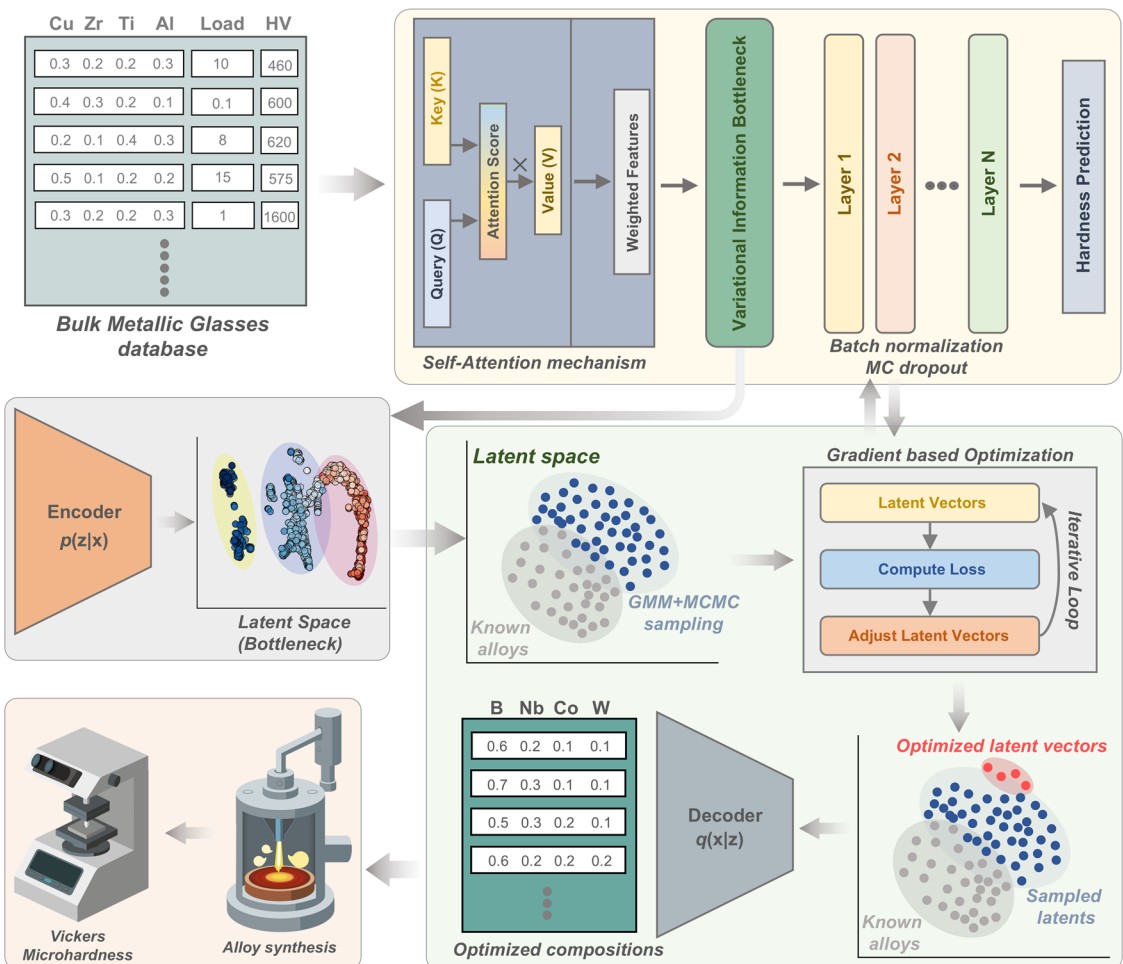

**Fig. 1 | Closed-loop VIBANN workflow for inverse design of bulk multi-component metallic glasses.** The input database contains alloy compositions, indentation loads, and measured Vickers hardness values. These data are used to train an attention-based neural surrogate in which a self-attention block reweights the compositional inputs, followed by a variational information bottleneck that compresses each composition–load pair into a low-dimensional latent representation. Batch normalization and Monte Carlo dropout are used in the predictor to improve generalization and quantify predictive uncertainty. The learned latent space is then used for inverse design by fitting a latent density model, sampling plausible candidate latents with a Gaussian-mixture-guided MCMC strategy, and iteratively refining promising seeds by gradient-based optimization of an uncertainty-aware objective. The optimized latent vectors are decoded into simplex-constrained alloy compositions, which are then synthesized and experimentally validated by Vickers microhardness measurements. This framework links data-driven hardness prediction, uncertainty-aware latent-space exploration, candidate generation, and experimental verification within a single alloy-discovery loop.

coefficient $\beta$ is adjusted by feedback control to maintain a target information rate through the bottleneck[26]. Accordingly, $\beta$ evolves rapidly during early epochs as the model establishes the latent distribution (Supplementary Fig. 8a), and then stabilizes (to 0.0007) once the Kullback-Leibler (KL) divergence per dimension approaches the target value. This early-stage rise and subsequent decay suggest a feedback-driven annealing-like mechanism: during initial training, stronger KL regularization (i.e., higher $\beta$) encourages exploration of a broader set of latent configurations, promotes disentanglement, and reduces overfitting to spurious correlations in the training data. This phase of the workflow is critical for forming a meaningful latent manifold that captures the intrinsic data-generating structure of MMG compositions under varying loads. As training progresses and the latent space becomes more organized, $\beta$ begins to decay, following a feedback-driven annealing strategy where the model dynamically suppresses the KL penalty to refine reconstruction accuracy[27].

Consistent with this behavior, the reconstruction loss decreases sharply during early training and then plateaus at a low value (0.16; Supplementary Fig. 8b), demonstrating that the decoder preserves composition-level consistency while the bottleneck enforces a bounded information content in the latent representation. The KL term increases from its initial value and approaches a steady regime without collapse, indicating that the latent variables remain active and that the model avoids degenerate solutions in which the latent channel carries negligible information. Together, the $\beta$ trajectory, reconstruction loss, and KL evolution support the conclusion that the bottleneck maintains an informative yet regularized latent representation suitable for subsequent latent traversal in inverse design.

The calibration analysis in Supplementary Fig. 9 evaluates whether the Monte Carlo dropout uncertainty is statistically consistent with the observed prediction errors. The reliability diagram in Supplementary Fig. 9a compares nominal confidence levels with empirical coverage of the corresponding prediction intervals and shows close agreement with the perfect calibration line across the full range of confidence levels. The resulting mean absolute calibration error (MACE) and integrated mean calibration error (IMCE) are both low, with MACE = 0.0559 and IMCE = 0.0553, indicating only modest deviation between nominal and empirical coverage. Supplementary Fig. 9b reports the mean prediction interval width as a function of nominal confidence, showing the expected monotonic

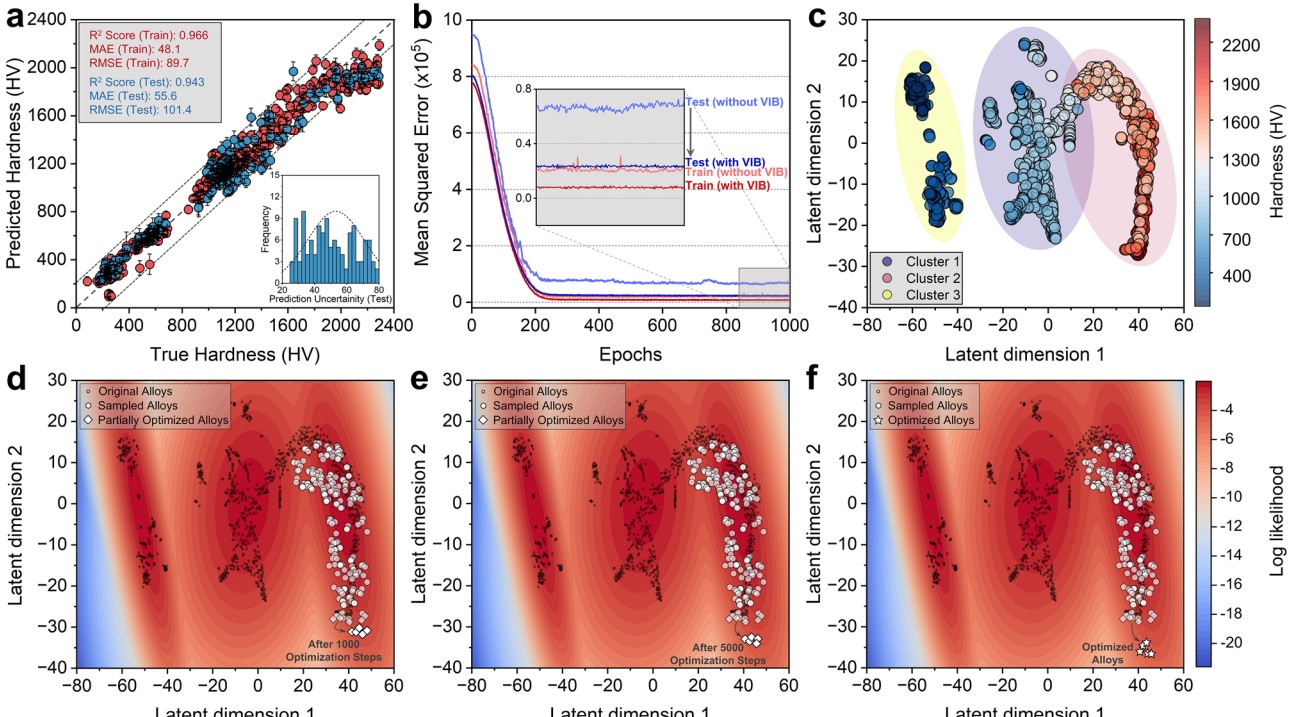

**Fig. 2 | Predictive performance of VIBANN and latent space guided inverse design of high-hardness bulk multicomponent metallic glasses. a** Predicted hardness versus measured hardness for the training set ($n = 538$) and the held-out test set ($n = 135$), where one data point represents one composition–load–hardness observation from the curated dataset. Symbols denote the predictive mean hardness, and the error bars denote ±1σ predictive uncertainty from $n = 100$ stochastic Monte Carlo dropout forward passes for each observation. The gray dashed diagonal indicates perfect agreement between predicted and measured values. The inset shows the distribution of predictive uncertainty for the test set. The dashed curve is the kernel density estimate of the uncertainty distribution. **b** Mean squared error during training for models with and without the variational information bottleneck. The inset shows the low-error region at later epochs, where the model

with the bottleneck exhibits lower, more stable training and test errors. **c** Two-dimensional projection of the learned latent representation of the alloy dataset, colored by measured hardness. Three composition clusters are indicated to highlight the organization of low, intermediate, and high hardness alloys in latent space. **d**–**f** Gaussian mixture model log-likelihood landscape defined over the latent space and used to constrain candidate generation. Black dots denote the original alloy dataset. White circles denote sampled latent points. White diamonds denote partially optimized candidates after (**d**) 1000 optimization steps and (**e**) 5000 optimization steps. White stars in (**f**) denote the final optimized candidates. The progression shows candidate refinement within data-supported latent regions toward the high hardness sector of the learned design space. Source data are provided as a Source Data file.

increase in interval width with increasing coverage. Supplementary Fig. 9c provides the probability integral transform (PIT) distribution, which is broadly distributed over the unit interval without strong pile-up at 0 or 1, consistent with uncertainty estimates that are neither systematically under-dispersed nor dominated by extreme overconfidence. To evaluate model calibration more explicitly, we computed error statistics for the full test set and upper-hardness subsets, as summarized in Supplementary Table 3. Over the full test set, the model remains nearly unbiased, with a mean signed error of −4.79 HV. However, the mean signed error becomes progressively more negative in the upper tail, reaching −82.13 HV for the top 20% of true hardness values and −122.14 HV for the top 10%. The corresponding mean absolute errors also increase from 55.61 HV for the full test set to 122.55 HV and 151.8 HV in these subsets. The fraction of predictions lying within the reported ±1σ interval decreases from 0.62 over the full test set to 0.41 and 0.34 in the top 20% and top 10% subsets, respectively. Together, these diagnostics indicate that the VIBANN model's uncertainty is overall sufficiently calibrated to support conservative scoring and inverse design, while residual analysis and signed-error statistics show a mild progressive weakening of coverage in the sparsest, high-hardness tail. This systematic under-prediction in the highest-hardness regime suggests that hardness substantially above 2500 HV may involve features that are only incompletely captured by the present training distribution, where such extreme values remain sparsely represented (Supplementary Fig. 7). This may reflect either distinct deformation behavior at the

extreme upper tail or limited sampling of the relevant compositional space.

Supplementary Figs. 10–12 benchmark VIBANN against standard regression baselines, including linear models, nearest neighbors, random forests, gradient boosting, and multilayer perceptrons. All baseline models exhibit greater variance and tend to underpredict in the high-hardness regime. VIBANN outperforms all benchmarks in both accuracy and calibration. Ablation studies further show that removing either the attention module or the VIB regularization degrades performance, with higher RMSE and weaker generalization, confirming that both components contribute materially to predictive stability and to robust uncertainty estimates (Supplementary Fig. 13). Detailed benchmarking against several recently developed ML frameworks for alloy design further underscores VIBANN's integrative advantage, combining model accuracy, explainability, uncertainty handling, and closed-loop inverse design (Supplementary Note 12 and Supplementary Table 4).

## VIB latent space as a design landscape

The composition-property landscape of MMGs is strongly nonlinear and high-dimensional, which limits direct interpretation and hampers inverse search when compositions are treated in the original simplex coordinates. The VIBANN framework addresses this by mapping each normalized composition, along with its associated load value, to a low-dimensional latent representation trained to preserve hardness-relevant information while suppressing redundant variability. This

latent representation provides a compact coordinate system in which chemically related alloys are organized into coherent neighborhoods and can be explored in a controlled manner during inverse design.

The latent embedding obtained from the trained VIB model shows a structured organization of the dataset. The t-SNE visualization in Supplementary Fig. 15 reveals an organized manifold, with compositions grouped into three topologically distinct regions, consistent with the clustering analysis and the elbow criterion (Supplementary Fig. 14). The spatial layout of the latent embedding reflects a smooth, arched distribution with clear separation boundaries, suggestive of underlying non-linear compositional gradients captured by the VIB. Importantly, the same latent space, when colored by hardness values (Fig. 2c), shows a continuous stratification of hardness along the occupied latent regions. This alignment confirms that the latent coordinates capture property-relevant structure rather than arbitrary compression. The hardness distributions within the three regions further support this interpretation (Supplementary Fig. 16). The high-hardness region (Cluster 2) contains compositions with higher fractions of refractory and metalloid additions, whereas the low-hardness region (Cluster 3) is enriched in chemistries with Mg, Cu, and Zr, which reduce shear localization resistance due to lower bond strength and cohesive energy. These trends are consistent with established metallurgical expectations for stiffness and shear resistance in bulk MGs and indicate that the embedding organizes alloys according to hardness-relevant compositional motifs[28–31].

The element-resolved attribution provides an additional internal consistency check on the latent organization. The cluster-specific permutation analysis (Supplementary Fig. 17) identifies distinct dominant contributors across the three regions, with the high hardness region showing strong dependence on elements that increase bond strength and constrain local shear transformations[32], whereas the low hardness region is dominated by elements associated with reduced resistance to shear localization. The agreement between latent structure and hardness stratification shows that the VIB-designed latent space encodes meaningful representations of composition-hardness relationships within the investigated chemistry domain.

An ablation study confirms that this latent organization is enabled by the VIB and is directly relevant for inverse design. Supplementary Fig. 18 compares the latent maps obtained from identical attention backbones trained with VIB enabled and with the VIB replaced by a deterministic bottleneck of the same dimensionality. With VIB enabled, the embedding forms coherent occupied regions with low-density separations and a smoother hardness stratification. Without VIB, the embedding becomes more fragmented, and high-hardness points are distributed across disconnected islands, which is unfavorable for latent traversal and gradient-based refinement. The quantitative latent space navigability diagnostics reported in Supplementary Note 6 and Supplementary Fig. 19 corroborate these visual trends. With VIB enabled, the k-nearest neighbor graph in latent space is dominated by a single connected component and exhibits lower local hardness variation per latent displacement, whereas the deterministic ablation shows more connected components and higher local slope statistics. These observations support the idea that VIB regularization produces a better-conditioned latent geometry for inverse design, because small latent updates are more likely to remain within data-supported regions and induce controlled changes in both decoded composition and predicted hardness.

## Inverse design via latent space optimization
The latent space learned by VIB provides a connected, locally smooth representation of the training compositions. This property is essential for inverse design because the refinement step follows gradients in latent space. A fragmented embedding with abrupt local property changes would make continuous latent updates unstable and increase the probability of traversing low-density regions where decoded compositions are weakly supported by the data.

We performed inverse design for a set of fixed loads (0.5–10 N) by treating the trained VIBANN surrogate as a differentiable mapping from a latent coordinate to hardness via the decoder and prediction heads. The design search is conducted in latent space, but all candidate compositions are generated on the simplex through the decoder, ensuring non-negativity and normalization by construction. To constrain the search to data-supported regions, we modeled the distribution of training embeddings using a weighted Gaussian mixture model (GMM)[33] with three components, consistent with the dominant structure observed in the latent embedding (Fig. 2c). The fitted mixture defines an explicit density model over latent space and therefore an operational notion of distributional support for any proposed latent point. Latent proposals are restricted to regions of sufficiently high likelihood under this model so that decoded candidates remain consistent with the training manifold rather than arising from unconstrained extrapolation.

Candidate generation proceeds in two stages. We performed mixture-aware multi-chain sampling in the latent space to obtain a diverse set of plausible latent points concentrated in the high hardness region of the latent embedding. Sampling of the new points was carried out using Markov-chain Monte Carlo (MCMC)[34], with the GMM density as the prior, and a hardness utility derived from Monte Carlo dropout evaluation at the fixed load, so that proposals are biased toward high mean hardness while penalizing high epistemic uncertainty. In addition to the GMM plausibility term, the sampler includes soft penalties for proposals whose conservative bound falls below a training-anchored target and for proposals whose uncertainty exceeds a reference level, thereby concentrating samples within data-supported regions that are robust under uncertainty (see Methods for details). The resulting samples populate the high-density regions of the latent manifold while avoiding low-density troughs (Fig. 2d), consistent with constrained traversal within the learned latent support.

We then refined a subset of ten high-quality latent seeds using gradient-based optimization. Each seed is updated by back-propagating an uncertainty-aware hardness objective with respect to the latent coordinate, decoding the updated latent point to a composition, and re-evaluating the predicted hardness under the same load. This latent space optimization continues until convergence, i.e., the hardness gain per step becomes negligible (Supplementary Fig. 20). The optimization is guided by a custom loss function that penalizes deviations from a target hardness window (here set to 2200–2500 HV) and also discourages straying far from the latent manifold learned from real data. As a result, the latent vectors progressively move deeper into the high-hardness region of latent space. Figure 2d, e illustrate the optimization trajectories after 1000 and 5000 iterations, where the optimizing points have gradually moved within the high-likelihood region toward the higher hardness sector of the latent space relative to their initial positions (dashed circles). Final bulk MMG candidates were selected using an explicit acceptance rule that requires joint satisfaction of latent plausibility under the GMM, a minimum novelty distance from the training set, an uncertainty cap, and a conservative performance threshold defined by a lower confidence bound score. This design rule yields continuous trajectories in latent space and a reduction in the optimization objective (Supplementary Fig. 20), with convergence behavior that is consistent across independent initializations, indicating that the refined solutions are not isolated numerical artifacts but correspond to a reproducible, high-hardness region in the learned design landscape.

After optimization, the five refined latent vectors cluster near but slightly beyond the original high-hardness region (Fig. 2f, white stars). These latent vectors decode into candidate compositions that represent controlled departures from the nearest training alloys while remaining within regions of high latent support. Quantitatively, the

**Table 1 | Compositional profiles, thermal properties (T$_g$, T$_x$, ΔT$_x$), and experimentally measured versus VIBANN-predicted hardness (HV) across multiple loads for the five optimized bulk MMGs**

| Alloy ID | Alloy | Load (N) | Predicted hardness (HV) | Predicted uncertainty (HV) | Experimental hardness (HV) | T$_g$ (K) | T$_x$ (K) | ΔT$_x$ (K) |
|---|---|---|---|---|---|---|---|---|
| A1 | $B_{68}Nb_{24}Fe_4W_4$ | 0.5 | 2342.1 | 124.7 | 2446.4 ± 44.0 | 952 | 1002 | 50 |
| | | 1 | 2286.9 | 110.8 | 2371.0 ± 43.2 | | | |
| | | 2 | 2231.9 | 98.3 | 2292.0 ± 46.6 | | | |
| | | 3 | 2168.7 | 93.5 | 2231.0 ± 47.4 | | | |
| | | 5 | 2071.7 | 81.3 | 2124.0 ± 54.4 | | | |
| | | 10 | 2008.3 | 72.1 | 2048.0 ± 50.1 | | | |
| A2 | $B_{62}Nb_{12}Fe_4Hf_8Ru_6W_8$ | 0.5 | 2246.8 | 114.0 | 2348.7 ± 42.1 | 934 | 983 | 49 |
| | | 1 | 2186.9 | 102.7 | 2271.5 ± 49.9 | | | |
| | | 2 | 2098.9 | 91.9 | 2167.0 ± 51.5 | | | |
| | | 3 | 2039.1 | 78.3 | 2091.3 ± 53.1 | | | |
| | | 5 | 1921.6 | 72.7 | 1972.9 ± 48.5 | | | |
| | | 10 | 1866.8 | 68.2 | 1901.4 ± 49.8 | | | |
| A3 | $B_{64}Nb_{23}Fe_5Co_8$ | 0.5 | 2185.2 | 106.5 | 2279.3 ± 44.6 | 930 | 976 | 46 |
| | | 1 | 2111.5 | 92.9 | 2185.6 ± 48.1 | | | |
| | | 2 | 2009.9 | 84.6 | 2072.6 ± 49.1 | | | |
| | | 3 | 1961.1 | 77.8 | 2008.6 ± 50.2 | | | |
| | | 5 | 1884.7 | 71.3 | 1920.4 ± 44.7 | | | |
| | | 10 | 1822.3 | 65.7 | 1861.8 ± 47.4 | | | |
| A4 | $B_{66}Nb_{21}Fe_4Hf_4Ru_5$ | 0.5 | 2070.8 | 98.3 | 2160.8 ± 49.3 | 905 | 962 | 57 |
| | | 1 | 2002.0 | 91.2 | 2074.7 ± 47.9 | | | |
| | | 2 | 1921.2 | 83.4 | 1978.4 ± 50.8 | | | |
| | | 3 | 1875.0 | 74.2 | 1914.5 ± 48.4 | | | |
| | | 5 | 1791.2 | 70.8 | 1839.5 ± 52.1 | | | |
| | | 10 | 1736.8 | 63.8 | 1772.6 ± 51.7 | | | |
| A5 | $B_{61}Nb_{18}Fe_3Co_5W_8Zr_5$ | 0.5 | 1964.9 | 96.3 | 2036.2 ± 47.3 | 910 | 966 | 56 |
| | | 1 | 1902.5 | 87.2 | 1962.7 ± 51.0 | | | |
| | | 2 | 1814.5 | 78.4 | 1870.2 ± 49.2 | | | |
| | | 3 | 1758.6 | 70.2 | 1801.6 ± 46.9 | | | |
| | | 5 | 1679.5 | 63.8 | 1708.5 ± 47.5 | | | |
| | | 10 | 1628.8 | 58.9 | 1651.8 ± 41.2 | | | |

Prediction uncertainties represent ±1σ intervals.

optimized compositions differ from their nearest neighbors by a median Euclidean distance of ~0.38 in composition space (Supplementary Fig. 22, Supplementary Table 5) and ~1.2 in latent space, confirming controlled extrapolation. The corresponding latent likelihood remains high at these points (~ −6), indicating that the candidates remain within the support of the learned distribution. These prospective bulk MMGs are reported in Table 1 together with the associated predictive statistics.

All five inverse-designed candidates (A1–A5) lie within a broader B-Nb-Fe-centered high-hardness compositional region. This places them in chemical continuity with earlier reported hard Fe-Nb-B thin-film MGs[35], rather than in a wholly unrelated chemistry class. Among the present candidates, A1 is compositionally closest to the boride-rich $Fe_3Nb_{25}B_{72}$ thin-film composition reported previously, whereas A2–A5 represent progressively larger multicomponent extensions of this broader hard-BMG basin. This convergence is consistent with the objective of the present inverse-design framework, which explicitly favors candidates that jointly satisfy high predicted hardness, latent-space plausibility, and low epistemic uncertainty within the training-supported bulk metallic-glass domain. The resulting alloys should therefore be interpreted as uncertainty-screened bulk multi-component refinements of a chemically credible high-hardness region, rather than as unconstrained extrapolations into a disconnected alloy family. This is further examined through synthesis of the nominal $B_{72}Nb_{25}Fe_3$ ternary reference alloy and model-based comparisons at

representative loads (provided in Supplementary Note 10, Supplementary Figs. 29–31, and Supplementary Table 8).

To test whether the learned structure was confined to this boride-centered region, we performed a leave-family-out validation on the Mo–Ta–Si–W family, selected from the eligible non-boride subset based on its comparatively high median hardness and sufficient representation. When this entire family was excluded from training, the model retained good accuracy ($n = 19$, $R^2 = 0.89$, RMSE = 32.59 HV, MAE = 24.46 HV; Supplementary Fig. 38). Thereafter, under an explicit boron-free inverse-design constraint, VIBANN identified chemically plausible non-boride candidates with predicted hardness up to 1876.4 HV and lower confidence bounds up to 1722.8 HV (Supplementary Table 9). These analyses indicate that the learned latent structure is not reducible to a single boride-rich region, although its interpretive scope remains restricted to the bulk MMG manifold represented in the data.

To benchmark the VIBANN framework against a widely used surrogate optimization baseline, we implemented a local Gaussian process-Bayesian optimization (GP BO) framework trained on the same composition-load dataset and evaluated under the same fixed load condition used throughout our inverse design (Supplementary Note 8). This GP-BO baseline serves as a reference for acquisition-driven local optimization near the best-observed-hardness region. The GP-BO baseline achieves rapid early improvement in conservative performance but produces proposals that remain clustered near the training compositions with limited novelty and diversity

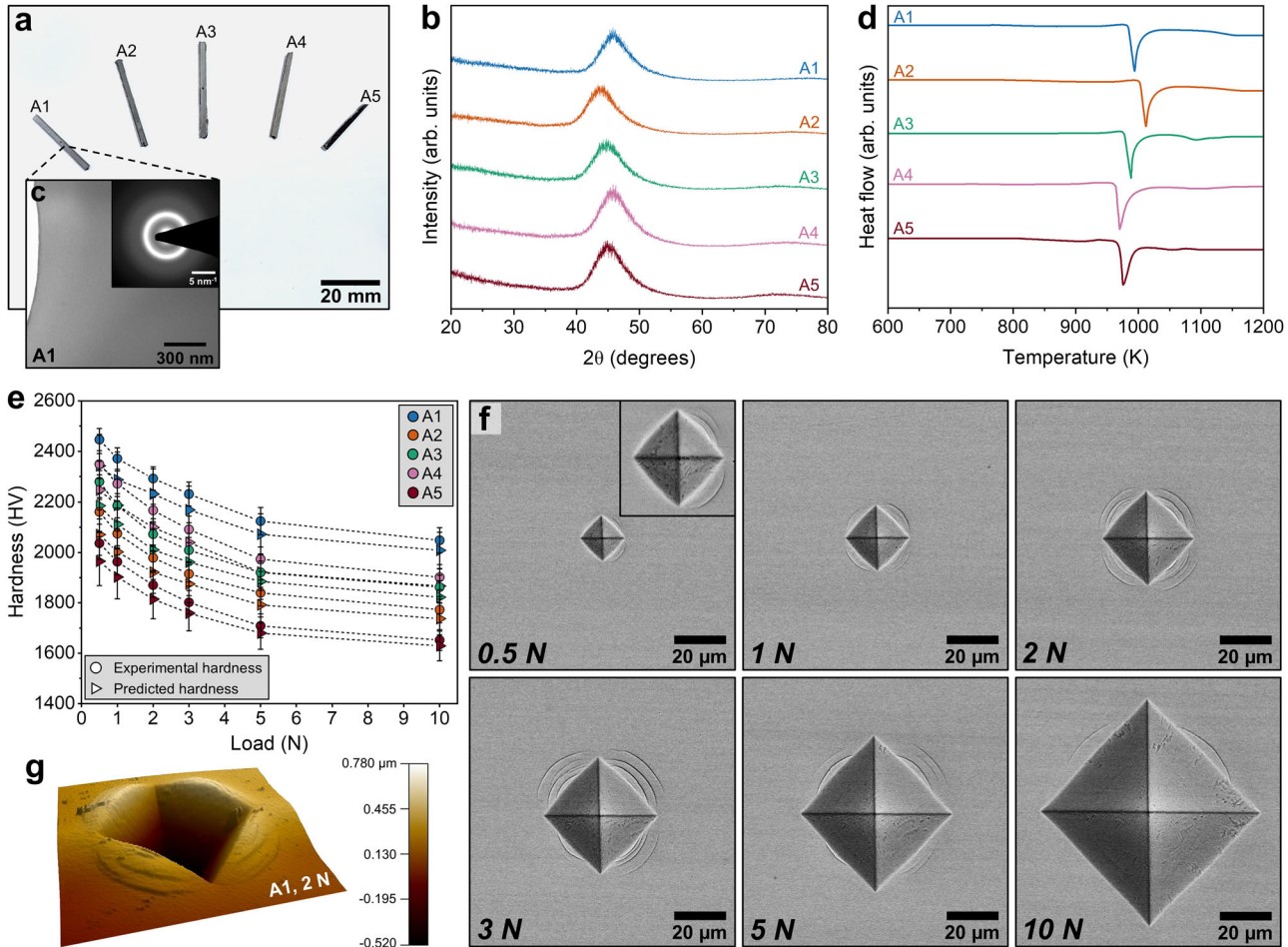

**Fig. 3 | Experimental synthesis and mechanical characterization of optimized bulk MMG alloys. a** Cast rod geometries of alloys A1-A5 reveal successful synthesis via copper mold suction casting. **b** X-ray diffraction patterns of all alloys exhibit broad diffuse halos, characteristic of amorphous structures. **c** Bright-field TEM image and corresponding SAED pattern of A1 confirms its fully amorphous nature. **d** Differential scanning calorimetry (DSC) curves highlight the glass transition ($T_g$) and crystallization ($T_x$) events; A1 shows the largest supercooled liquid region ($\Delta T_x$), consistent with enhanced thermal stability. **e** Vickers microhardness as a function of load (0.5–10 N). Experimental points are presented as mean hardness values from $n = 6$ replicate indentations for each alloy at each load, and error bars denote $\pm 1$ standard deviation (s.d.) across the six indentations. Predicted points denote the VIBANN predictive mean for the same alloy–load conditions with error bars denoting the Monte Carlo uncertainty (from 100 stochastic passes) associated with the prediction; A1 consistently exhibits the highest hardness across loads. **f** SEM images of representative indentations (0.5–10 N) for A1 alloy reveal minimal cracking and observable pile-up, supporting robust plastic resistance. **g** 3D AFM image of a 2 N indentation in the A1 alloy demonstrates symmetric pile-up morphology. Source data are provided as a Source Data file.

(Supplementary Fig. 23 and Supplementary Table 5). In contrast, the VIBANN framework combines a learned information-controlled latent geometry with explicit plausibility and epistemic risk constraints during candidate generation and refinement, yielding candidates that maintain conservative performance while achieving substantially greater separation from known alloys, as summarized in Supplementary Fig. 22 and Supplementary Table 5.

### Experimental validation of VIBANN-designed alloys

The five candidate compositions (designated A1–A5, Table 1) were experimentally investigated to test whether the predicted high hardness and load dependence translated to bulk MMGs under practical processing and measurement conditions. Figure 3 summarizes the synthesis and the structural, thermal, and mechanical characterization of the designed alloys.

All five compositions were fabricated by arc melting followed by suction casting, yielding 2 mm diameter rods with uniform surfaces and no macroscopic casting defects (Fig. 3a). Transmission electron microscopy (TEM) confirms an amorphous structure for each alloy, as indicated by the absence of contrast associated with crystalline grains and by diffuse halo rings in the selected area electron diffraction

(SAED) patterns (Fig. 3c and Supplementary Fig. 24). Consistent with TEM, X-ray diffraction shows broad amorphous maxima and no resolvable Bragg reflections within the measured 2θ range, supporting fully amorphous structures in the cast rods (Fig. 3b). Differential scanning calorimetry further reveals distinct glass transition events followed by crystallization, with measurable supercooled liquid regions for all compositions, indicating thermal stability in the undercooled regime (Fig. 3d). The measured glass transition temperatures ($T_g$), crystallization onset temperatures ($T_x$), and supercooled liquid region widths ($\Delta T_x$) for all five alloys are summarized in Table 1.

Hardness was evaluated by Vickers microhardness indentation over a broad load range from 0.5 N to 10 N to probe both hardness magnitude and the indentation size effect. Figure 3e compares measured hardness values with model predictions at each load. Across all alloys and loads, the measured values follow the predicted trends closely and remain within the prediction intervals. Deviations are generally within the combined experimental scatter and model uncertainty, indicating that the surrogate captures both the absolute hardness level and the load dependence. Because pile-up effects can bias the projected contact area and therefore hardness at higher loads,

we performed atomic force microscopy (AFM)-based pile-up correction using line profiles along the indent diagonals and a reconstructed contact perimeter for each load condition (Supplementary Note 9). Corrected hardness values are reported in Table 1, and representative topography and profiles are shown in Fig. 3g and Supplementary Figs. 26, 27.

Importantly, all five alloys exhibit exceptionally high hardness values, approaching the upper range reported for bulk amorphous alloys[36,37]. While a slight indentation size effect (ISE) is observed, as indicated by the decrease in hardness with increasing load, the designed alloys retain remarkably high hardness even at 10 N, with four of the five alloys remaining above 1700 HV. This robustness across loading scales affirms the intrinsic resistance of these alloys to plastic deformation. In particular, the alloy $B_{68}Nb_{24}Fe_4W_4$ (at.% composition) achieves the highest measured hardness, reaching $2447 \pm 44$ HV at 0.5 N. This value exceeds the typical hardness reported for bulk Zr-based MGs, and it is even higher than many Fe-based and Co-based bulk MGs that commonly show 11–13 GPa Vickers hardness[38–40]. Our bulk $B_{68}Nb_{24}Fe_4W_4$ alloy thus achieves hardness levels comparable to the upper range reported for certain hard thin-film MGs and hard ceramics, under comparable load and specimen constraints[35,41]. Furthermore, these alloy compositions were discovered autonomously by an AI-driven inverse design framework and realized in bulk form, representing a notable advancement over traditional serendipitous or exhaustive empirical screening approaches.

The deformation morphology around indents is consistent with the mechanical response expected for hard bulk MGs. Scanning electron microscopy (SEM) shows sharp indent boundaries with limited cracking, while shear offsets emanate from indent corners, consistent with localized plasticity mediated by STZs and shear banding (Fig. 3f, Supplementary Figs. 26, 27). The extent and geometry of these offsets were consistent across samples and matched well with the load-dependent AFM depth profiles (Supplementary Figs. 26, 27). Shear banding is a characteristic deformation mode in MGs, where inelastic atomic rearrangements localize into narrow regions under applied stress. Because MGs lack crystallographic slip systems, deformation cannot proceed by conventional dislocation glide and instead localizes into such narrow regions of high strain[8]. With increasing load, shear band traces extend further from the indent perimeter, indicating an expanded plastic zone, while the overall indent impressions remain well defined[42].

Load-depth curves further support the hardness ranking across the designed alloys (Supplementary Fig. 28). Among the five alloys, $B_{68}Nb_{24}Fe_4W_4$ (alloy A1) shows the shallowest penetration depths and the steepest loading slopes at each load, consistent with the highest resistance to plastic flow. This aligns with the absence of radial cracks or fracture impressions observed in the SEM images of the indentations (Fig. 3f). Such behavior is expected, given the high atomic packing density and strong interatomic bonding conferred by the presence of both Nb and W, which have high elastic moduli and cohesive energies[43]. Furthermore, boron (B) adds covalent character and contributes to structural stiffness by forming strong B-M bonds (where M = Co, Fe, Nb, W), which resist shear deformation[44,45]. The hysteresis between the loading and unloading curves in each plot quantifies the energy dissipated during deformation. All five alloys show moderate hysteresis, indicating the operation of STZs. Notably, the lack of extensive displacement excursions under high loads (5–10 N) across all curves suggests improved resistance to shear band proliferation compared to softer MGs, such as Zr- or Cu-based MGs, where excessive shear localization often leads to catastrophic failure even at low loads[46]. In contrast, the designed MMGs exhibit deformation features consistent with distributed plasticity, potentially indicating a favorable balance between strength and resistance to localized failure.

For comparison with the previously reported Fe-Nb-B thin-film MG literature[35], we examined the nominal $B_{72}Nb_{25}Fe_3$ ternary baseline alloy under the same bulk processing route used for our inverse-designed alloys. When prepared by arc melting followed by suction casting, this ternary composition did not vitrify as a bulk rod. Its XRD pattern shows sharp crystalline reflections, predominantly indexed to $NbB_2$, with no broad amorphous halo, indicating crystallization during casting (Supplementary Fig. 31). We further evaluated this baseline composition using the trained VIBANN model at representative loads of 0.5 N and 5 N and compared it with the experimentally validated candidates (Supplementary Note 10, Supplementary Table 8, Supplementary Fig. 29). In this bulk-load regime, the nominal ternary baseline alloy is predicted to have lower hardness than all five designed bulk MMGs. In addition, stepwise compositional paths from the ternary alloy toward representative designed bulk MMGs show monotonic increases in model-predicted hardness with progressive W or Hf/Ru addition (Supplementary Fig. 30). These observations indicate that the role of the added alloying elements is not only to modify hardness but also to shift the system from a crystallization-prone ternary alloy toward a bulk-realizable high-hardness amorphous region.

## Physical mechanisms underpinning hardness in designed bulk MMGs

Following the experimental validation of fully amorphous structures and the measured load-dependent hardness response of A1–A5 alloys, we used molecular dynamics (MD) simulations to identify the atomic-scale structural features associated with the observed hardness ranking. The five latent-optimized alloys, although obtained from different initializations, converge toward a B-Nb-Co-Fe-W-Hf-Ru-Zr-rich compositional domain. This convergence is non-trivial because W and Ru occur infrequently in the training set yet are repeatedly selected among the refined alloy candidates. All five alloys exhibit experimentally validated peak Vickers hardness above 1700 HV, with A1 reaching approximately 2450 HV, placing it among the highest hardness values reported for bulk MMGs.

Before structural interpretation, we validated the MD-simulated atomistic configurations against high-energy X-ray scattering experiments. Figure 4a and Supplementary Figs. 32–35a show that the MD-simulated total radial distribution functions reproduce the experimental synchrotron-derived g(r), and the corresponding experimental structure factor S(Q) is shown in the insets. This agreement supports the use of simulated structures for quantitative analysis of short- and medium-range order in A1–A5 alloys.

The total radial distribution functions (RDFs) in Fig. 4a and Supplementary Figs. 32–35a exhibit high-intensity primary peaks near ~2.3–2.5 Å and slightly broadened secondary peaks, consistent with dense short-range order and a disordered medium-range environment. These characteristics are consistent with topologically disordered, yet densely packed, amorphous matrices, ideal for impeding shear transformations. Crucially, the full-width-at-half-maximum (FWHM) of the first peak narrows with increasing hardness across alloys, indicating decreasing bond length disorder. Figure 5a–d compare the area under the primary peak, primary peak height, its FWHM, and the height ratio between the first and second peaks. A1 shows the largest first peak height, the highest peak sharpness as quantified by height divided by FWHM, and the largest first-to-second peak height ratio. These trends collectively indicate a tighter distribution of near-neighbor separations and a more uniform first-neighbor shell, suggesting low free volume and limited room for shear transformations. In contrast, alloy A5, despite being compositionally similar, shows the lowest values across all three metrics, suggesting a relatively more diffuse local environment. Alloys A2, A3, and A4 exhibit intermediate peak ratios, but alloy A4 shows a notable dip in primary peak sharpness, correlating with its slightly lower hardness compared to alloy A1.

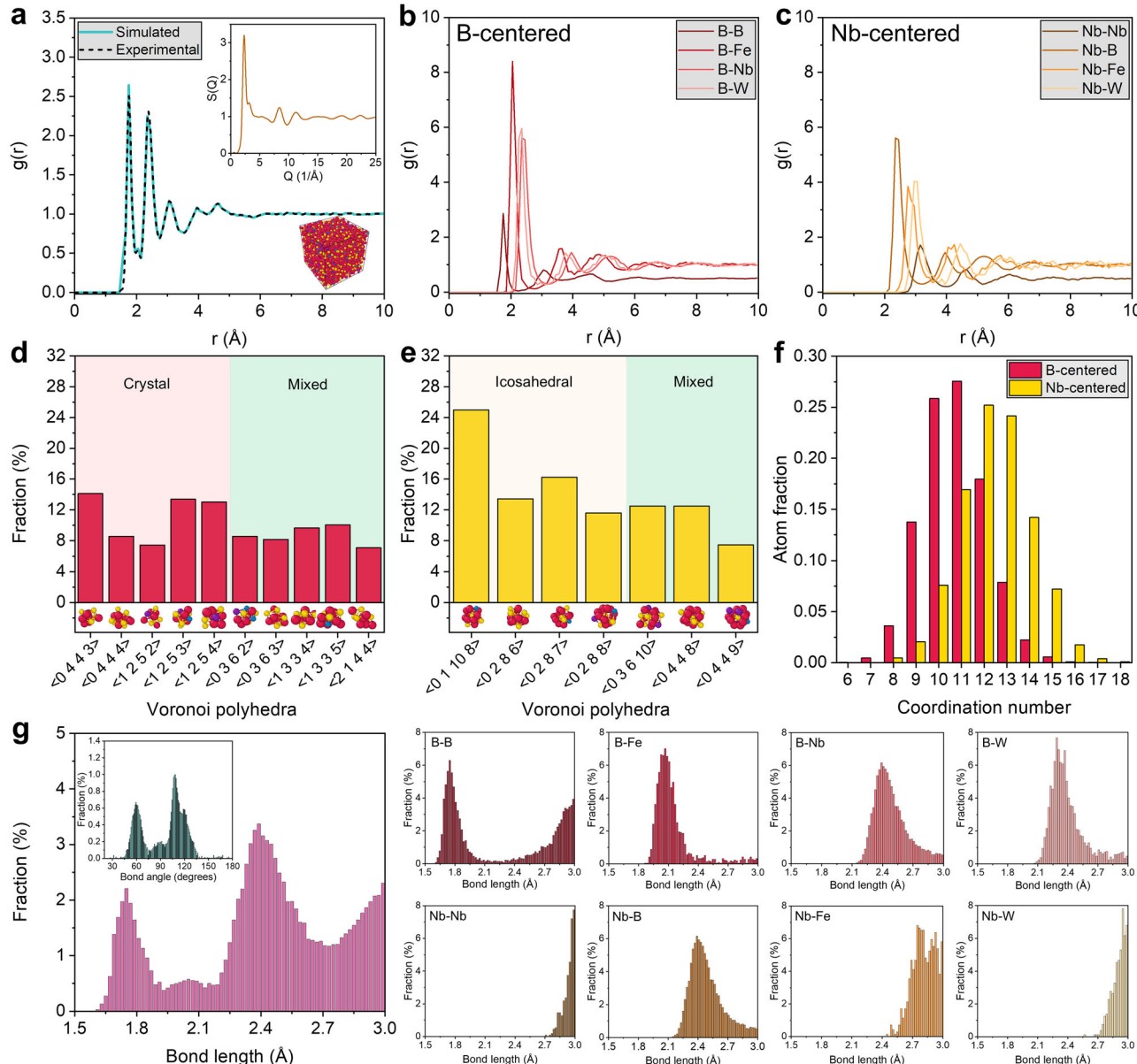

**Fig. 4 | Atomic-scale structural analysis of A1 alloy. a** Simulated and experimental radial distribution functions (RDFs) showing excellent agreement, validating structural fidelity; the **inset** shows the experimental structure factor S(Q) and a representative atomic model. **b**, **c** Partial RDFs for B-centered and Nb-centered clusters, respectively, highlighting distinct local environments around each species. **d** Coordination number distributions for B- and Nb-centered clusters reveal predominant 11-, 12- and 13-fold packing motifs. **e**, **f** Voronoi polyhedra statistics for B- and Nb-centered atoms show dominant crystal-like motifs and icosahedral order, suggesting competing short- and medium-range orders. **g** Total and partial bond-length distributions for atomic pairs indicate well-defined bonding shells and species-specific bonding characteristics; the **inset** in the left panel shows the bond-angle distribution with peaks near 109.5° (icosahedral) and 60° and 120° (planar motifs), reinforcing local order. Source data are provided as a Source Data file.

Voronoi analysis further differentiates the alloys based on local topology. Figure 4d, e and Supplementary Figs. 32–35d, e show that the harder alloys (A1 and A4) contain higher fractions of icosahedral and distorted crystal-like environments, which are widely associated with increased resistance to local shear due to topological frustration and efficient packing. Figure 5h, i reveals that alloys A1 and A3 contain the highest fraction of icosahedral clusters (~0.44 and ~0.52, respectively), followed closely by A4, while alloy A5 again shows the lowest (~0.35). Similarly, crystal-like clusters (e.g., distorted FCC-like) are enriched in alloys A1 and A4, indicating a dual presence of locally ordered and topologically frustrated motifs. A4 MMG shows a moderate fraction of both cluster types, while alloy A5 exhibits a diminished signature of either. The coexistence of icosahedral frustration (preventing shear localization) and locally ordered motifs (enhancing

packing density) in A1 and A4 is consistent with greater resistance to incipient plasticity.

The coordination environment analysis (Fig. 4f and Supplementary Figs. 32–35f) further delineates structural differences. Alloy A1 has the highest total atomic fraction with coordination number (CN)=12 or 13 (~0.34), while A5 exhibits the lowest fraction (~0.24). The fraction of B-centered environments with coordination number 12 or 13 is highest in A1 and A4, followed by A2 (Fig. 5e–g). This observation suggests that B atoms in well-packed 12-fold configurations act as nanoscale pinning centers that resist deformation. In contrast, alloy A5 exhibits the lowest B-centered coordination, indicating a disrupted or less-optimized local network. Nb-centered clusters with CN = 12 or 13 show the same ordering, peaking for alloys A1 and A2. The concurrent enrichment of well-packed B-centered and Nb-centered environments

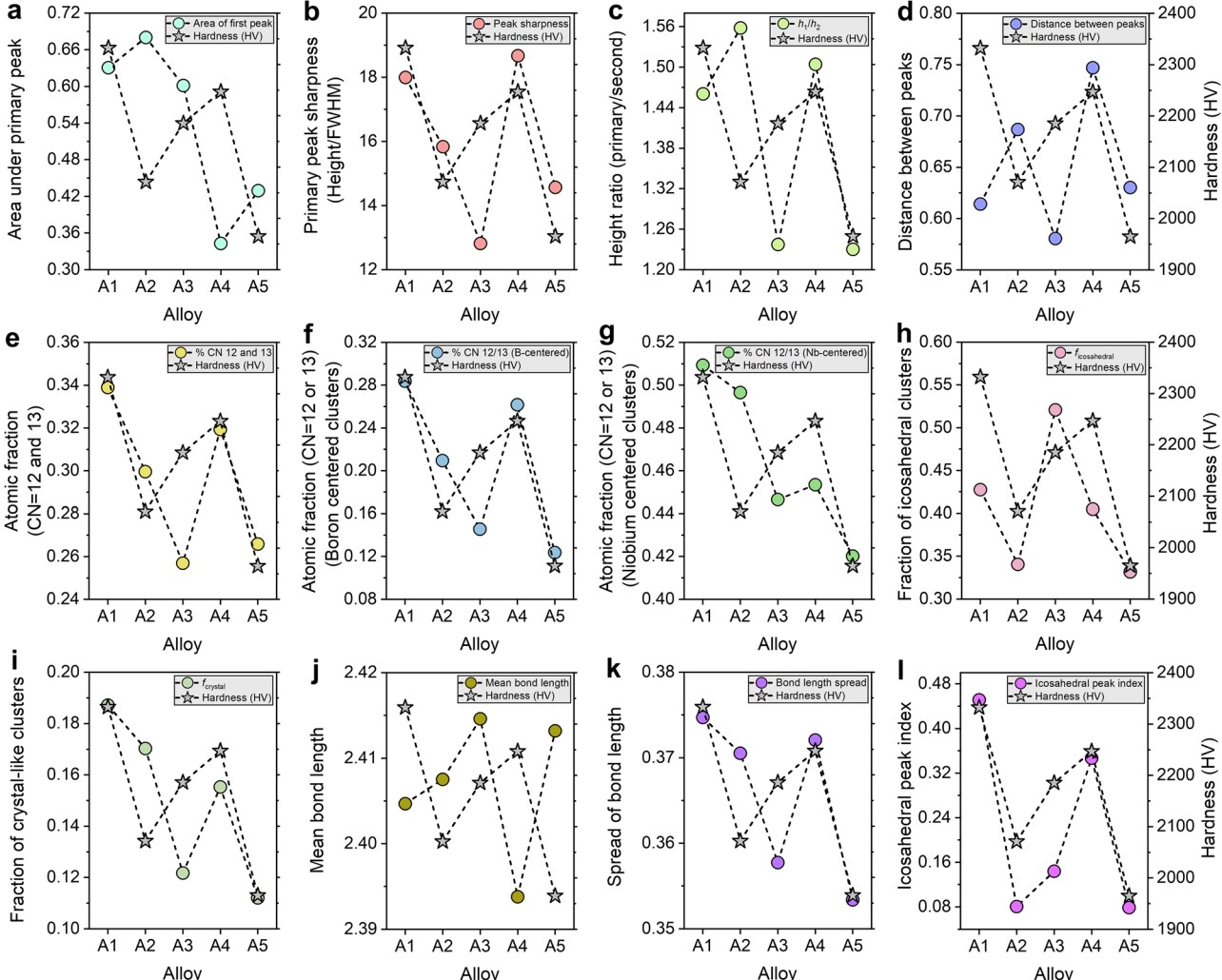

**Fig. 5 | Correlation between atomistic structural descriptors and experimentally measured hardness in the designed bulk multicomponent metallic glasses. a–d** Descriptors derived from the total radial distribution function (RDF): (**a**) area under the primary peak; (**b**) primary peak sharpness, defined as peak height divided by full-width-at-half-maximum (FWHM); (**c**) height ratio between the first and second RDF peaks ($h_1/h_2$); and (**d**) separation between the first and second RDF peaks. **e–g** Fractions of densely coordinated environments with coordination number (CN) 12 and 13: (**e**) total atomic fraction; (**f**) B-centered fraction; and (**g**) Nb-centered fraction. **h, i** Fractions of icosahedral and crystal-like clusters obtained

from Voronoi analysis. **j, k** Mean bond length and bond-length spread, respectively. **l** Icosahedral peak index. In each panel, colored circles represent the structural descriptor for alloys A1–A5, whereas gray star symbols connected by dashed lines denote the experimentally measured Vickers hardness of the corresponding alloys. Together, these comparisons show that higher hardness is associated with tighter short-range packing, larger fractions of densely coordinated and icosahedral environments, and stronger local structural regularity. Source data are provided as a Source Data file.

in the harder alloys is consistent with a densely connected local atomic network that raises the resistance to the initiation of shear transformation activity.

Bond length and bond angle distributions further corroborate these observations. Figure 4g and Supplementary Figs. 32–35g show narrower bond length distributions in the harder alloys. Figure 5j–k shows that A4 has the shortest average bond length (~2.395 Å) and a relatively narrow spread, consistent with a stiffer, more homogeneous bonding network. In contrast, alloys A3 and A5 exhibit the longest average bond lengths, consistent with a more disordered, less stiff network. A1 and A2 MMGs also exhibit short bond lengths with moderate spreads. These trends underscore that tight bond networks, both in terms of length and uniformity, are among the central features that govern hardness.

Bond angle distributions (Fig. 4g and Supplementary Figs. 32–35g) exhibit pronounced features near 60°, 109°, and 120°, and their sharpness varies systematically across alloys, indicative of tetrahedral and distorted octahedral configurations. A1 alloy exhibits

the highest index (~0.46), followed closely by A4, suggesting strong local angular rigidity, a known factor that impedes STZ activation. Alloy A5, in contrast, shows the flattest and broadest angular distribution (~0.08), suggesting a flexible, easily deformable STZ network (Fig. 5l).

Across all structural measures, A1 ranks highest in first-neighbor shell sharpness, fraction of densely coordinated environments, abundance of icosahedral and crystal-like motifs, and bond length and bond angle uniformity. A4 follows closely, with strong bonding, angular rigidity, and substantial enrichment of dense local environments, while A2 and A3 show intermediate behavior. A5 shows systematically weaker signatures of dense packing and topological order despite compositional proximity, indicating that modest compositional shifts within a similar chemistry window can produce measurable changes in local structure and hardness.

Together, the experimental measurements and atomistic analysis show that the inverse-designed compositions achieve exceptionally high hardness in bulk amorphous form, and that hardness trends

correlate with structural features, including increased local packing uniformity, enrichment of densely coordinated environments, and higher fractions of icosahedral and crystal-like motifs.

## Discussion

We established an experimentally validated inverse design framework for exceptionally hard bulk MMGs, in which a variationally regularized latent representation enables candidate generation, explicit feasibility control, and latent space refinement. The results demonstrate high predictive accuracy, uncertainty calibration under conservative scoring, and successful synthesis of five designed alloys with exceptionally high hardness. We now discuss the information encoded by the VIBANN framework, focusing on the suitability of the latent landscape for traversal and refinement, the elemental dependencies learned by the model, and their consistency with experimental observations and MD-derived atomistic signatures.

To examine how the learned latent representation organizes hardness-relevant information, we performed latent-traversal analyses, perturbing a single latent coordinate while holding the others fixed. For each reference point, 25 evenly spaced perturbations were applied per coordinate over the same latent range, and the resulting hardness was evaluated using the trained VIBANN predictor. Figure 6a shows the traversal responses for the most hardness-sensitive coordinates, defined as those producing the largest change in predicted hardness over the perturbation range. Figure 6b shows the corresponding set of moderately sensitive coordinates, and Fig. 6c shows the low-sensitivity coordinates whose traversal curves remain close to flat. In all three panels, the curves are evaluated from the same high hardness reference point, so differences across panels reflect differences in how strongly each latent direction controls hardness rather than differences in baseline chemistry. Figure 6d provides the compact quantitative summary of this result by reporting, for each coordinate, the traversal-induced hardness variation. Supplementary Fig. 36 shows the same traversal analysis for a randomly chosen latent vector from the center of the latent space. In both cases, the hardness varies strongly along only a small subset of coordinates, with $z_3$, $z_5$, $z_{12}$, $z_{13}$, and $z_{15}$ producing the largest and most systematic changes (Fig. 6a). Several additional coordinates exhibit weaker but directionally consistent curvature (Fig. 6b), whereas other coordinates yield nearly flat responses over the same perturbation range (Fig. 6c), indicating that a large fraction of latent variability is not hardness-determining in the local neighborhood of a high hardness solution. The consistency of these trends across distinct starting compositions indicates that the latent directions encode global factors governing hardness rather than idiosyncratic variations specific to a given alloy. The traversal curves are smooth over the explored range, indicating a locally continuous mapping from latent coordinate to hardness, which is required for stable refinement when inverse design updates follow gradients in latent space.

To test whether the same coordinates dominate for the complete dataset, rather than near a single high-hardness reference, we evaluated global associations between latent coordinates and predicted hardness across the full set of encoded samples. Supplementary Fig. 37a reports Pearson and Spearman correlations between each latent coordinate and hardness, showing that the strongest correlations again concentrate in the same small subset highlighted by the traversal analysis. We then computed gradient-based sensitivities by averaging the absolute partial derivatives $|\partial Hardness/\partial z|$ with respect to each latent coordinate across 1000 latent samples, as shown in Supplementary Fig. 37b. This provides a complementary measure that does not depend on the finite traversal range and again identifies a small group of dominant coordinates whose sensitivities exceed the remainder by a large margin. The agreement between Fig. 6d and Supplementary Fig. 37a, b indicates that the VIB-regularized representation concentrates hardness controlling information into a limited number of latent degrees of freedom, consistent with the intended information bottleneck behavior.

The remaining latent coordinates (Fig. 6c) define a hardness-insensitive subspace over the local neighborhoods probed here. Variations along these directions can change composition without substantially affecting predicted hardness, consistent with the existence of composition families that lie on approximately iso-hardness manifolds. In metallurgical terms, these directions can be interpreted as encoding compensating trade-offs among elements with broadly similar effects on hardness within the chemistry space represented by the data. This separation is practically useful because it implies that hardness can be adjusted through a small number of controlling latent directions, while leaving additional latent degrees of freedom for tuning secondary considerations, subject to constraints imposed by glass formation and processing feasibility.

Beyond individual latent dimensions, we examined the global organization of the latent space using principal component analysis (PCA). Supplementary Fig. 37c shows that the first principal component ($PC_1$) explains the majority of latent variance and correlates strongly with hardness ($R^2 > 0.95$), whereas Supplementary Fig. 37d shows that the second principal component ($PC_2$) explains little variance and has negligible association with hardness. This indicates that the dominant axis of variation in the learned representation aligns with the hardness gradient, while the remaining variance largely reflects latent variability that is orthogonal to hardness.

Taken together, these results show that the VIB-regularized latent representation is structured such that hardness varies primarily along a small set of latent directions and, to first order, along a single dominant latent axis. This property-organized structure provides a quantitative basis for treating the latent space as a landscape that supports controlled traversal and gradient-based refinement without inducing unstable or discontinuous changes in predicted hardness over the local ranges relevant to inverse design. While the exact metallurgical quantities underlying each coordinate (e.g., bond energy concentration, atomic mismatch, VEC) remain open to interpretation, the predictability and consistency of their effects make them powerful tools for alloy exploration. Importantly, this transforms the latent representation from an abstract vector encoding to a navigable design landscape, with clear trajectories for optimizing hardness and other orthogonal properties simultaneously.

Beyond candidate generation, VIBANN provides an interpretable view of the compositional features that govern high hardness within the bulk-MMG domain represented by the dataset. Figures 6e, f summarize the elemental dependencies that the trained model associates with high hardness within the bulk MMG domain represented in the dataset. The attention ranking indicates which elemental fractions the network emphasizes when constructing the hardness-relevant internal representation. Integrated gradients provide a signed attribution that quantifies the local change in the predicted hardness for an incremental change in an elemental fraction, evaluated under the composition sum constraint. Consistency between attention and integrated gradients is therefore important because it indicates that the model not only attends to a feature but also uses it in a directionally consistent manner to raise the predicted hardness across a broad set of compositions.

Within this domain, boron is repeatedly identified as a primary hardness-promoting constituent. This outcome is consistent with established bulk MG design considerations. Boron strengthens many MG networks through strong heteroatomic bonding with transition metals and through a large local size contrast that increases packing frustration. These effects reduce the population of easily dilatable atomic environments and increase the stress required to activate shear transformations[47–49]. In addition, boron-containing BMGs are disproportionately represented among the highest-hardness reports in the available literature, particularly when combined with selected

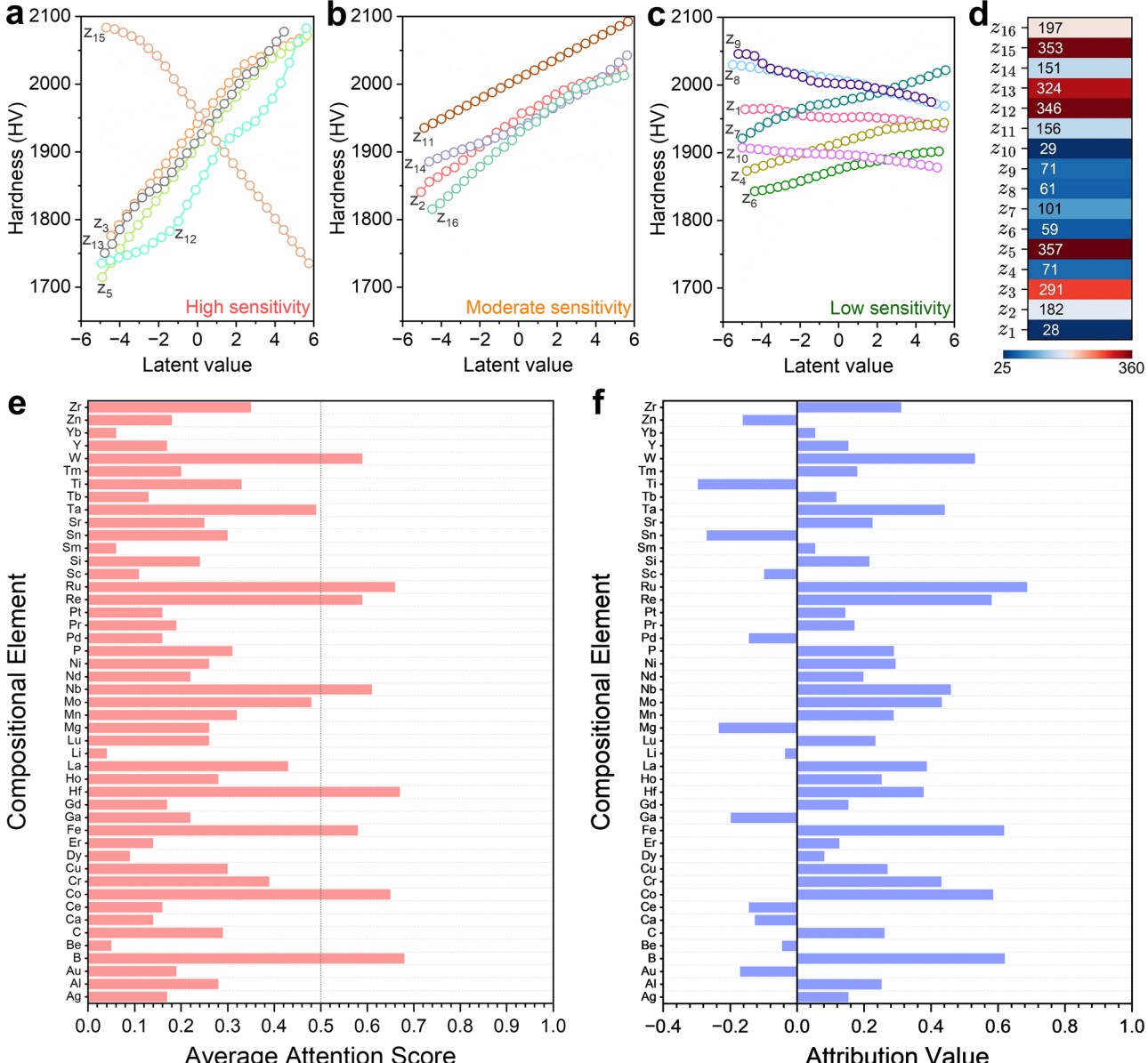

**Fig. 6 | Latent space sensitivity and element-level explainability of the VIBANN framework. a–c** Latent traversals for a representative latent vector sampled from the high hardness region of the learned manifold. Each curve is obtained by perturbing a single latent coordinate over the same range while holding all remaining coordinates fixed and evaluating the predicted Vickers hardness. For clarity, the traversals are grouped by response strength into (**a**) high, (**b**) moderate, and (**c**) low sensitivity sets, showing that hardness variations are controlled by a limited subset of latent degrees of freedom, whereas most coordinates induce only weak changes over the same perturbation range. **d** Summary heatmap quantifying the traversal amplitude for each latent coordinate, reported as the peak-to-peak change in predicted hardness across the sweep, confirming that the dominant hardness response is concentrated in a small subset of latent coordinates. **e** Average attention scores derived from the attention layer of the VIBANN model highlight the most influential elements contributing to hardness prediction. **f** Integrated gradients attribution values for elemental fractions, quantifying the signed influence of small increases in elemental content on the predicted hardness under the simplex constraint, where positive values promote and negative values reduce hardness. Source data are provided as a Source Data file.

transition metals and refractory additions, while retaining glass-forming feasibility. The VIBANN model, therefore, learns a robust association between boron enrichment and the high-hardness tail of the BMG distribution, and the experimentally validated alloys in this work (Fig. 3, Table 1) lie within the same boron-containing regime highlighted by the interpretability maps.

The interpretability analyses also emphasize a set of refractory, high-stiffness transition-metal additions, including W, Ru, Re, Ta, Hf, Mo, and Nb. These elements are highlighted even at modest atomic fractions, consistent with a strengthening role arising from increased elastic stiffness, chemical heterogeneity, and local strain heterogeneity in the amorphous network[43,50]. In BMGs, such heterogeneity

is relevant because it increases the barrier for cooperative atomic rearrangements and limits the growth of a shear transformation into a propagating shear band. The modeling results support this interpretation. The highest hardness alloys exhibit the strongest signatures of a rigid short-range structure, including sharper short-range features, higher fractions of densely coordinated environments, and narrower distributions of bond-length and bond-angle metrics (Figs. 4 and 5). These trends are consistent with reduced local compliance and a higher resistance to shear transformation activation. The experimental and modeling outcomes, therefore, align with the interpretability results, in that the alloying directions emphasized by the VIBANN framework are realized in the synthesized

compositions and are accompanied by atomistic indicators of increased short-range rigidity.

Elements such as Fe, Co, and Nb also receive consistently positive attribution and close attention scores. These elements are common constituents of strong glass-forming alloys and contribute through high cohesive energy, metallic bonding, and favorable packing with B[48,51]. In particular, high valence-electron counts in Nb, Co, and Fe are associated with strong interatomic bonding and resistance to shear, which may be captured by VIBANN's attention to high valence elements[52]. This suggests that one of the latent dimensions may encode effects related to valence electron concentration (VEC), with the attention mechanism subsequently amplifying the contribution of high-VEC elements when the model predicts high hardness.

Elements that rank low in attention or exhibit weak or negative integrated gradients are also consistent with established behavior of BMG families that are comparatively soft under similar testing conditions. Rare-earth-dominated MGs and Mg or alkaline-earth-rich MGs often exhibit lower hardness because their bonding is more compliant and their elastic stiffness is lower, which reduces the resistance to local dilation and shear transformation activation. Their large atomic volumes and packing characteristics can also increase free volume, which facilitates local rearrangements during indentation[53]. Yb-based and Ce-based bulk MMGs are known for very low glass transition temperatures and poor hardness (~1.5 GPa)[54]. These elements have a low valence electron count involved in bonding (Ce effectively contributes 3 electrons, Yb contributes 2, compared to transition metals contributing ~4–10), which means that the metallic bonds in RE-based bulk MGs are weaker (lower electron density). They also have large atomic sizes and low packing efficiencies, which introduce excess free volume and make the bulk MG more deformable[55,56]. Noble metal-containing bulk MGs can form a stable amorphous matrix in specific systems, but they do not generally raise hardness unless paired with sufficient metalloid content or strong bonding additions that increase short-range rigidity[57]. These tendencies are reflected in the sign and magnitude of the integrated gradients, indicating that the VIBANN model has potentially learned chemically consistent strengthening and softening directions within the BMG domain defined by our data space.

It is important to note that the attention, attribution, and latent-space analyses should be interpreted as structure-property correlations learned within the restricted chemical and processing space defined by the present dataset, rather than as universally transferable laws for all amorphous materials. In this context, the model's preference for boron-rich compositions does not imply that it has learned the general statement that stronger covalent bonding always yields higher hardness. The more precise interpretation is that, within the bulk MMG space sampled here, boron is the most prevalent small metalloid associated with increased bond directionality, enhanced local short-range rigidity, and continued compatibility with bulk MG formation. The model, therefore, identifies boron-rich and refractory-stabilized compositions as the dominant high-hardness direction within the present bulk MMG manifold, rather than inferring that all amorphous systems with stronger covalent bonding should be selected.

The carbon-based comparison reinforces this point. Carbon-bearing MG systems are known to exhibit high hardness[28], but in MGs, carbon acts within a metal-rich amorphous matrix under the coupled constraints of bulk glass formation, local packing, short-range chemical order, and competing crystallization[9,48]. The relevant question for the present framework is therefore not whether carbon forms strong bonds in general, but whether carbon-containing compositions occupy a high-hardness and glass-formable region within the specific bulk MG manifold represented in the dataset. Within our design space, boron remains the more consistent high-hardness direction identified

by the model, indicating that VIBANN is learning domain-bounded alloying correlations rather than a universally transferable law.

These considerations define the principal interpretive boundary of the framework. The attention and integrated-gradient patterns are most reliable for chemically adjacent exploration within the bulk MMG manifold represented by the dataset. They should not be interpreted as predictions for chemically distinct amorphous classes outside that space. Within the bulk MMG domain, however, the combined evidence from attention and attribution, uncertainty-penalized candidate selection, experimental validation, and MD-based structural analysis supports a coherent picture. The highest-hardness compositions identified here correspond to densely packed amorphous networks in which boron-enriched short-range environments are reinforced by refractory and transition-metal additions that increase local stiffness and resistance to shear transformation activation. Consistent with this domain-bounded interpretation, the additional leave-family-out validation on a hard non-boride family and the boron-free constrained inverse-design results show that the learned hardness trends are not confined to a single boride-enriched basin, but extend to chemically distinct non-boride regions within the dataset-supported bulk-MMG manifold (Supplementary Fig. 38, Supplementary Table 9).

In summary, this study establishes VIBANN as an uncertainty-aware closed-loop inverse design framework for discovering exceptionally hard bulk MMGs under limited data conditions. Unlike traditional machine learning approaches, the VIBANN framework, by combining variational information bottleneck learning, attention-based composition encoding, latent-space-constrained candidate generation, and experimental validation, moves beyond forward prediction and enables closed-loop alloy discovery within a bounded bulk MMG design space. Through a combination of probabilistic sampling and latent space optimization, VIBANN identified five new B-Nb-Fe-W-Co/Hf/Ru/Zr-rich bulk MMG candidates with predicted hardness above 2200 HV. Experimental synthesis confirmed fully amorphous 2 mm rods and Vickers hardness up to about 2450 HV, placing these alloys among the highest reported for bulk MMGs. Notably, these alloys emerged despite the underrepresentation of refractory elements in the training data, underscoring the framework's ability to conservatively extrapolate metallurgically sound solutions beyond interpolation regimes. The learned attention, attribution, and latent-space trends define domain-bounded structure–property correlations within the bulk MMG manifold represented by the dataset. Combined with experimental validation, and MD-based atomistic calculations, these trends indicate that high hardness is associated with dense packing, boron-enriched short-range environments, and refractory-stabilized local rigidity. Taken together, this study shows that predictive modeling, uncertainty-aware search, experimental validation, and atomistic analysis can be integrated into a single workflow for the discovery of multi-component amorphous alloys under sparse data conditions.

## Methods

### Dataset curation and representation
A curated dataset of 673 bulk MG compositions with reported Vickers hardness (HV) values and indentation loads was compiled from peer-reviewed literature[58]. Each sample was represented by a 56-dimensional simplex-normalized composition vector together with the applied indentation load as a scalar input. Supplementary Note 1 and Supplementary Figs. 1, 2 provide detailed exploratory data analysis (EDA) of this dataset. Elemental fractions were projected onto the simplex to enforce non-negativity and unit sum. The indentation load was standardized using statistics computed from the training data, and hardness was standardized to a zero mean and unit variance for model training, with all inverse transformations applied only for reporting in physical units.

## Train-test splitting and sample weighting

To reduce label leakage from near-duplicate chemistries and preserve compositional diversity across splits, the dataset was partitioned using cluster-aware stratification based only on composition. Normalized composition vectors were clustered by K-means with $k = 3$, and the resulting cluster labels were used in a stratified shuffle split (Supplementary Note 2) to generate an 80% training set and a 20% test set, corresponding to 538 and 135 alloys, respectively. Supplementary Fig. 3 confirms compositional parity between splits.

Within the training set, 20% of the samples were reserved as a calibration subset for uncertainty calibration and threshold setting in the inverse-design stage. The remaining training samples were used for model fitting.

During training, per-sample weights were used to reduce the overdominance of the most common compositional constituents. Samples containing the most frequently occurring elements were down-weighted by a factor of 0.35, and the final weights were clipped to the interval 0.25 to 4.0.

## VIBANN architecture

The VIBANN model takes two inputs. The composition input is a 56-dimensional vector. The load input is a single standardized scalar. The composition pathway begins with two dense layers (128- and 64-neuron with ReLU activation), each followed by batch normalization and Monte Carlo dropout, and includes a self-attention block that produces an element-reweighted representation used downstream. The load pathway uses a lighter, 32-neuron stack with batch normalization and Monte Carlo dropout. The two pathways are concatenated and passed to a variational information bottleneck (VIB) layer, which outputs a latent vector $z \in \mathbb{R}^d$, parameterized by $\mu(x)$ and $\log \sigma^2(x)$, using the standard reparameterization trick.

**Variational bottleneck and KL per dimension tracking.** The VIB layer computes the Kullback-Leibler (KL) divergence penalty between the approximate posterior $q_\phi(z|x)$ and a standard normal prior $p(z) = \mathcal{N}(0, I)$, normalizes it by the latent dimensionality $d$ (optimized to 16 using Bayesian hyper-parameter search), and adds it to the model loss through the layer-level add-loss mechanism. The KL penalty encourages the latent distribution to approximate a prior Gaussian distribution while an adaptive weighting factor $\gamma$ balances this regularization against the task-specific prediction loss[59]. Mathematically, this trade-off is formalized through the optimization of an Information Bottleneck Lagrangian:

$$\mathscr{L}_{VIB} = \mathbb{E}_{p(x,y)}\left[\mathbb{E}_{q(z|x)}[log p(y|z)]\right] - \gamma \mathbb{E}\left[\frac{1}{d}D_{KL}\left(q_\phi(z|x)||p(z)\right)\right] \quad (1)$$

where $\gamma$ controls the strength of the bottleneck and $p(z)$ denotes a prior distribution, typically standard Gaussian. The first term ensures predictive fidelity by encouraging accurate recovery of the target property, while the second term imposes information compression and structure in the latent space.

**Latent to composition decoder head.** A composition reconstruction head branches directly from the latent vector. It maps $z$ to a simplex-normalized composition $\hat{x}$ using two fully connected layers with 128 and 64 hidden units, ReLU activations, each followed by batch normalization and Monte Carlo dropout, and a final *softmax* layer with 56 outputs. This head is jointly trained with the hardness predictor and used as the generator in inverse design.

**Hardness prediction head and attention outputs.** The hardness prediction head maps the latent representation to a standardized hardness prediction $\hat{y}_s$. The VIBANN model also exports attention tensors as additional outputs for interpretability. These attention outputs are not supervised and carry a zero-loss weight during training.

The final decoder maps the latent representation to a 56-component simplex-normalized composition vector, ensuring non-negativity and unit-sum normalization of decoded candidate alloys during inverse design.

## Training objective and adaptive information budget

The total loss minimized during training is the sum of three terms.

Supervised hardness regression loss

$$\mathscr{L}_{HV} = MSE(y_s, \hat{y}_s) \quad (2)$$

Composition reconstruction loss for the decoder head

The decoder is trained with a KL divergence reconstruction objective

$$\mathscr{L}_{recon} = D_{KL}(x||\hat{x}) = \sum_i x_i log\left(\frac{x_i}{\hat{x}_i}\right) \quad (3)$$

weighted by a fixed coefficient $\lambda_{recon} = 0.5$.

VIB regularization

$$\mathscr{L}_{VIB} = \mathbb{E}_{p(x,y)}\left[\mathbb{E}_{q(z|x)}[log p(y|z)]\right] - \beta \mathbb{E}\left[\frac{1}{d}D_{KL}\left(q_\phi(z,|,x)||p(z)\right)\right] \quad (4)$$

where the KL per dimension is computed in the VIB layer.

Accordingly,

$$\mathscr{L}_{Total} = \mathscr{L}_{HV} + \lambda_{recon}\mathscr{L}_{recon} + \mathscr{L}_{VIB} \quad (5)$$

**Beta controller for KL per dimension.** Rather than using a fixed $\beta$ schedule, $\beta$ is updated during training by a feedback controller that aims to maintain a target KL per dimension of 0.15. After each epoch, $\beta$ is multiplied by 1.05 if the observed KL per dimension exceeds the target and divided by 1.05 otherwise, subject to minimum and maximum bounds. This dynamic $\beta$-annealing strategy ensures that latent compression is prioritized during the initial training epochs, gradually transitioning to fine-tuning the predictive task[26]. This stabilizes the information budget across folds and hyperparameter settings.

## Training protocol and hyperparameter search

Models were trained using the Adam optimizer for up to 1000 epochs with early stopping (patience 100) and monitored on a validation split. Mini-batches of 16 samples were used with an initial learning rate of $10^{-4}$. We trained the entire workflow on 10 random seeds and report seed-averaged performance to improve robustness. Hyperparameter selection for latent dimensionality and dropout rate was performed using Optuna with a TPE sampler, optimizing validation loss across 50 Bayesian optimization trials. The best VIBANN configuration had a latent dimensionality of 16, a dropout rate of 0.33, and 4 attention heads. The specific hyperparameter combinations and their validation losses are summarized in Supplementary Table 2. Extended theoretical discussion of the variational information bottleneck and attention formulation is provided in Supplementary Note 3.

## Monte Carlo dropout uncertainty estimation

Monte Carlo dropout was used to estimate epistemic uncertainty while keeping batch normalization layers frozen in inference mode. For standard model evaluation and for the uncertainty intervals reported with predicted hardness values, 100 stochastic forward passes were performed for each input to obtain the predictive mean $\mu_{HV}$ and predictive standard deviation $\sigma_{HV}$. For uncertainty calibration at the design load during inverse design, 300 stochastic forward passes were

performed on the calibration subset reserved from the training data. Conservative candidate ranking used a one-sided lower confidence bound defined as $LCB = \mu_{HV} - z\sigma_{HV}$, with $z = 1.645$. Independent random seeds were used across stochastic forward passes to avoid correlation between uncertainty samples.

### Latent space analysis

Latent embeddings were extracted by passing inputs through the trained VIB encoder to obtain the bottleneck latent vector. Two-dimensional t-SNE projections were used only for visualization of latent organization. K-means clustering was used to summarize latent partitions, whereas latent-space support for inverse design was modeled in the full latent space using a Gaussian mixture model. The number of mixture components was selected by Bayesian information criterion, and the final latent-support model used three components. Additional latent traversals, latent-gradient analyses, and latent-hardness correlation diagnostics are provided in Supplementary Note 6.

### Inverse design of exceptionally hard bulk MMGs

A salient feature of this framework is the encoder-decoder inverse design loop. Inverse design was carried out at a fixed design load of 0.5 N, and the final candidate alloys were subsequently evaluated across 0.5–10 N. The pipeline consists of four steps: latent density modeling, mixture-aware multi-chain Markov chain Monte Carlo (MCMC) sampling in the latent space, gradient-based refinement of selected seeds, and uncertainty-gated down selection.

**Latent density model.** A three-component (selected by Bayesian information criterion) Gaussian mixture model was fitted to the latent embeddings of the training set evaluated at the design load. The fitted model provides an explicit latent density $p_{GMM}(z)$ and an operational notion of data support through the latent log-likelihood.

A feasibility threshold on latent support was defined as

$$logp_{GMM}(z) \geq logp_{thr} \tag{6}$$

where $logp_{thr}$ was set to the 10th percentile of training log-likelihoods.

**Uncertainty calibration and conservative performance targets.** Uncertainty calibration was performed using MC dropout on the calibration subset of the training data at the design load. The procedure used 300 stochastic forward passes and up to 5000 samples, then computed $\mu_{HV}$, $\sigma_{HV}$, and the one-sided lower confidence bound

$$LCB(z) = \mu_{HV}(z) - z\sigma_{HV}(z) \tag{7}$$

with $z = 1.645$.

A data-anchored robustness floor was defined from training statistics at the design load. First, a near-peak subset was identified as samples with $\mu_{HV}$ above the 98th percentile. Then the target lower bound was set to the 85th percentile of LCB values in this near-peak subset. A reference uncertainty scale was defined as $\sigma_{ref,q80}$, the 80th percentile of $\sigma_{HV}$ values in the same calibration subset.

**Mixture-aware multi-chain MCMC in latent space.** Latent exploration was performed using a mixture-aware multi-chain MCMC scheme. Proposals combined global moves based on mixture components with local random-walk perturbations, and feasibility checks were applied using the latent log-likelihood threshold. Each proposed latent vector was decoded to a composition using the trained decoder, projected to the simplex, and evaluated with a small number of MC dropout passes inside the MCMC loop.

Uncertainty enters the MCMC target in a risk-aware utility

$$U(z) = \mu_{HV}(z) - z_u\sigma_{HV}(z) \tag{8}$$

with $z_u = 1.02$ used inside the sampler. The sampler also enforces soft robustness through penalties tied to target LCB (80th percentile of LCB values) and includes additional terms that discourage samples from collapsing into a small region of latent space.

**Gradient-based refinement.** A subset of high-quality latent seeds was refined by gradient-based optimization of a composite objective that rewards high hardness while penalizing leaving the high support region of the latent GMM and violating robustness. The refinement updates were performed directly in the latent space using back-propagation through the decoder and the hardness head. The Adam optimizer was employed with a learning rate of $1e^{-4}$, and early stopping is applied to ensure efficient convergence.

**Down selection with feasibility, novelty, and uncertainty gates.** Candidates decoded from sampled and refined latent points were subjected to an acceptance mask that required simultaneous satisfaction of four criteria.

Latent support

$$logp_{GMM}(z) \geq logp_{thr} \tag{9}$$

Novelty in composition space

Novelty was quantified by the minimum Euclidean distance to the training set

$$d_{min}(x) = \min_{x_i \in \mathscr{D}_{train}} ||x - x_i||_2 \tag{10}$$

with the novelty threshold set to the 80th percentile of reference training spacing distribution.

Uncertainty cap

$$\sigma_{HV}(z) \leq 2.5\sigma_{ref,q80} \tag{11}$$

Conservative hardness requirement

$$LCB(z) \geq LCB\_TARGET \tag{12}$$

with LCB_TARGET calibrated from the training distribution at the design load.

Candidates that passed this acceptance mask were ranked and reported for experimental validation.

### Model interpretability

Attention maps were obtained from the exported attention tensors and analyzed. These tensors are deterministic outputs of the learned projection matrices and downstream layers and are implicitly learned through backpropagation from the predictive and auxiliary losses. They were not regularized by any explicit attention supervision, and their training loss weights were set to zero to avoid imposing heuristic priors without ground truth rationales.

For quantitative attribution, Integrated Gradients (IG) was used to compute feature attributions for composition and load with respect to the hardness output. The baseline inputs were the mean training composition and mean training load, and attributions were computed using 200 integration steps.

All machine learning and data analysis workflows were implemented in Python v3.13 using TensorFlow v2.18.0, Keras v3.8.0, Optuna v4.2.1, scikit-learn v1.6.1, NumPy v2.2.2, pandas v2.2.3, and matplotlib v3.10.0.

## Experimental methods

The VIBANN-filtered compositions (see Table 1) were synthesized by arc melting high-purity elemental constituents ($\geq$ 99.9%, sourced from Sigma-Aldrich) under an inert argon atmosphere. Each alloy was re-melted and flipped at least five times to ensure chemical homogeneity. The molten alloy buttons were then suction cast into water-cooled copper molds to produce 2 mm diameter cylindrical rods. These rods were sectioned using a low-speed diamond saw and polished to a mirror finish for subsequent characterization.

**Structural characterization.** X-ray diffraction (XRD) patterns were collected using a D8 Advance A25-X1 diffractometer equipped with Cu-$K_\alpha$ radiation ($\lambda = 1.5406$ Å). Data were acquired over a 2θ range of 20° to 90° with a step size of 0.02°, enabling phase identification and verification of the amorphous structure. To further investigate the nanoscale phase constitution and structural motifs, transmission electron microscopy (TEM) was performed. Cross-sectional TEM lamellae were prepared using a focused ion beam (FIB) milling system on a FEI Helios Nanolab 600 dual-beam system, with final cleaning performed at low Ga ion voltages to minimize surface damage. A JEOL JEM-2200FS TEM operating at 200 kV was used for high-resolution imaging and selected area electron diffraction (SAED). The atomic-scale structure of the as-cast bulk MMG samples was investigated using high-energy synchrotron X-ray diffraction (HE-XRD). Measurements were performed at the P02.1 Powder Diffraction and Total Scattering Beamline at PETRA III, DESY (Germany). The beamline was operated at 60 keV ($\lambda = 0.207381$ Å), and data were collected with a Varex XRpad 4343CT fast area detector (2880 × 2880 pixels). The sample-to-detector distance was ~300 mm, and the incident beam size was confined to 1 × 1 mm². A quarter-section of the Debye-Scherrer rings (azimuthal angle 180°−270°) was recorded for each scan, with an exposure time of 15 s per sample. The 2D diffraction patterns were azimuthally integrated using GSAS-II software, yielding 1D diffraction intensity profiles over 2θ range of 1° to 18°. From these patterns, the structure factor S(Q) was extracted after background correction and normalization. Subsequently, radial distribution functions (RDFs), g(r), were computed via Fourier transformation of the corrected S(Q) profiles.

**Thermal analysis.** Differential scanning calorimetry (DSC) was performed using a Setaram SETSYS-18 instrument equipped with a platinum-rhodium type B thermocouple. Measurements were performed up to 1300 K under flowing high-purity argon. Cylindrical samples (2 mm diameter × 2 mm height) were placed in alumina crucibles, and heating/cooling rates were set to 20 K/min. The instrument was calibrated using Au, Ni, and Pd standards, yielding an accuracy of ±1 K for melting points. The glass transition temperatures ($T_g$) were determined with an accuracy of ±5 K, and crystallization onset temperatures ($T_x$) with ±8 K. Each alloy was subjected to two heating–cooling cycles to ensure thermal stability and reproducibility.

**Mechanical testing and pile-up correction.** Load-dependent Vickers microhardness testing was performed using a MicroCombi Tester (CSM Instruments) equipped with a Vickers diamond indenter. Indentations were made at loads ranging from 0.5 N to 10.0 N, with six repetitions per load to ensure statistical confidence. The diagonal lengths of each indent were measured optically, and the Vickers hardness was calculated using the standard formula. To quantitatively correct for pile-up artifacts in hardness estimation, atomic force microscopy (AFM) was conducted using an Oxford Asylum environmental AFM operating in tapping mode. Measurements were carried out under a dry nitrogen atmosphere (99.999% purity) in a sealed AFM chamber. A double-sided Ti-Ir-coated silicon tip (ASYELEC.01-R2, tip radius ~25 nm, resonant frequency ~75 kHz) was used. Scans were collected over a 30 μm × 30 μm area at a scan rate of 0.6 Hz. Line profiles along the indent diagonals were extracted to measure both pile-up height and indent depth. For prominent deformation features, four adjacent scans were stitched to create composite topographic maps. Corrected hardness was obtained from the AFM-derived projected contact area rather than from the optical diagonal alone. For each indent, pile-up was quantified along both indent diagonals, the actual projected contact area $A_{AFM}$ was reconstructed by including the outward protrusions associated with pile-up, and the corrected Vickers hardness was then calculated using

$$HV_{corr} = \frac{1.8544 \times F}{A_{AFM}} \quad (13)$$

Representative topography maps and depth profiles are provided in Supplementary Note 9.

Load-displacement curves were recorded in situ during indentation to provide insights into the elastic-plastic deformation behavior. SEM imaging in backscattered electron mode (Zeiss Sigma, operated at 15 kV and 7 nA) was used to examine indents, with a focus on pile-up, crack formation, and plastic zone development.

## Molecular dynamics simulations

To elucidate the atomic-level structural features that influence hardness, molecular dynamics (MD) simulations were performed using the SevenNet machine learning interatomic potential (version 11, July 2024) within the LAMMPS simulation package (release 4 Feb 2025)[60]. SevenNet is built on the Neural Equivariant Interatomic Potentials (NequIP) framework, which leverages rotationally and translationally equivariant neural networks to learn accurate, transferable force fields across diverse chemistries[61]. Its predictive fidelity has been demonstrated through top-tier performance on the Matbench Discovery benchmark for universal force fields[62]. Visualization and structural inspection were performed in OVITO (version 3.11.3, December 2024). For each MMG composition, a body-centered cubic (BCC) supercell comprising approximately 30,000 atoms (24 × 24 × 24 unit cells) was generated using an automated composition-to-atom mapping protocol. Each element was assigned based on atomic fractions, and atom types were appropriately indexed to interface with the SevenNet model.

The simulation protocol involved initial heating from 50 K to 4000 K, followed by equilibration at 4000 K, rapid quenching to 50 K, and a final low-temperature equilibration to stabilize the amorphous structure. All simulations were performed under the NPT ensemble using a 1 fs timestep and periodic boundary conditions. The final configurations of the MD models for all five alloys, together with representative LAMMPS input files and supporting metadata, have been deposited in Zenodo[58]. Following equilibration, a suite of structural descriptors was computed from the MD trajectories using in-house-developed Python scripts. Radial distribution functions (RDFs) were extracted for all pairwise elemental combinations, capturing both total and partial g(r) distributions that characterize short- and medium-range order. Coordination number histograms were derived to quantify the nearest-neighbor configurations for each atomic species, using a cutoff distance of 3 Å. Bond length and bond angle distributions were also calculated, allowing a quantitative assessment of packing compactness and angular disorder across compositions. Voronoi polyhedral analysis was performed to classify the local topological environments.

## Reporting summary

Further information on research design is available in the Nature Portfolio Reporting Summary linked to this article.

## Data availability

The curated literature-derived alloy dataset, processed composition-load-hardness tables used for model training and evaluation,

processed experimental datasets for the designed alloys, AFM topography data used for pile-up correction, X-ray diffraction data, differential scanning calorimetry data, molecular-dynamics final configurations, representative molecular-dynamics input files, and supporting metadata generated in this study have been deposited in Zenodo at https://doi.org/10.5281/zenodo.19548005. Source data are provided with this paper.

## Code availability

The custom Python code used for dataset curation, VIBANN model training, uncertainty quantification, latent-space analysis, inverse design, Gaussian-process Bayesian-optimization benchmarking, and post-processing has been deposited in Zenodo at https://doi.org/10.5281/zenodo.19548005. The repository includes the scripts required to reproduce the analyses reported in this study together with a package manifest and usage instructions.

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

## Acknowledgements

This work was primarily supported by the research fellowship provided to A.B. by the Alexander von Humboldt Foundation (hosted by D.R.). B.R. is also grateful for the financial support of the Alexander von Humboldt Foundation fellowship (hosted by D.R.). D.R. is grateful for financial support from the European Union through the ERC Advanced grant ROC (Grant Agreement No. 101054368). D. Klapproth is acknowledged for his assistance in alloy sample synthesis. B. Breitbach is acknowledged for the XRD measurements. HE-XRD measurements were carried out at beamline P02.1, PETRA III of Deutsches Elektronen-Synchrotron (DESY, proposal numbers I-20230183, I-20231121). F. Stein and L. Christiansen are acknowledged for the DSC measurements. H. Bögershausen and L. Eckhardt are acknowledged for the microindentation tests. The views expressed are solely those of the authors and do not necessarily reflect those of the European Union, the ERC, or the granting authority. Neither the European Union nor the granting authority can be held responsible for them.

## Author contributions

A.B. conceived the project, developed the machine-learning framework, acquired the data, trained the models, performed the analyses, carried out the experimental validation, generated the data visualizations and post-processing analyses, and wrote the initial draft of the manuscript. J.W. performed molecular dynamics simulations and reviewed the manuscript. B.R. contributed to experimental validation, visualization, and reviewed the manuscript. B.M.Ş. performed atomic force microscopy measurements. F.K. contributed to technical discussions. D.R. supervised the project, contributed to the conceptualization, and provided critical feedback throughout manuscript development. All authors discussed the results and contributed to the final manuscript.

## Funding

## Competing interests

The authors declare no competing interests.
