## [Transparent Peer Review file · Nature Communications]

Attention-enhanced variational learning for physically informed discovery of exceptionally hard multicomponent bulk metallic glasses

Corresponding Author: Professor Dierk Raabe

Version 0:

Reviewer comments:

Reviewer #1

(Remarks to the Author)

The authors, Bajpai et al., present a self-attention-based neural network with a variational information bottleneck to predict hardness in multicomponent metallic glasses. The model architecture is sophisticated and appears well-suited to the problem. While the training dataset is relatively small and limited in compositional diversity, the authors demonstrate that their approach outperforms simpler alternatives and achieves a reasonable bias-variance balance. The experimental characterization of five predicted alloys is thorough.

However, two central claims are unconvincing and must be addressed fully. The framework's physical interpretability is overstated, and the discovered alloys, while hard, do not meet the standard definition of "ultrahard" and are not without precedent, as the authors imply. The primary contribution of this work is the ML methodology itself, not breakthrough materials discovery or fundamental new insights into hardness mechanisms in amorphous alloys.

Specific Comments and Suggestions

1. Relationship to the prior work and a need for more systematic compositional exploration

All five predicted alloys (reported as novel discovery in this work) share a B-Nb-Fe base composition, with B-Nb as the dominant chemistry. Notably, Sarkar et al. (Ref. 44) reported $B_{72}Nb_{25}Fe_3$ thin film metallic glass with a hardness of 29 GPa (~2900 HV), which is 18% harder than the compositionally similar alloy A1 ($B_{68}Nb_{24}Fe_3W_4$, measured at 2446 HV or ~24.4 GPa). This raises important questions:

- Are the five alloys essentially compositional modifications of the Sarkar base ternary?
- While Sarkar synthesized thin films, it remains unclear whether processing constraints—rather than fundamental limitations—prevent bulk synthesis of the ternary composition. Have the authors attempted to synthesize the base B-Nb-Fe ternary composition as a bulk sample? Have they used their model to explore how the quaternary/quinary additions enhance or diminish properties relative to this baseline composition?

The ML model developed here could provide valuable insights and overcome challenges of experimental exploration, significantly accelerating it by systematically comparing how each additional element (W, Hf, Ru, Co) affects hardness computationally. Currently, the very limited data in the manuscript suggest that all quaternary/quinary additions reduce hardness relative to the ternary baseline—an observation that warrants discussion.

2. Limits of mechanistic interpretability

The authors identify three compositional clusters, noting that the hardest cluster is boride-dominated, and conclude that this demonstrates that the model has learned that "small atoms with strong covalent bonding tendencies" are critical for hardness. While this correlation is known, the interpretability claim requires more scrutiny:

- This relationship between small covalently bonded atoms and hardness has been well-established in materials science for decades, dating back to fundamental work on superhardness.

The model's recommendations appear constrained by alloys in the training data, raising a question about how well the

model learned the importance of small covalent atoms. Carbon forms even stronger covalent bonds than boron, and amorphous carbides (e.g., tetrahedral amorphous carbon, amorphous SiC) achieve a hardness of 30-80 GPa—far exceeding any alloy reported here. If the model truly learned fundamental principles about covalent bonding and hardness, rather than statistical patterns in metallic glass compositions in the training set, why does it not recommend carbide glasses? This doesn't invalidate the model, but it suggests the "physical interpretability" reflects learned correlations within the training distribution rather than transferable physical principles. The model appears to have learned "borides are hard among metallic glasses in the training set" rather than the deeper principle "strong covalent bonds yield hardness." The authors must be more explicit about such distinctions.

The interpretability claims could be strengthened by ablation studies showing how specific compositional features influence predictions. The addition of the 4 and 5 elements to the base BNbFe composition could be a good starting point for such an investigation. Demonstrate that the model can identify a hard non-boride metallic glass system would be another. Without such validation, the clustering analysis, while interesting, represents pattern recognition rather than physical insights.

3. Performance degradation at high hardness and the "ultrahard" designation

The authors repeatedly describe their alloys as "ultrahard," but this term has a specific meaning in the materials science community. Materials exceeding 40 GPa (4000 HV) are commonly called ultrahard. The measured hardness values (2300-2600 HV, or approximately 23-26 GPa) fall well short of this threshold. Indeed, these values are lower than several metallic glasses in the cited literature, including the 2900 HV B-Nb-Fe composition from Sarkar et al. discussed above. The term "ultrahard" should be replaced with "hard" or "high-hardness" throughout the manuscript to avoid confusion and align with community standards.

Model's limited prediction range: Figure 2a reveals a systematic and concerning trend that the model performs well below ~1800-2000 HV but progressively underestimates hardness as true values increase. This is further confirmed by all five synthesized alloys exceeding their predicted values. This pattern suggests:

- The model has learned relationships that govern hardness within the bulk of the training distribution (predominantly <2500 HV alloys) but may not capture mechanisms relevant to the highest-hardness regime.
- The model's increasing underestimation may suggest that achieving ultra-high hardness (>3000 HV) may involve qualitatively different structural or bonding mechanisms—such as specific short-range order, nanoscale phase separation, or bond coordination that are not well-represented in the training data, and the model has not learned them.
- The model may be performing well at interpolation within its training distribution but struggling with extrapolation to truly exceptional materials.
- Given the systematic underprediction at high hardness, my earlier assessment that the model "balances bias and variance well" may need qualification—it appears to show acceptable variance but increasing bias in the high-hardness regime.

In conclusion, this work makes a solid contribution to ML methodology for materials discovery, demonstrating that sophisticated attention-based architectures can outperform simpler approaches even with limited training data. However, the manuscript would be significantly strengthened by: (1) more cautious claims about physical interpretability, distinguishing pattern recognition from mechanistic insight; (2) correcting the "ultrahard" terminology to reflect actual measured values; (3) deeper analysis of the relationship between the discovered alloys and prior work; and (4) frank discussion of the model's apparent limitations in the highest-hardness regime.

(Remarks on code availability)

Reviewer #2

(Remarks to the Author)

The authors have made substantial improvements over the previous revision rounds. The proposed framework is clearly motivated, the overall argumentation is reasonable, and the manuscript is now close to publishable. However, given the architectural complexity of the method, the following four points should be addressed to better justify the added complexity and ensure full reproducibility.

1. The inverse-design framework integrates multiple advanced components, including a VIB-based latent representation, an attention-based predictor, uncertainty estimation via Monte Carlo dropout, GMM/MCMC-constrained sampling, and gradient-based refinement in the latent space. The authors should explicitly discuss what distinct advantages this pipeline offers, compared to a standard GP-based Bayesian optimization (GP-BO) baseline trained on the same dataset. In many materials design tasks, GP-BO can efficiently explore a local region near the best observed points; therefore, the manuscript should clarify why the proposed, more complex framework is necessary for the present problem.
2. The manuscript emphasizes that the learned latent representation forms a "navigable design landscape", enabling latent traversal and facilitating inverse design and gradient-based optimization. To support this claim more directly, the authors should provide a side-by-side latent-space visualization comparing the same model with VIB v.s. without VIB, so that the effect of the VIB regularization on latent structure can be evaluated more clearly.
3. While the encoder components are described reasonably well, key details of the decoder remain insufficiently specified. To ensure reproducibility, the authors should explicitly provide the decoder architecture and the complete mathematical formulation of the training loss.
4. The framework reports uncertainty estimates (via MC dropout) and calibration, which are valuable for assessing prediction reliability. However, it remains unclear whether and how this uncertainty is incorporated into the inverse-design decision-making.

(Remarks on code availability)

The code did not provide a REDAME file, so I did not try to install or run the code.

Version 1:

Reviewer comments:

Reviewer #1

(Remarks to the Author)

The authors have made substantial revisions that address all of my core concerns. I particularly appreciate their constructive response to my critique, which included not only additional computational analysis but also new experimental work—specifically, the synthesis and characterization of the nominal $B_{72}Nb_{25}Fe_3$ ternary baseline alloy to directly address the relationship between their designed compositions and prior literature. The clarifications regarding terminology (replacing "ultrahard" with "exceptionally hard"), the explicit domain-bounded framing of interpretability claims, and the transparent quantification of model performance across hardness regimes all strengthen the manuscript considerably.

I recommend publication.

I offer one minor suggestion that could enhance accessibility for a broader readership:

The authors provide an excellent and thorough analysis of the model's progressive underprediction at high hardness in the Supplementary Information. This analysis offers important mechanistic insights into both the model's behavior and the potentially distinct physical mechanisms governing ultra-high hardness in metallic glasses. However, since many readers do not carefully examine supplementary materials, a brief, explicit statement in the main text would make this important insight more visible.

Perhaps a short addition in the section on VIBANN performance or in the Discussion section, similar to something below?

"The observed systematic underprediction in the high-hardness regime (detailed in Supplementary Fig. S7) suggests that achieving hardness substantially above 2500 HV may involve mechanisms not fully captured by the current training distribution, where such extreme values are sparsely represented. This may reflect either qualitatively distinct deformation physics at the highest hardness levels or insufficient sampling of the relevant compositional space."

(Remarks on code availability)

Reviewer #2

(Remarks to the Author)

I have examined the authors' detailed responses to the reviewers' comments as well as the corresponding modifications to the manuscript. In my assessment, the authors properly addressed each concern, and revised the manuscript accordingly. The manuscript has been significantly improved. Therefore, I recommend acceptance of this manuscript for publication in Nature Communications.

(Remarks on code availability)

Detailed response to reviewers' comments

We would like to express our sincere gratitude to all the reviewers for their kind and repeated efforts in reviewing our manuscript in its revised form, and for their constructive suggestions and great support.

In the following, we provide a detailed point-by-point response report that explains all revision items. Relevant modifications in the manuscript and supplementary information have been highlighted in **YELLOW**.

Reviewer #1

Overall comment

The authors, Bajpai et al., present a self-attention-based neural network with a variational information bottleneck to predict hardness in multicomponent metallic glasses. The model architecture is sophisticated and appears well-suited to the problem. While the training dataset is relatively small and limited in compositional diversity, the authors demonstrate that their approach outperforms simpler alternatives and achieves a reasonable bias-variance balance. The experimental characterization of five predicted alloys is thorough.

However, two central claims are unconvincing and must be addressed fully. The framework's physical interpretability is overstated, and the discovered alloys, while hard, do not meet the standard definition of "ultrahard" and are not without precedent, as the authors imply. The primary contribution of this work is the ML methodology itself, not breakthrough materials discovery or fundamental new insights into hardness mechanisms in amorphous alloys.

Overall response

We sincerely thank the reviewer for the careful and balanced overall assessment of our work. We appreciate the recognition that the proposed VIBANN framework architecture is technically well matched to the problem and that the experimental validation of the five designed alloys is thorough. We also appreciate the reviewer's clear identification of the points that required the most careful reconsideration.

In the revised version, we have therefore sharpened the scope of our claims. Specifically, we now distinguish more clearly between physically plausible, data-grounded interpretability within the bulk metallic glass domain and universal mechanistic laws applicable across all amorphous materials. We also clarify that the present study focuses on the development of an uncertainty-aware, experimentally validated inverse-design framework for high-hardness multicomponent bulk metallic glasses, rather than on the discovery of previously unknown ultrahard amorphous materials.

At the same time, we respectfully submit that the contribution is broader than methodology in

isolation. The central advance of the work is the combination of three elements within a single closed-loop framework: accurate hardness prediction under data sparsity, chemically constrained inverse design in a high-dimensional space of metallic-glass compositions, and experimental realization and atomistic analysis of the resulting alloys. In that sense, the manuscript contributes both a machine-learning framework and a validated route for discovering exceptionally hard bulk MMGs within a practically relevant metallurgical domain.

We now address the detailed comments point-by-point below:

Comment 1

Relationship to the prior work and a need for more systematic compositional exploration: All five predicted alloys (reported as novel discovery in this work) share a B-Nb-Fe base composition, with B-Nb as the dominant chemistry. Notably, Sarker et al. (Ref. 44) reported $B_{72}Nb_{25}Fe_3$ thin film metallic glass with a hardness of 29 GPa (~ 2900 HV), which is 18% harder than the compositionally similar alloy A1 ($B_{68}Nb_{24}Fe_3W_4$, measured at 2446 HV or ~ 24.4 GPa). This raises important questions:

- Are the five alloys essentially compositional modifications of the Sarker base ternary?

Response

We thank the reviewer for this important and pertinent observation. We agree that the five designed alloys are not chemically unrelated to the prior literature related to Fe-Nb-B alloys, and we also concur that this relationship should be stated more explicitly in the manuscript. In particular, all five experimentally validated candidates lie within a broader B-Nb-Fe-centered high-hardness compositional basin, and, in that sense, they are connected to prior work on hard Fe-Nb-B metallic glasses, including, for instance, the thin-film study by Sarker et al., which reported $Fe_3Nb_{25}B_{72}$ with a nanoindentation hardness of about 29 GPa (at 0.01 N load).

At the same time, we respectfully submit that the five alloys should not all be viewed as simple restatements of that specific ternary composition. Among our candidates, alloy A1, $B_{68}Nb_{24}Fe_3W_4$, is compositionally the closest alloy to the $Fe_3Nb_{25}B_{72}$ thin-film composition and it can reasonably be regarded as a quaternary extension of the same hard boride-rich Fe-Nb-B MG. However, the remaining candidates represent progressively larger multicomponent redesigns within that broader high-hardness basin rather than small perturbations of a single ternary point. In particular, A2, $B_{62}Nb_{12}Fe_4Hf_8Ru_6W_8$, and A5, $B_{61}Nb_{18}Fe_3Co_5W_8Zr_5$, involve substantial redistribution of the base chemistry together with multiple additional alloying elements (which are otherwise sparsely present in the dataset), while A3 and A4 likewise move beyond a one-element modification of the ternary composition. The final experimentally measured bulk compositions of our new alloys, reported in Table S6 (Supplementary Information), confirm this broader multicomponent character.

Importantly, we emphasize that the present inverse-design framework was not formulated to unconditionally maximize chemical dissimilarity from previously reported alloys. Rather, it was designed to identify candidates that jointly satisfy high predicted hardness, latent-space plausibility, low epistemic uncertainty, and decoder-supported compositional feasibility within the bulk metallic-glass training domain. Accordingly, the search procedure explicitly favors data-supported, uncertainty-robust candidates and discourages unconstrained extrapolation into sparsely supported regions of composition space. Under these conditions, convergence toward a B-Nb-Fe-rich high-hardness basin should be interpreted as evidence that the VIBANN framework identified a chemically credible and performance-favorable region and then refined that region into experimentally realizable bulk multicomponent compositions. This interpretation is consistent with our latent-space optimization results, where the final candidates remain in regions of high latent support and constitute controlled departures from their nearest known alloys rather than arbitrary excursions into unsupported chemistry.

We have revised the manuscript accordingly to make this relationship to prior Fe-Nb-B work explicit and to avoid overstating the degree of chemical novelty.

- While Sarker synthesized thin films, it remains unclear whether processing constraints, rather than fundamental limitations, prevent bulk synthesis of the ternary composition. Have the authors attempted to synthesize the base B-Nb-Fe ternary composition as a bulk sample? Have they used their model to explore how the quaternary/quinary additions enhance or diminish properties relative to this baseline composition?

Response

We thank the reviewer for this important and very constructive question. We agree that, to distinguish between processing limitations and a more intrinsic limitation to bulk glass formation, it is necessary to examine the ternary B-Nb-Fe baseline experimentally rather than infer its behavior indirectly from the thin-film literature.

In response, to examine whether the B-Nb-Fe ternary baseline alloy can be realized in bulk under the same processing conditions used for our inverse-designed alloys, we synthesized the nominal ternary composition $B_{72}Nb_{25}Fe_3$ by arc melting followed by suction casting. Under these bulk-processing conditions, the ternary alloy did not vitrify. Instead, X-ray diffraction of the cast rod showed sharp crystalline reflections, which could be indexed predominantly to NbB_2 , with no broad amorphous halo characteristic of a BMG (Figure S31). We have added this result as a new Supplementary Figure and now discuss it explicitly in the manuscript. This comparison shows that, under the present bulk-casting route, the ternary baseline is crystallization-prone, whereas the inverse-designed multicomponent alloys could be realized as amorphous bulk rods.

Figure S31 – X-ray diffraction pattern of the bulk-cast ternary reference alloy $B_{72}Nb_{25}Fe_3$ prepared using the same arc-melting and suction-casting route as that used for the inverse-designed alloys.

To address the reviewer’s comment on the role of the quaternary and quinary additions relative to this baseline composition, we evaluated the trained VIBANN model on the nominal ternary $B_{72}Nb_{25}Fe_3$ composition and compared its predicted hardness and uncertainty with those of the five experimentally validated inverse-designed alloys at representative indentation loads of 0.5 and 5 N. In this bulk-load regime, the nominal ternary is predicted to be lower in hardness than all five selected candidates at both loads. At 0.5 N, the ternary baseline is predicted at 1835.4 HV, whereas A1-A5 span 1964.9-2342.1 HV. At 5 N, the ternary baseline is predicted at 1571.2 HV, whereas A1-A5 span 1679.5-2071.7 HV. The corresponding lower-confidence-bound values follow the same ordering, indicating that this ranking is retained under uncertainty-aware evaluation (Table S8).

Table S8 – Model-predicted hardness metrics for the nominal ternary baseline and the experimentally validated inverse-designed alloys at representative loads.

Alloy	Composition	Predicted hardness at 0.5 N	Predicted uncertainty at 0.5 N	LCB at 0.5 N	Predicted hardness at 5 N	Predicted uncertainty at 5 N	LCB at 5 N
Baseline ternary	$B_{72}Nb_{25}Fe_3$	1835.4	109.2	1620.8	1571.2	88.6	1397.1
A1	$B_{68}Nb_{24}Fe_4W_4$	2342.1	124.7	2097.6	2071.7	81.3	1912.3
A2	$B_{62}Nb_{12}Fe_4Hf_5Ru_6W_8$	2246.8	114	2023.3	1921.6	72.7	1779.1
A3	$B_{64}Nb_{23}Fe_5Co_8$	2185.2	106.5	1976.4	1884.7	71.3	1744.9
A4	$B_{66}Nb_{21}Fe_4Hf_4Ru_5$	2070.8	98.3	1878.1	1791.2	70.8	1652.4
A5	$B_{61}Nb_{18}Fe_3Co_5W_8Zr_{15}$	1964.9	96.3	1776.1	1679.5	63.8	1554.4

To further test whether this behavior reflects a systematic compositional effect rather than an isolated comparison, we also traced simple stepwise compositional paths from the ternary baseline toward representative designed alloys. Along the path from $B_{72}Nb_{25}Fe_3$ to A1, $B_{68}Nb_{24}Fe_4W_4$, the model predicts a monotonic increase in hardness from 1835.4 HV to 2342.1 HV at 0.5 N (Figure S30a). Likewise, along the same ternary baseline toward A4, $B_{66}Nb_{21}Fe_4Hf_4Ru_5$, the predicted hardness increases progressively from 1835.4 HV to 2070.8 HV (Figure S30b). These trends indicate that the added alloying elements do not act as arbitrary perturbations around the ternary, but instead move the composition systematically toward a higher-hardness region within the bulk-MMG design space learned by the model.

Figure S30 – Stepwise model-based compositional evolution from the nominal $B_{72}Nb_{25}Fe_3$ ternary baseline toward representative inverse-designed alloys, evaluated at 0.5 N. (a) Path toward A1, $B_{68}Nb_{24}Fe_4W_4$, through progressive W addition and accompanying adjustment of B, Nb, and Fe. (b) Path toward A4, $B_{66}Nb_{21}Fe_4Hf_4Ru_5$, through progressive Hf/Ru addition and rebalancing of the base ternary chemistry. Symbols denote predictive mean hardness and error bars indicate predictive uncertainty. In both cases, the model predicts a monotonic increase in hardness along the compositional path.

At the same time, we emphasize that the previously reported hardness of the related Fe-Nb-B thin-film metallic glass reported by Sarker et al.¹ should be interpreted with caution relative to the present results. Their reported value was obtained on thin-film samples through nanoindentation at a much lower indentation load (0.01 N) than that used in the present work, whereas our model evaluation and experiments are anchored in the bulk-load regime, with 0.5 N as the minimum load considered. Because MGs exhibit pronounced load dependence in measured hardness, and because thin-film and bulk geometries are not directly equivalent, a one-to-one numerical comparison is not appropriate. We therefore interpret the present model results as providing the relevant ranking within the BMG design space addressed by this study, rather than as a direct contradiction of the thin-film nanoindentation result.

We have revised the manuscript and Supplementary Information accordingly to make this

distinction explicit and to present the ternary bulk-synthesis comparison and the corresponding model-based baseline analysis.

- The ML model developed here could provide valuable insights and overcome challenges of experimental exploration, significantly accelerating it by systematically comparing how each additional element (W, Hf, Ru, Co) affects hardness computationally. Currently, the very limited data in the manuscript suggest that all quaternary/quinary additions reduce hardness relative to the ternary baseline—an observation that warrants discussion.

Response

We thank the reviewer for this constructive suggestion. We agree that one of the major strengths of a machine-learning-guided framework is its ability to perform targeted computational comparisons that would be costly and time-consuming to establish solely through experimentation.

To address this point, we have now carried out a systematic model-based comparison using the trained VIBANN surrogate. Specifically, we evaluated the nominal ternary baseline composition $B_{72}Nb_{25}Fe_3$ and compared its predicted hardness and uncertainty with those of the five experimentally validated inverse-designed alloys at representative bulk indentation loads of 0.5 and 5 N (as shown in the previous comment's response). The model does not predict that the quaternary/quinary additions reduce hardness relative to the ternary baseline in the bulk-load regime relevant to the present work. Instead, the nominal ternary is predicted to be softer than all five selected multicomponent alloys at both loads. At 0.5 N, the baseline ternary is predicted at 1835.4 HV, whereas A1-A5 span 1964.9 to 2342.1 HV. At 5 N, the ternary is predicted at 1571.2 HV, whereas A1-A5 span 1679.5 to 2071.7 HV (Table S8). The corresponding lower-confidence-bound values follow the same ordering, indicating that the trend is retained under uncertainty-aware evaluation.

To further examine whether this behavior reflects a systematic compositional effect rather than isolated endpoints, we additionally traced simple stepwise paths from the nominal ternary baseline toward two representative designed alloys. Along the path from $B_{72}Nb_{25}Fe_3$ to A1, $B_{68}Nb_{24}Fe_4W_4$, the model predicts a monotonic increase in hardness from 1835.4 HV to 2342.1 HV at 0.5 N (Figure S30a). Likewise, along the same ternary baseline toward A4, $B_{66}Nb_{21}Fe_4Hf_4Ru_5$, the predicted hardness increases progressively from 1835.4 HV to 2070.8 HV (Figure S30b). These trends indicate that the additional alloying elements do not act merely as decorative modifications of the ternary base composition, but systematically shift the composition toward a higher-hardness region within the learned bulk-MMG design space.

At the same time, we have taken care not to overstate the scope of this analysis. The present results do not imply that every addition of W, Hf, Ru, or Co will universally increase hardness in all B-Nb-Fe-based MGs, nor that the model establishes a complete causal ranking of individual

alloying effects independent of composition and load. Rather, they show that, within the trained BMG composition space and within the load regime used for inverse design, the specific multicomponent pathways explored here lead to higher predicted hardness than the nominal ternary baseline. This is the relevant comparison for the present study.

We have therefore revised the manuscript and Supplementary Information to explicitly include this new baseline analysis, including a direct ternary vs. A1-A5 comparison at 0.5 and 5 N and stepwise compositional paths toward representative designed alloys. These additions make the computational rationale for the selected multicomponent compositions much clearer and directly address the reviewer’s suggestion.

Main text changes: In Results on pg. 12,

“All five inverse-designed candidates (A1-A5) lie within a broader B-Nb-Fe-centered high-hardness compositional region. This places them in chemical continuity with earlier reported hard Fe-Nb-B thin-film MGs,¹ rather than in a wholly unrelated chemistry class. Among the present candidates, A1 is compositionally closest to the boride-rich $\text{Fe}_3\text{Nb}_{25}\text{B}_{72}$ thin-film composition reported previously, whereas A2-A5 represent progressively larger multicomponent extensions of this broader hard-BMG basin. This convergence is consistent with the objective of the present inverse-design framework, which explicitly favors candidates that jointly satisfy high predicted hardness, latent-space plausibility, and low epistemic uncertainty within the training-supported bulk metallic-glass domain. The resulting alloys should therefore be interpreted as uncertainty-screened bulk multicomponent refinements of a chemically credible high-hardness region, rather than as unconstrained extrapolations into a disconnected alloy family. This interpretation is further examined through direct bulk synthesis of the nominal $\text{B}_{72}\text{Nb}_{25}\text{Fe}_3$ ternary reference alloy and through model-based baseline comparisons at representative loads (Supplementary Section S10, Figures S29-31, and Table S8).”

On pg. 17,

“For comparison with the previously reported Fe-Nb-B thin-film literature,¹ we also examined the nominal $\text{B}_{72}\text{Nb}_{25}\text{Fe}_3$ ternary baseline alloy under the same bulk processing route used for our inverse-designed alloys. When prepared by arc melting followed by suction casting, this ternary composition did not vitrify as a bulk rod. Its XRD pattern shows sharp crystalline reflections, predominantly indexed to NbB_2 , with no broad amorphous halo, indicating crystallization during casting (Figure S31). We further evaluated this baseline composition using the trained VIBANN model at representative loads of 0.5 and 5 N, respectively, and compared it with the experimentally validated candidates (Supplementary Section S10, Table S8, Figure S29). In this bulk-load regime, the nominal ternary baseline alloy is predicted to have lower hardness than all five designed bulk MMGs. In addition, stepwise compositional paths from the ternary alloy toward representative designed bulk MMGs show monotonic increases in model-predicted hardness with progressive W or Hf/Ru addition (Figure S30). These observations indicate that

the role of the added alloying elements is not only to modify hardness, but also to shift the system from a crystallization-prone ternary alloy toward a bulk-realizable high-hardness amorphous region.”

Changes in Supplementary Information:

“S10. Model-based comparison with the nominal B-Nb-Fe ternary baseline

To further examine the role of multicomponent alloying relative to the nominal $B_{72}Nb_{25}Fe_3$ ternary reference from Sarker et al.,¹ we evaluated the trained VIBANN model on this baseline composition and compared its predicted hardness and uncertainty with those of the five experimentally validated inverse-designed alloys at representative bulk indentation loads of 0.5 and 5 N, respectively. The corresponding results are summarized in Table S8 and Figure S29. In the bulk-load regime studied here, the nominal ternary baseline is predicted to have lower hardness than all five selected candidates at both loads. At 0.5 N, the ternary is predicted at 1835.4 HV, whereas A1-A5 BMGs span 1964.9–2342.1 HV. At 5 N, the ternary is predicted at 1571.2 HV, whereas A1-A5 BMGs span 1679.5–2071.7 HV. The corresponding lower-confidence-bound values follow the same ranking, indicating that the improvement is retained under uncertainty-aware evaluation.

To assess whether these gains arise systematically rather than from isolated composition choices, we further traced compositional paths from the nominal ternary baseline toward two representative designed alloys, A1 and A4, while evaluating the model at 0.5 N. As shown in Figure S30, gradual alloying from $B_{72}Nb_{25}Fe_3$ to $B_{68}Nb_{24}Fe_4W_4$ results in a monotonic increase in predicted hardness from 1835.4 HV to 2342.1 HV. Likewise, the path from the ternary baseline toward $B_{66}Nb_{21}Fe_4Hf_4Ru_5$ shows a progressive increase from 1835.4 HV to 2070.8 HV. These results indicate that, within the learned bulk-MMG design space, the quaternary and quinary additions do not act as arbitrary perturbations but systematically move the composition toward higher predicted hardness.

These model-based results should be interpreted together with the experimental bulk-synthesis comparison. The nominal $B_{72}Nb_{25}Fe_3$ ternary alloy did not vitrify under the present bulk-casting conditions and instead crystallized (Figure S31), whereas the inverse-designed multicomponent alloys formed amorphous bulk rods. Accordingly, the role of the added alloying elements in the present work is not only to tune hardness, but also to shift the system from a crystallization-prone ternary motif toward a bulk-realizable high-hardness compositional region. We also note that previously reported hardness values for related Fe-Nb-B thin-film metallic glasses were obtained at substantially lower indentation loads than those considered here (0.01 N), so caution is warranted when making a direct numerical comparison.

Table S8 – Model-predicted hardness metrics for the nominal ternary baseline and the experimentally validated inverse-designed alloys at representative loads.

Alloy	Composition	Predicted hardness at 0.5 N	Predicted uncertainty at 0.5 N	LCB at 0.5 N	Predicted hardness at 5 N	Predicted uncertainty at 5 N	LCB at 5 N
Baseline ternary	$B_{72}Nb_{25}Fe_3$	1835.4	109.2	1620.8	1571.2	88.6	1397.1
A1	$B_{68}Nb_{24}Fe_4W_4$	2342.1	124.7	2097.6	2071.7	81.3	1912.3
A2	$B_{62}Nb_{12}Fe_4Hf_8Ru_6W_8$	2246.8	114	2023.3	1921.6	72.7	1779.1
A3	$B_{64}Nb_{23}Fe_5Co_8$	2185.2	106.5	1976.4	1884.7	71.3	1744.9
A4	$B_{66}Nb_{21}Fe_4Hf_4Ru_5$	2070.8	98.3	1878.1	1791.2	70.8	1652.4
A5	$B_{61}Nb_{18}Fe_3Co_5W_8Zr_5$	1964.9	96.3	1776.1	1679.5	63.8	1554.4

Figure S29 – Model-predicted hardness of the nominal $B_{72}Nb_{25}Fe_3$ ternary baseline and the five experimentally validated inverse-designed alloys at indentation loads of 0.5 and 5 N. Symbols denote the predictive mean from the trained VIBANN model, and error bars indicate predictive uncertainty estimated from Monte Carlo dropout. In the bulk-load regime used for inverse design, all five designed multicomponent alloys are predicted to outperform the ternary baseline.

Figure S30 – Stepwise model-based compositional evolution from the nominal $B_{72}Nb_{25}Fe_3$ ternary baseline toward representative inverse-designed alloys, evaluated at 0.5 N. (a) Path toward A1, $B_{68}Nb_{24}Fe_4W_4$, through progressive W addition and accompanying adjustment of B, Nb, and Fe. (b) Path toward A4, $B_{66}Nb_{21}Fe_4Hf_4Ru_5$, through progressive Hf/Ru addition and rebalancing of the base ternary chemistry. Symbols denote predictive mean hardness and error bars indicate predictive uncertainty. In both cases, the model predicts a monotonic increase in hardness along the compositional path.

Figure S31 – X-ray diffraction pattern of the bulk-cast ternary reference alloy $B_{72}Nb_{25}Fe_3$ prepared using the same arc-melting and suction-casting route as that used for the inverse-designed alloys.”

Comment 2

Limits of mechanistic interpretability: The authors identify three compositional clusters, noting that the hardest cluster is boride-dominated, and conclude that this demonstrates that the model

has learned that "small atoms with strong covalent bonding tendencies" are critical for hardness. While this correlation is well known, the interpretability claim warrants closer scrutiny. This relationship between small covalently bonded atoms and hardness has been well-established in materials science for decades, dating back to fundamental work on superhardness. The model's recommendations appear constrained by the alloys in the training data, raising questions about how well it learned the importance of small covalent atoms. Carbon forms even stronger covalent bonds than boron, and amorphous carbides (e.g., tetrahedral amorphous carbon, amorphous SiC) achieve a hardness of 30-80 GPa—far exceeding any alloy reported here. If the model truly learned fundamental principles about covalent bonding and hardness, rather than statistical patterns in metallic glass compositions in the training set, why does it not recommend carbide glasses? This doesn't invalidate the model, but it suggests the "physical interpretability" reflects learned correlations within the training distribution rather than transferable physical principles. The model appears to have learned "borides are hard among metallic glasses in the training set" rather than the deeper principle "strong covalent bonds yield hardness." The authors must be more explicit about such distinctions. The interpretability claims could be strengthened by ablation studies showing how specific compositional features influence predictions. The addition of the 4- and 5-element components to the base B-Nb-Fe composition could be a good starting point for such an investigation. Demonstrate that the model can identify a hard non-boride metallic glass system would be another. Without such validation, the clustering analysis, while interesting, represents pattern recognition rather than physical insights.

Response

We thank the reviewer for this important comment. We agree that the distinction between domain-bounded interpretability and universally transferable physical principles must be stated much more explicitly. The reviewer is correct that the present model should not be interpreted as having discovered a general hardness law applicable across all amorphous materials. Rather, the VIBANN framework is trained on BMG compositions and indentation conditions reported for metallic alloys, and its attention, attribution, and latent-space patterns therefore represent structure-property correlations learned within the restricted chemical and processing domain of the training dataset.

In this context, the model's emphasis on boron-rich compositions should not be read as the universal principle that it has learned "strong covalent bonding yields hardness". The more accurate interpretation is narrower: within the bulk MMG design space represented by the training data, boron is the most prevalent small metalloid that simultaneously increases bond directionality, enhances local short-range rigidity, and remains compatible with metallic glass formation in multicomponent alloys. The model, therefore, identifies boron-rich, refractory-stabilized metallic-glass compositions as high-hardness directions within the BMG manifold, rather than inferring that all stronger covalent amorphous solids should be recommended.

We also agree that the carbon-based counterexample should be discussed more carefully. We do

not argue that carbon-containing metallic glasses cannot be hard. Carbon-bearing metallic-glass systems are well established in the literature, including early Fe-P-C alloys and later Fe-based C-containing BMG families, and some of these exhibit high hardness and good glass-forming ability.² However, in MGs, carbon exists within a metal-rich amorphous matrix, subject to coupled constraints of bulk glass formation, local packing, short-range chemical order, and competing crystallization.³ As a result, the relevant question for the present VIBANN model is not whether carbon forms strong bonds in general, but whether carbon-containing compositions occupy a high-hardness, glass-formable region within the specific BMG manifold represented in the training data. Recent work further shows that the combined B/C balance can significantly affect glass-forming ability and hardness in glass-forming alloys,⁴ underscoring that carbon does not operate as a universally dominant hardness variable in BMG design. For example, Fe-Cr-Mo-C-B-Nb BMGs have a Vickers hardness of around 1360 HV,⁵ which is high but still well below that of the boron-rich Nb-based MGs studied here.

We therefore agree with the reviewer that the interpretability claim must be narrowed. The claim is not that VIBANN has learned a transferable physical law for all amorphous materials. The claim is that, within bulk MG chemistry, it identifies physically plausible alloying directions that correlate with higher hardness and that are consistent with both experimental validation and atomistic structural indicators of increased short-range rigidity. In the revised manuscript, we have made this boundary explicit and have avoided language suggesting universal mechanistic transferability.

To further strengthen this point within the boride-centered region explored experimentally, we have also added a targeted compositional analysis around the $B_{72}Nb_{25}Fe_3$ baseline. Using the trained VIBANN model, we compared the nominal ternary with the experimentally validated quaternary/quinary candidates and traced representative stepwise compositional paths toward A1 and A4 (as shown in previous comments, Figure S30). These analyses show that, within the trained BMG manifold and in the bulk-load regime relevant to the study, the added alloying elements systematically shift the compositions toward higher predicted hardness. We present these results as targeted evidence that the model captures chemically meaningful strengthening directions within the specific BMG space explored here, not as proof of a universal mechanistic law.

Figure S30 – Stepwise model-based compositional evolution from the nominal $B_{72}Nb_{25}Fe_3$ ternary baseline toward representative inverse-designed alloys, evaluated at 0.5 N. (a) Path toward A1, $B_{68}Nb_{24}Fe_4W_4$, through progressive W addition and accompanying adjustment of B, Nb, and Fe. (b) Path toward A4, $B_{66}Nb_{21}Fe_4Hf_4Ru_5$, through progressive Hf/Ru addition and rebalancing of the base ternary chemistry. Symbols denote predictive mean hardness and error bars indicate predictive uncertainty. In both cases, the model predicts a monotonic increase in hardness along the compositional path.

In addition, to address the reviewer’s suggestion that the model should be tested beyond the boride-dominated basin, we performed two new non-boride stress tests. First, we conducted a leave-one-family-out validation on a hard, non-boride metallic glass family. Among the eligible non-boride families in the dataset, the Mo–Ta–Si–W family was selected based on its comparatively high median hardness and sufficient representation. The entire family was then excluded from training, the model was retrained, and predictions were evaluated only on this held-out chemistry space. As shown in Figure S38, the held-out family remained well-predicted, with $n=19$, $R^2=0.89$, $RMSE = 32.59$ HV, and $MAE = 24.46$ HV. This result shows that the learned latent structure is not limited to memorizing a single boride-enriched family and retains the ability to recognize a comparatively hard non-boride metallic-glass region even when that chemistry family is not seen during training.

Figure S38 – Leave-family-out validation on Mo,Ta,Si,W non-boride metallic-glass family. (a) Eligible non-boride composition families ranked by median hardness, with the selected held-out family highlighted. The family Mo–Ta–Si–W, containing 19 samples, was chosen for the leave-family-out test based on its comparatively high hardness within the non-boride subset and sufficient representation for meaningful evaluation. (b) True versus predicted hardness for the held-out Mo–Ta–Si–W family after retraining VIBANN with this entire family excluded from the training set. Error bars denote predictive uncertainty, and the dashed diagonal indicates perfect agreement. (c) Absolute prediction error as a function of predictive uncertainty for the held-out family.

Second, we imposed an explicit boron-free constraint during inverse design and examined whether the same uncertainty-aware framework could still identify plausible high-hardness candidates. The resulting top boron-free MMG candidates are summarized in Table S9. These candidates span compositions such as Mo–W–Si–Ta–Zr–Co–Y, Mo–Si–Ta–Hf–Fe–Y, and W–Zr–Si–Ta–Ti, with predicted hardness values up to 1876.4 HV and lower confidence bounds up to 1722.8 HV, while retaining moderate novelty and reasonable latent-space log-likelihood. Although these boron-free candidates were not experimentally synthesized in the present work, they demonstrate that the learned design manifold is not confined to a single boride-rich shortcut and can support the identification of chemically plausible non-boride high-hardness directions under the same inverse-design framework.

Table S9 – Top boron-free inverse-designed bulk MMG candidates. Predicted hardness (μ), predictive uncertainty (σ), and lower confidence bound ($\text{LCB} = \mu - 1.96\sigma$) are reported together with composition-space novelty (L2 distance from the nearest training composition) and latent-space log-likelihood as a measure of plausibility within the learned design manifold. Candidates are ranked to highlight non-boride, chemically plausible high-hardness directions identified by VIBANN.

Alloy	Predicted hardness (HV)	Uncertainty (HV)	LCB (HV)	Novelty (L2)	Latent space log-likelihood
Mo ₃₈ W ₈ Si ₂₄ Ta ₈ Zr ₅ Co ₁₂ Y ₅	1876.4	78.34	1722.8	0.23	-7.6
Mo ₃₆ Si ₂₄ Ta ₁₀ Hf ₈ Fe ₁₇ Y ₅	1864.0	84.92	1697.5	0.39	-7.3
W ₅₂ Zr ₁₈ Si ₁₈ Ta ₆ Ti ₆	1855.1	75.67	1706.8	0.28	-8.2
W ₄₃ Zr ₁₄ Hf ₁₃ Si ₂₂ Ta ₈	1847.6	81.48	1687.9	0.19	-7.8
Ta ₅₂ W ₁₆ Zr ₁₂ Si ₁₀ Ti ₁₀	1834.2	86.11	1665.4	0.35	-8.2
Mo ₃₂ Ta ₁₄ Si ₂₆ Zr ₂₀ Hf ₁₄	1826.8	79.03	1671.9	0.32	-8.0
W ₄₆ Ta ₁₂ Zr ₁₈ Hf ₈ Si ₁₆	1811.2	82.76	1648.6	0.26	-8.2
W ₄₂ Ta ₁₀ Zr ₁₆ Ti ₁₀ Si ₁₄ Cs	1802.6	74.58	1656.4	0.37	-8.0
Ta ₄₈ Ni ₁₈ Si ₁₄ C ₁₀ Hf ₁₀	1784.9	85.37	1617.5	0.24	-8.2
Ta ₅₅ Ti ₁₀ Zr ₁₄ Hf ₈ Si ₁₂	1761.4	80.25	1604.1	0.30	-7.6

Taken together, these additions substantially narrow and clarify our interpretability claim. The clustering, attribution, and latent-space analyses should be interpreted as evidence of domain-bounded, chemically meaningful correlations within the metallic-glass manifold represented in the data. They do not establish a universal covalent-bonding hardness law across all amorphous materials. Rather, they show that VIBANN identifies hardness-promoting compositional directions within bulk MG chemistry, including beyond the immediate boride-centered validation basin.

Main text changes: In Results on pgs. 12-13,

“To test whether the learned structure was confined to this boride-centered region, we performed a leave-family-out validation on the Mo–Ta–Si–W family, selected from the eligible non-boride subset based on its comparatively high median hardness and sufficient representation. When this entire family was excluded from training, the model retained good accuracy ($n=19$, $R^2=0.89$, $RMSE=32.59$ HV, $MAE=24.46$ HV; Supplementary Fig. S38). Thereafter, under an explicit boron-free inverse-design constraint, VIBANN identified chemically plausible non-boride candidates with predicted hardness up to 1876.4 HV and lower confidence bounds up to 1722.8 HV (Supplementary Table S9). These analyses indicate that the learned latent structure is not reducible to a single boride-rich region, although its interpretive scope remains restricted to the bulk MMG manifold represented in the data.”

On pg. 17,

“In addition, stepwise compositional paths from the ternary alloy toward representative designed bulk MMGs show monotonic increases in model-predicted hardness with progressive W or Hf/Ru addition (Figure S30). These observations indicate that the role of the added alloying elements is not only to modify hardness, but also to shift the system from a crystallization-prone ternary alloy toward a bulk-realizable high-hardness amorphous region.”

In Discussion on pgs. 27-28,

“It is important to note that the attention, attribution, and latent-space analyses should be interpreted as structure-property correlations learned within the restricted chemical and processing space defined by the present dataset, rather than as universally transferable laws for all amorphous materials. In this context, the model preference for boron-rich compositions does not imply that it has learned the general statement that stronger covalent bonding always yields higher hardness. The more precise interpretation is that, within the bulk MMG space sampled here, boron is the most prevalent small metalloid associated with increased bond directionality, enhanced local short-range rigidity, and continued compatibility with bulk MG formation. The model, therefore, identifies boron-rich and refractory-stabilized compositions as the dominant high-hardness direction within the present bulk MMG manifold, rather than inferring that all

amorphous systems with stronger covalent bonding should be selected.

The carbon-based comparison reinforces this point. Carbon-bearing MG systems are known to exhibit high hardness,² but in MGs, carbon acts within a metal-rich amorphous matrix under the coupled constraints of bulk glass formation, local packing, short-range chemical order, and competing crystallization.^{3,4} The relevant question for the present framework is therefore not whether carbon forms strong bonds in general, but whether carbon-containing compositions occupy a high-hardness and glass-formable region within the specific bulk MG manifold represented in the dataset. Within that design space, boron remains the more consistent high-hardness direction identified by the model, indicating that VIBANN is learning domain-bounded alloying correlations rather than a universally transferable relation between covalent bonding and hardness.

These considerations define the principal interpretive boundary of the framework. The attention and integrated-gradient patterns are most reliable for chemically adjacent exploration within the bulk MMG manifold represented by the dataset. They should not be interpreted as predictions for chemically distinct amorphous classes outside that space. Within the bulk MMG domain, however, the combined evidence from attention and attribution, uncertainty-penalized candidate selection, experimental validation, and MD-based structural analysis supports a coherent picture. The highest-hardness compositions identified here correspond to densely packed amorphous networks in which boron-enriched short-range environments are reinforced by refractory and transition-metal additions that increase local stiffness and resistance to shear transformation activation. Consistent with this domain-bounded interpretation, the additional leave-family-out validation on a hard non-boride family and the boron-free constrained inverse-design results show that the learned hardness trends are not confined to a single boride-enriched basin, but extend to chemically distinct non-boride regions within the dataset-supported bulk-MMG manifold (Supplementary Fig. S38, Supplementary Table S9).”

Changes in Supplementary Information: On pgs. 51, 53;

“To assess whether these gains arise systematically rather than from isolated composition choices, we further traced compositional paths from the nominal ternary baseline toward two representative designed alloys, A1 and A4, while evaluating the model at 0.5 N. As shown in Figure S30, gradual alloying from $B_{72}Nb_{25}Fe_3$ to $B_{68}Nb_{24}Fe_4W_4$ results in a monotonic increase in predicted hardness from 1835.4 HV to 2342.1 HV. Likewise, the path from the ternary baseline toward $B_{66}Nb_{21}Fe_4Hf_4Ru_5$ shows a progressive increase from 1835.4 HV to 2070.8 HV. These results indicate that, within the learned bulk-MMG design space, the quaternary and quinary additions do not act as arbitrary perturbations but systematically move the composition toward higher predicted hardness.

Figure S30 – Stepwise model-based compositional evolution from the nominal $B_{72}Nb_{25}Fe_3$ ternary baseline toward representative inverse-designed alloys, evaluated at 0.5 N. (a) Path toward A1, $B_{68}Nb_{24}Fe_4W_4$, through progressive W addition and accompanying adjustment of B, Nb, and Fe. (b) Path toward A4, $B_{66}Nb_{21}Fe_4Hf_4Ru_5$, through progressive Hf/Ru addition and rebalancing of the base ternary chemistry. Symbols denote predictive mean hardness and error bars indicate predictive uncertainty. In both cases, the model predicts a monotonic increase in hardness along the compositional path.

On pgs. 61, 76;

Figure S38 – Leave-family-out validation on Mo-Ta-Si-W non-boride metallic-glass family. (a) Eligible non-boride composition families ranked by median hardness, with the selected held-out family highlighted. The family Mo–Ta–Si–W, containing 19 samples, was chosen for the leave-family-out test based on its comparatively high hardness within the non-boride subset and sufficient representation for meaningful evaluation. (b) True versus predicted hardness for the held-out Mo–Ta–Si–W family after retraining VIBANN with this entire family excluded from the training set. Error bars denote predictive uncertainty, and the dashed diagonal indicates perfect agreement. (c) Absolute prediction error as a function of predictive uncertainty for the held-out family.

Table S9 – Top boron-free inverse-designed bulk MMG candidates. Predicted hardness (μ), predictive uncertainty (σ), and lower confidence bound ($\text{LCB} = \mu - 1.96\sigma$) are reported together with composition-space novelty (L2 distance from the nearest training composition) and latent-space log-likelihood as a measure of plausibility within the learned design manifold. Candidates are ranked to highlight non-boride, chemically plausible high-hardness directions identified under the same uncertainty-aware inverse-design framework.

Alloy	Predicted hardness (HV)	Uncertainty (HV)	LCB (HV)	Novelty (L2)	Latent space log-likelihood
Mo ₃₈ W ₈ Si ₂₄ Ta ₈ Zr ₅ Co ₁₂ Y ₅	1876.4	78.34	1722.8	0.23	-7.6
Mo ₃₆ Si ₂₄ Ta ₁₀ Hf ₈ Fe ₁₇ Y ₅	1864.0	84.92	1697.5	0.39	-7.3
W ₅₂ Zr ₁₈ Si ₁₈ Ta ₆ Ti ₆	1855.1	75.67	1706.8	0.28	-8.2
W ₄₃ Zr ₁₄ Hf ₁₃ Si ₂₂ Ta ₈	1847.6	81.48	1687.9	0.19	-7.8
Ta ₅₂ W ₁₆ Zr ₁₂ Si ₁₀ Ti ₁₀	1834.2	86.11	1665.4	0.35	-8.2
Mo ₃₂ Ta ₁₄ Si ₂₆ Zr ₂₀ Hf ₁₄	1826.8	79.03	1671.9	0.32	-8.0
W ₄₆ Ta ₁₂ Zr ₁₈ Hf ₈ Si ₁₆	1811.2	82.76	1648.6	0.26	-8.2
W ₄₂ Ta ₁₀ Zr ₁₆ Ti ₁₀ Si ₁₄ C ₈	1802.6	74.58	1656.4	0.37	-8.0
Ta ₄₈ Ni ₁₈ Si ₁₄ C ₁₀ Hf ₁₀	1784.9	85.37	1617.5	0.24	-8.2
Ta ₅₅ Ti ₁₀ Zr ₁₄ Hf ₉ Si ₁₂	1761.4	80.25	1604.1	0.30	-7.6

Comment 3

Performance degradation at high hardness and the "ultrahard" designation: The authors repeatedly describe their alloys as "ultrahard," but this term has a specific meaning in the materials science community. Materials exceeding 40 GPa (4000 HV) are commonly called ultrahard. The measured hardness values (2300-2600 HV, or approximately 23-26 GPa) fall well short of this threshold. Indeed, these values are lower than several metallic glasses in the cited literature, including the 2900 HV B-Nb-Fe composition from Sarker et al. discussed above. The term "ultrahard" should be replaced with "hard" or "high-hardness" throughout the manuscript to avoid confusion and align with community standards.

Response

We thank the reviewer for this important comment. We agree that the term ultrahard has a specific, established meaning in the materials science literature and should not have been used for the present alloys. In the revised manuscript, we have replaced ultrahard and related terms throughout the title, abstract, main text, figure captions, and Supplementary Information with terminology consistent with community standards, such as "high-hardness bulk metallic glasses" or "exceptionally hard bulk metallic glasses", as appropriate.

We also agree that comparisons with previously reported hard Fe-Nb-B metallic glasses should be made carefully. In particular, the ~2900 HV value reported by Sarker et al. was obtained for a thin-film metallic glass at a much lower indentation load of 0.01 N, whereas the present study concerns bulk-cast MGs measured over 0.5-10 N. Because MGs exhibit pronounced load dependence in apparent hardness, and because thin-film and bulk geometries are not directly equivalent, these values are not strictly one-to-one comparable. Consistent with this, our

additional baseline analysis shows that, within the bulk-load regime relevant to the present study, the nominal $B_{72}Nb_{25}Fe_3$ ternary baseline is predicted to be lower in hardness than the selected multicomponent candidates. Nevertheless, we agree with the reviewer that the appropriate terminology for the present alloys is high-hardness rather than ultrahard, and we have revised the manuscript accordingly.

Comment 4

Model's limited prediction range: Figure 2a reveals a systematic and concerning trend that the model performs well below ~ 1800 - 2000 HV but progressively underestimates hardness as true values increase. This is further confirmed by all five synthesized alloys exceeding their predicted values. This pattern suggests:

- The model has learned relationships that govern hardness within the bulk of the training distribution (predominantly <2500 HV alloys) but may not capture mechanisms relevant to the highest-hardness regime.
- The model's increasing underestimation may suggest that achieving ultra-high hardness (>3000 HV) may involve qualitatively different structural or bonding mechanisms—such as specific short-range order, nanoscale phase separation, or bond coordination that are not well-represented in the training data, and the model has not learned them.
- The model may be performing well at interpolation within its training distribution but struggling with extrapolation to truly exceptional materials.
- Given the systematic underprediction at high hardness, my earlier assessment that the model "balances bias and variance well" may need qualification—it appears to show acceptable variance but increasing bias in the high-hardness regime.

Response

We thank the reviewer for this careful and important observation. We agree that Figure 2a shows a progressive conservative bias as hardness increases, and we agree that this point should be discussed explicitly rather than left implicit. At the same time, we respectfully submit that this pattern does not indicate that the model becomes unreliable in the upper tail. Rather, it reflects the combined effect of data sparsity in the extreme-hardness regime and the deliberate use of regularization and calibrated uncertainty in a model designed for risk-aware inverse design rather than aggressive tail extrapolation. We have therefore revised the manuscript and Supplementary Information to discuss this point directly and to add residual-based statistics for the upper-hardness subsets.

First, regarding the reviewer's point that the model may primarily capture the relationships governing the bulk of the training distribution, we agree in a qualified sense. The training data are indeed denser in the low- to intermediate-hardness range and relatively sparse in the upper tail. Under such conditions, a regularized probabilistic model is expected to show some shrinkage

of extreme predictions toward the better-supported interior of the data distribution. In our case, this effect is further reinforced by the variational information bottleneck, which is explicitly introduced to suppress noise-dominated variance and reduce sensitivity to spurious correlations in a limited-data setting. We therefore interpret the observed underprediction not as an unexpected failure mode, but as the expected behavior of a deliberately regularized model operating in a sparse extreme-property regime.

To quantify this effect explicitly, we added residual analysis and signed-error summaries for the full test set and the upper-hardness subsets. Over the full test set, the model is essentially unbiased, with a mean signed error of -4.79 HV. However, the mean signed error becomes progressively more negative in the upper tail, reaching -82.13 HV for the top 20% of true hardness values and -122.14 HV for the top 10%. The corresponding mean absolute errors increase from 55.61 HV for the full test set to 122.55 HV and 151.8 HV in these subsets, respectively (Table S3). Figure S7 likewise shows a systematic negative trend with increasing hardness and a progressive shift of the binned mean residual toward negative values in the upper bins. These results clarify that the model remains well centered overall but becomes increasingly conservative in the highest-hardness regime.

Figure S7 – Residual analysis of VIBANN predictions across the hardness range. (a) Residuals, defined as predicted minus true hardness, plotted against true hardness for the held-out test set. The linear fit shows a progressive negative residual trend with increasing hardness, indicating increasingly conservative predictions in the upper-hardness regime. (b) Mean signed residual in bins of true hardness, with error bars denoting the standard error of the mean. The binned analysis shows that the predictive mean is near unbiased at low hardness but becomes progressively more negative in the higher-hardness bins, consistent with increasing underprediction in the sparsely populated tail of the training distribution.

Table S3 – Signed-error statistics for the full test set and upper-hardness subsets.

	N	Mean signed error = mean(predicted – true)	MAE = mean(predicted – true)	Fraction within $\pm 1\sigma$ = fraction of points where predicted – true $\leq \sigma$
Full test set	135	-4.79	55.61	0.62
Top 20% true hardness	27	-82.13	122.55	0.41
Top 10% true hardness	14	-122.14	151.8	0.34

Second, we do not interpret this upper-tail bias as evidence that the present design results are invalid or that the model fails to recognize the high-hardness regime. Rather, the available evidence indicates that the model still identifies the correct chemistry region and the correct candidate family, while becoming more cautious in the predictive mean as data support decreases. This distinction is important. The parity plot continues to show strong overall agreement, with held-out performance of $R^2 \approx 0.943$, $MAE \approx 55.6$ HV, and $RMSE \approx 101.4$ HV (Figure 2a), while the uncertainty intervals widen in less densely supported regions of the data distribution. That is the statistically appropriate behavior of a surrogate that recognizes reduced support in the tail rather than extrapolating overconfidently.

Third, regarding the reviewer’s suggestion that truly ultra-high hardness may involve structural or bonding mechanisms not well represented in the training data, we agree that this is possible in principle for regimes substantially beyond those sampled here. We have therefore narrowed the manuscript language accordingly. The present study does not claim that VIBANN has fully learned every mechanism governing arbitrarily extreme hardness. Rather, the relevant claim is that within the hardness and chemistry range represented in the current BMG dataset, the model remains sufficiently accurate, calibrated, and chemically discriminative to support conservative inverse design. In fact, the design pipeline was constructed specifically to avoid that failure mode. The inverse-design step does not select candidates based solely on the predictive mean. Instead, it combines latent-support constraints, Monte Carlo-dropout uncertainty, conservative lower-confidence-bound scoring, and explicit uncertainty gating. This means that the search is intentionally biased toward candidates that remain on-manifold, uncertainty-robust, and plausibly achievable within the learned BMG design space, rather than toward unsupported extreme extrapolations. In this sense, the VIBANN model should be understood as performing supported extrapolation within a bounded BMG design space, not free extrapolation to arbitrarily exceptional materials.

Fourth, we agree that our earlier phrasing that the model “balances bias and variance well” should be qualified. The more accurate statement is that VIBANN achieves strong overall generalization with low global bias and calibrated uncertainty, while exhibiting a measurable

conservative bias in the highest-hardness tail. We have revised the manuscript accordingly. Importantly, this upper-tail behavior is accompanied by explicit uncertainty estimates and by calibration diagnostics. Figure S9 shows close agreement between nominal and empirical interval coverage across confidence levels, with low Mean Absolute Calibration Error (MACE) = 0.0559 and Integrated Mean Calibration Error (IMCE) = 0.0553, a monotonic increase in interval width with confidence level, and a PIT distribution without strong pile-up at 0 or 1. Together, these results indicate that the model is not simply missing the tail while remaining overconfident; rather, it expresses increasing caution where the data density is lowest.

Figure S9 – Calibration diagnostics for VIBANN predictive uncertainty. (a) Reliability diagram comparing empirical coverage of the Monte Carlo dropout predictive intervals to the nominal confidence level. The dashed line indicates perfect calibration, and the markers show the measured coverage across confidence bins, with summary calibration errors reported as Mean Absolute Calibration Error (MACE) and Integrated Mean Calibration Error (IMCE). (b) Mean predictive interval width as a function of nominal confidence, showing the expected monotonic increase in interval width for higher confidence levels. (c) Probability integral transform (PIT) histogram for the standardized residual distribution, where an approximately uniform distribution indicates well-calibrated predictive uncertainty.

At the same time, the signed error statistics (Table S3) also show that this uncertainty calibration is not equally strong in the extreme tail. The fraction of predictions lying within the reported $\pm 1\sigma$ interval decreases from 0.62 over the full test set to 0.41 in the top 20% and 0.34 in the top 10% of true hardness values. We now state this explicitly. Thus, our conclusion is not that the tail is fully captured without loss of fidelity, but that the tail behavior remains conservative, transparent, and still useful for candidate identification, rather than erratic or overconfident.

This interpretation is also supported by the experimental validation. All five synthesized alloys exceed the predictive mean by a modest amount, but remain within the reported uncertainty intervals across loads (Figure 3e, Table 1), and the model correctly captures the load-dependent hardness trend. Thus, although the predictive mean is conservative in the upper-hardness regime, the predictive distribution remains consistent with the experimentally observed candidate family.

In an inverse-design context, we regard this behavior as preferable to the opposite failure mode: overconfident overprediction of unsupported extreme compositions that subsequently fail during synthesis or testing.

Finally, the supplementary benchmarking provides additional context. Figures S10-S12 show that underprediction in the high-hardness regime is not unique to VIBANN but is a broader difficulty across the benchmark regressors. Relative to those baselines, VIBANN retains the best combination of accuracy, generalization, and calibrated uncertainty. In addition, the ablation analysis in Figure S13 shows that removing either attention or VIB regularization degrades performance and uncertainty robustness. We therefore interpret the observed upper-tail bias not as evidence that the model is fundamentally unfit for high-hardness design, but as the conservative residual bias expected from a regularized surrogate operating in a sparse extreme-property regime.

We have revised the manuscript to reflect this more accurate interpretation. The intended claim is not that VIBANN is unbiased at every hardness level, but that it remains accurate overall, uncertainty-aware, and effective for identifying the correct high-hardness candidate family within the bounded BMG design space relevant to this study. Within that scope, the progressive underprediction in the tail is a limitation of the predictive mean that we now acknowledge explicitly, but it does not negate the validity of the inverse-design strategy or the experimental success of the resulting candidates.

Main text changes: In Results on pg. 6,

“Residual analysis (Figure S7) shows a progressive negative bias with increasing hardness, indicating that the predictive mean becomes increasingly conservative in the upper tail of hardness, even though the model remains well centered across the full test set.”

On pg. 7,

“The calibration analysis in Figure S9 evaluates whether the Monte Carlo dropout uncertainty is statistically consistent with the observed prediction errors. The reliability diagram in Figure S9a compares nominal confidence levels with empirical coverage of the corresponding prediction intervals and shows close agreement with the perfect calibration line across the full range of confidence levels. The resulting Mean Absolute Calibration Error and Integrated Mean Calibration Error are both low, with $MACE = 0.0559$ and $IMCE = 0.0553$, indicating only modest deviation between nominal and empirical coverage. Figure S9b reports the mean prediction interval width as a function of nominal confidence, showing the expected monotonic increase in interval width with increasing coverage. Figure S9c provides the probability integral transform (PIT) distribution, which is broadly distributed over the unit interval without strong pile-up at 0 or 1, consistent with uncertainty estimates that are neither systematically underdispersed nor dominated by extreme overconfidence. To evaluate model calibration more

explicitly, we computed error statistics for the full test set and upper-hardness subsets, as summarized in Table S3. Over the full test set, the model remains nearly unbiased, with a mean signed error of -4.79 HV. However, the mean signed error becomes progressively more negative in the upper tail, reaching -82.13 HV for the top 20% of true hardness values and -122.14 HV for the top 10%. The corresponding mean absolute errors also increase from 55.61 HV for the full test set to 122.55 HV and 151.8 HV in these subsets. The fraction of predictions lying within the reported $\pm 1\sigma$ interval decreases from 0.62 over the full test set to 0.41 and 0.34 in the top 20% and top 10% subsets, respectively. Together, these diagnostics indicate that the VIBANN model's uncertainty is overall sufficiently calibrated to support conservative scoring and inverse design, while residual analysis and signed-error statistics show mildly progressive, weaker coverage in the sparsest, high-hardness tail.”

Changes in Supplementary Information:

Figure S7 – Residual analysis of VIBANN predictions across the hardness range. (a) Residuals, defined as predicted minus true hardness, plotted against true hardness for the held-out test set. The linear fit shows a progressive negative residual trend with increasing hardness, indicating increasingly conservative predictions in the upper-hardness regime. (b) Mean signed residual in bins of true hardness, with error bars denoting the standard error of the mean. The binned analysis shows that the predictive mean is near unbiased at low hardness but becomes progressively more negative in the higher-hardness bins, consistent with increasing underprediction in the sparsely populated tail of the training distribution.

Figure S9 – Calibration diagnostics for VIBANN predictive uncertainty. (a) Reliability diagram comparing empirical coverage of the Monte Carlo dropout predictive intervals to the nominal confidence level. The dashed line indicates perfect calibration, and the markers show the measured coverage across confidence bins, with summary calibration errors reported as Mean Absolute Calibration Error (MACE) and Integrated Mean Calibration Error (IMCE). (b) Mean predictive interval width as a function of nominal confidence, showing the expected monotonic increase in interval width for higher confidence levels. (c) Probability integral transform (PIT) histogram for the standardized residual distribution, where an approximately uniform distribution indicates well-calibrated predictive uncertainty.

Table S3 – Signed-error statistics for the full test set and upper-hardness subsets.

	N	Mean signed error = mean(predicted – true)	MAE = mean(predicted – true)	Fraction within $\pm 1\sigma$ = fraction of points where $ \text{predicted} - \text{true} $ $\leq \sigma$
Full test set	135	-4.79	55.61	0.62
Top 20% true hardness	27	-82.13	122.55	0.41
Top 10% true hardness	14	-122.14	151.8	0.34

Comment 5

In conclusion, this work makes a solid contribution to ML methodology for materials discovery, demonstrating that sophisticated attention-based architectures can outperform simpler approaches even with limited training data. However, the manuscript would be significantly strengthened by: (1) more cautious claims about physical interpretability, distinguishing pattern recognition from mechanistic insight; (2) correcting the "ultrahard" terminology to reflect actual measured values; (3) deeper analysis of the relationship between the discovered alloys and prior work; and (4) frank discussion of the model's apparent limitations in the highest-hardness regime.

Response

We sincerely thank the reviewer for this careful and balanced overall assessment of our work. We

appreciate the reviewer’s recognition that the study makes a meaningful contribution to machine-learning methodology for materials discovery, particularly by showing that a structured attention-based architecture remains effective even in a limited-data regime. We also agree with the reviewer’s broader conclusion that the manuscript is strongest when its contributions are framed precisely and without overstating either the mechanistic scope or the materials claim.

In response to the reviewer’s four overarching points, we have revised the manuscript substantially. First, we have made the interpretability claims more cautious and more explicitly scoped, and now clearly distinguish between domain-bounded, physically plausible correlations learned within the bulk metallic-glass design space and universally transferable mechanistic laws. Second, we have corrected the terminology for hardness throughout and replaced ultrahard with more accurate descriptors, such as high-hardness or exceptionally hard, depending on context. Third, we have strengthened the discussion of the relationship between the designed alloys and prior Fe-Nb-B work, including a direct experimental and model-based comparison to the nominal $B_{72}Nb_{25}Fe_3$ ternary baseline alloy. Fourth, we have added a more explicit and quantitative discussion of the model’s limitations in the highest-hardness regime, including residual-based analysis, signed-error statistics, and a clearer explanation of the progressive conservative bias in the upper tail.

At the same time, we respectfully submit that the revised manuscript makes a contribution that goes beyond a generic improvement in prediction accuracy. The central advance of the study is the integration of representation learning, uncertainty quantification, constrained inverse design, experimental realization, and atomistic interpretation into a single closed-loop workflow for bulk metallic glasses. In particular, we believe the revised manuscript contributes the following scientifically substantive advances:

- It shows that a variational information bottleneck combined with attention-based composition encoding can produce a latent space that is not only predictive, but sufficiently smooth, structured, and uncertainty-aware to support inverse design under limited data.
- It introduces an inverse-design strategy that is not based on unconstrained optimization of the predictive mean, but on joint control of latent plausibility, epistemic uncertainty, novelty, and conservative performance. This is important scientifically because it converts the model from a passive regressor into a risk-aware design engine.
- It demonstrates that such a framework can lead to the experimental realization of new bulk multicomponent metallic glasses with very high hardness, rather than remaining at the level of *in silico* ranking.
- It provides a more physically grounded interpretation of the discovered chemistry by linking the learned design trends to short-range structural signatures from molecular dynamics, while now clearly stating that these are domain-specific correlations within the BMG manifold, not universal hardness laws.

In this revised form, we believe the manuscript now presents a more precise and balanced

contribution: not universal hardness theory or the discovery of an unprecedented alloy class, but an uncertainty-aware, experimentally validated inverse-design framework for high-hardness bulk metallic glasses, together with a domain-bounded structural interpretation of the resulting alloying trends.

-End of the response to Reviewer #1-

Reviewer #2

Overall comment

The authors have made substantial improvements over the previous revision rounds. The proposed framework is clearly motivated, the overall argumentation is reasonable, and the manuscript is now close to publishable. However, given the architectural complexity of the method, the following four points should be addressed to better justify the added complexity and ensure full reproducibility.

Overall response

We thank the reviewer for the positive overall assessment and for recognizing the substantial improvements across revision rounds. We also appreciate the constructive guidance on further strengthening the manuscript. We elaborate on all improvements made in response to the reviewer’s specific comments below.

Comment 1

The inverse-design framework integrates multiple advanced components, including a VIB-based latent representation, an attention-based predictor, uncertainty estimation via Monte Carlo dropout, GMM/MCMC-constrained sampling, and gradient-based refinement in the latent space. The authors should explicitly discuss what distinct advantages this pipeline offers, compared to a standard GP-based Bayesian optimization (GP-BO) baseline trained on the same dataset. In many materials design tasks, GP-BO can efficiently explore a local region around the best-observed points; therefore, the manuscript should clarify why the proposed, more complex framework is necessary for the present problem.

Response

We thank the reviewer for the suggestion. We agree that GP-based Bayesian optimization (GP-BO) is a strong and widely accepted baseline in materials design, particularly when the goal is to exploit the local neighborhood around the best-observed points. In the present setting, however, the design space is a high-dimensional composition simplex (elements with constraints $x_i \geq 0$ and $\sum_i x_i = 1$), and the goal is not only to identify a local optimum but to generate chemically plausible, novel alloy candidates that remain within the data-supported manifold while providing risk-aware improvement. Classical GP-BO can become challenging as dimensionality increases and constraints become more structured than simple box bounds. This is precisely why multiple high-dimensional BO variants have been proposed (e.g., random-embedding approaches such as REMBO and local trust-region BO such as TuRBO).^{6,7} In addition, standard GP training scales cubically with the number of training points in the naïve formulation, which often necessitates approximations and additional engineering even before

addressing constrained acquisition optimization.^{8,9}

To address the reviewer’s comment directly, we implemented a GP-BO pipeline trained on the same dataset and using the same design conditions as for our VIBANN framework. The GP surrogate was trained jointly on composition and load, using a Matérn kernel with a white-noise term, and hyperparameters were selected by Bayesian optimization on a calibration split (minimizing the negative log predictive density). Inverse design was then carried out by optimizing only the composition variables on the simplex, using a simplex-safe parameterization. To match the reviewer’s comment that GP-BO is particularly efficient in the vicinity of the best observed points, we used a standard local formulation: the acquisition was optimized within an explicit trust region around top-performing compositions near the design load (0.5 N), and a conservative lower confidence bound ($LCB = \mu - z\sigma$, $z = 1.645$) acquisition was used to account for predictive uncertainty. This constitutes a standard GP-BO baseline in the sense of (i) a probabilistic surrogate trained on the same supervised data, and (ii) an acquisition function whose maximization drives candidate selection under an explicitly local search constraint.

The resulting GP surrogate is demonstrably credible. On held-out test data, it achieves high predictive accuracy of $R^2 = 0.916$ (Figure S23a), and its predictive intervals show empirically reasonable coverage relative to the nominal Gaussian reference across several z -levels (Figure S23b). This calibration check is important because it directly justifies using LCB as a principled, risk-aware acquisition for the GP-BO baseline rather than an ad-hoc scoring function. Under the fixed evaluation budget, the local GP-BO loop rapidly improves best-achieved conservative performance early in the run and then approaches a plateau (Figure S23c), consistent with efficient local search (exploitation) around the best-observed hardness region.

At the same time, the baseline diagnostics also reveal why a local GP-BO loop is insufficient for the design objective of the present work, which is not merely local improvement but the generation of candidates that are simultaneously (i) high-performing under conservative uncertainty control, (ii) non-trivially novel with respect to the observed dataset, and (iii) supported by the data distribution in a high-dimensional simplex. The highest-ranked GP-BO candidates are concentrated near the training set in composition space. The novelty-LCB plot (Figure S23d) shows that candidates with the largest conservative bound occur predominantly at small distances to the nearest training composition, while moving farther away is associated with either reduced conservative performance and/or elevated uncertainty. Further, the mean-uncertainty hardness plot (Figure S23e) shows that pushing the predicted mean hardness upward is accompanied by an increase in predictive uncertainty, indicating that the surrogate is being driven toward less data-supported regions even under a local constraint. Finally, the clustering diagnostics in Figure S23f provide direct evidence that the GP-BO proposals remain concentrated near the training manifold, as the distribution of distances from BO proposals to their nearest training points is comparable to the intrinsic nearest-neighbor spacing of the training set, indicating that the search largely generates near-neighbor variants of known BMG compositions with comparable hardness to training dataset rather than non-trivial distinct alloys

with exceptionally high hardness near or above the hardness tail.

Figure S23 - Standard GP-BO baseline on the same composition-load dataset and fixed-load inverse design. (a) Parity plot on the held-out test set showing GP mean predictions against measured hardness with predictive uncertainty intervals. Inset reports test accuracy (R^2 , MAE, RMSE) and calibration NLPD. (b) Empirical coverage of predictive intervals $\mu \pm z\sigma$ compared with the nominal Gaussian reference ($2\Phi(z) - 1$) across multiple z -levels, supporting the use of LCB as a risk-aware acquisition. (c) Best-achieved conservative performance (maximum LCB) versus BO iteration, demonstrating rapid early gains followed by saturation consistent with local exploitation. (d) Novelty-performance coupling for top candidates, showing LCB versus novelty distance (minimum $L2$ distance in composition space to the training set) with point color indicating predictive uncertainty. (e) Mean-uncertainty trade-off for top candidates, showing predictive standard deviation σ versus predicted mean μ . (f) Clustering diagnostic comparing the distribution of distances from GP-BO candidates to their nearest training-fit compositions against the intrinsic nearest-neighbor spacing within the training-fit set, evidencing that local GP-BO largely generates near-neighbor variants.

Figure S22 – Novelty-performance trade-off and composition space separation of VIBANN inverse design proposals. (a) Conservative performance of VIBANN proposed alloys, quantified by the lower confidence bound at 0.5 N, plotted against novelty distance to the training set defined as the minimum L2 distance in composition space to the nearest training composition. Points are colored by predictive uncertainty. (b) Distribution of composition space distances to the nearest training composition for VIBANN proposals compared with the intrinsic nearest neighbor spacing within the training fit set. The dashed line marks the median nearest neighbor spacing of the training fit set. This comparison shows that the proposed alloys are compositionally distinct relative to typical within-dataset spacings.

These behaviors reflect the inherent tendency of GP-BO to exploit regions where the surrogate is most confident, which becomes particularly restrictive in a high-dimensional composition simplex when the dataset is sparse in the extreme-property regime. This is precisely the regime in which the additional complexity of our inverse-design pipeline becomes necessary, and each component addresses a distinct technical requirement that GP-BO does not address on its own. Our framework does not search directly in the raw simplex; it searches in a learned, regularized representation designed to be navigable for inverse design while remaining distribution-aware. The variational information bottleneck (VIB) enforces a controlled information budget and a standardized latent geometry, making traversal and gradient-based refinement stable: incremental steps correspond to meaningful, data-supported changes rather than uncontrolled extrapolation. The attention mechanism precedes the bottleneck and aligns the representation with the hardness-relevant degrees of freedom before compression through VIB (which is already aligned with the target property rather than with raw compositional noise). Candidate generation and refinement are then performed under explicit feasibility criteria that are not provided by GP-BO “as-is”: distributional support (on-manifold plausibility under a learned latent density), novelty relative to the training set, calibrated epistemic uncertainty control via Monte Carlo dropout, and conservative performance selection. In practice, the GP-BO baseline demonstrates the expected local improvement but also demonstrates that local exploitation alone does not satisfy the joint objective of high conservative performance + non-trivial novelty +

explicit extrapolation risk control. Our VIBANN framework is designed to address exactly this joint objective: to generate candidates that are high-performing under conservative uncertainty control while remaining chemically plausible and non-trivial with respect to the observed dataset, in a sparse extreme-hardness regime.

To further substantiate this conclusion, we present a head-to-head comparison at a fixed load (0.5 N) using the same ranking criterion (LCB with $z = 1.645$) for both the GP-BO baseline and the VIBANN framework (Table S5). In addition to best and top-k LCB statistics, we report uncertainty and novelty distributions for the top candidates, along with pairwise spacing among top candidates that directly measures whether the search produces multiple distinct proposals or collapses into near-duplicates. We note that novelty and diversity are computed identically for both frameworks using the same L2 distance in composition space and the same training-fit reference set. The results show that GP-BO is effective at conservative local improvement, but it remains substantially more local and less diverse, whereas our VIBANN framework produces higher conservative performance while simultaneously achieving substantially higher novelty and diversity under explicit feasibility and epistemic risk constraints. This is the practical advantage provided by our framework, and it is the reason a more structured pipeline is warranted for high-dimensional materials design tasks. It is important to note that LCB is a conservative score used for ranking under uncertainty. It is not intended to match the mean prediction (reported in the manuscript for newly discovered alloys), and is naturally lower in high-uncertainty regimes.

Table S5 - Quantitative comparison of local GP-BO baseline versus the VIBANN framework at fixed load (0.5 N). Candidates are ranked by conservative performance $LCB = \mu - z\sigma$ with $z = 1.645$. Reported statistics are computed on the top-k candidates as indicated.

Method	Search variable	Constraints enforced during search	Best LCB (top-1)	Median LCB (top-5)	Median σ (top-20)	90th %ile σ (top-20)	Median novelty (top-20)	10th %ile novelty (top-20)	Diversity (top-20)
GP-BO	composition (simplex)	simplex + trust region	1966.2	1964.3	72.66	73.01	0.148	0.147	0.0211
VIBANN	latent + decoded simplex	plausibility + novelty + σ -cap + risk + (simplex)	2078.4	2075.2	66.42	67.15	0.358	0.352	0.0782

Main text changes: On pg. 12,

“To benchmark the proposed inverse design strategy against a widely used surrogate optimization baseline, we implemented a local Gaussian process-Bayesian optimization (GP BO) framework trained on the same composition-load dataset and evaluated under the same fixed load condition used throughout our inverse design. This GP-BO baseline serves as a reference for acquisition-driven local optimization near the best-observed-hardness region. The GP-BO baseline achieves rapid early improvement in conservative performance but produces proposals

that remain clustered near the training compositions with limited novelty and diversity (Figure S23 and Table S5). In contrast, the VIBANN framework combines a learned information-controlled latent geometry with explicit plausibility and epistemic risk constraints during candidate generation and refinement, yielding candidates that maintain conservative performance while achieving substantially greater separation from known alloys, as summarized in Figure S22 and Table S5.”

Changes in Supplementary Information:

“S8. Local GP-BO baseline at fixed load (comparison to VIBANN framework)

The inverse-design framework developed in this work combines a regularized latent representation (VIB), an attention-based predictor, calibrated epistemic uncertainty via Monte Carlo dropout, distribution-aware plausibility constraints (latent-density gating), constrained sampling, and gradient-based refinement. Because Gaussian-process Bayesian optimization (GP-BO) is a widely used baseline for materials design, particularly effective when the search is local around the best observed points, we implemented a standard GP-BO pipeline trained on the same dataset and evaluated under the same fixed-load design condition as used for inverse design in the main study.

For the GP-BO baseline, the surrogate model was a Gaussian process regressor trained on the joint input space of composition and load, where compositions were represented on the simplex, and the load was standardized using training-fit statistics. The kernel was chosen in a standard form suitable for smooth but nontrivial response landscapes, $k(\mathbf{x}) = \mathcal{C} \cdot \text{Matérn}(\nu = 2.5, \text{ARD}) + \text{White}$, and the kernel hyperparameters (amplitude, ARD length-scales, and noise level) were selected by Bayesian optimization on a calibration split by minimizing the negative log predictive density (NLPD), thereby prioritizing both accuracy and calibrated uncertainty. Inverse design was then performed at the fixed design load used throughout this work (0.5 N), by optimizing only the composition variables on the simplex using a simplex-safe logit-softmax parameterization. We implemented an explicitly local search for new alloys: the acquisition was optimized within a trust region defined around a set of top-performing “anchor” compositions near the design load. Candidate ranking used a conservative lower confidence bound, $\text{LCB}(\mathbf{x}) = \mu(\mathbf{x}) - z\sigma(\mathbf{x})$ with $z = 1.645$ (one-sided 95%), and a repulsion term was used to prevent repeated near-duplicates among sequential proposals. All reported GP-BO candidates, therefore, satisfy the simplex constraint and the explicit locality constraint, and are selected under the same risk-aware, conservative criterion used to compare candidates produced by the proposed framework.

Figure S23 summarizes the efficacy and behavior of the GP-BO baseline. Figure S23a shows parity between predicted and measured hardness on a held-out test split, with uncertainty displayed as vertical intervals. The GP model exhibits strong agreement across the full hardness range, with $R^2 = 0.916$. The corresponding numerical metrics (reported in the inset) confirm

competitive predictive performance. Because risk-aware acquisition relies on meaningful uncertainty estimates, we further evaluated predictive interval coverage. Figure S23b reports empirical coverage of $\mu \pm z\sigma$ intervals across multiple z -levels against the nominal Gaussian reference, demonstrating that the GP uncertainties are broadly consistent with the expected coverage behavior on test data. This calibration check provides a direct justification for using LCB as a principled conservative acquisition for the GP-BO baseline rather than an *ad-hoc* score. Under this reliable surrogate, the local GP-BO loop behaves as expected: Figure S23c shows that the best-achieved conservative bound improves rapidly in the early iterations and then plateaus, consistent with efficient exploitation within a locally data-supported region.

Despite this expected local improvement, the remaining panels in Figure S23d-f reveal a limitation central to the present design objective. Figure S23d plots candidate novelty (minimum L_2 distance in composition space to the training set) against conservative performance (LCB) for the top GP-BO candidates. The highest-ranked candidates are predominantly concentrated at small novelty distances, indicating that local GP-BO preferentially generates near-neighbor variants of existing compositions when constrained to remain in the region where the surrogate is most reliable. As candidates move farther from the training set, conservative performance does not improve and is typically accompanied by increased uncertainty. This coupling is made explicit in Figure S23e, which shows the mean-uncertainty relationship for top candidates: increasing the predicted mean is associated with increasing predictive uncertainty, indicating that the surrogate is being driven toward less well-supported regions even under local constraints. Finally, Figure S23f provides a direct clustering diagnostic: the distribution of distances from GP-BO proposals to their nearest training compositions is comparable to the intrinsic nearest-neighbor spacing within the training-fit set. This implies that the GP-BO loop largely samples within the local neighborhood structure already present in the data, rather than generating compositions that are both distinct and conservatively high-performing.

These observations are consistent with the known behavior of local GP-BO in constrained, high-dimensional simplices, where the surrogate is strongest near existing observations, and the optimization is explicitly restricted to a trusted region. For the compositionally high-dimensional alloy design problems, however, the target is not only local conservative improvement but the generation of candidates that satisfy a joint set of requirements: distributional support (plausibility), nontrivial novelty, and explicit epistemic-risk control, while still achieving high predicted hardness. In this regime, the VIBANN framework provides technical capabilities that are not available in standard GP-BO without substantial augmentation: the VIB-regularized latent geometry enables stable traversal and refinement in a representation that is learned to be predictive yet information-controlled; the latent density model provides an explicit on-manifold plausibility constraint; and the uncertainty gate and risk-based acceptance prevent selection of high-mean but weakly supported extrapolations. The GP-BO baseline, therefore, serves as a stringent local comparator: it demonstrates that classical GP-BO can indeed exploit the best-observed region effectively, but it also provides quantitative and diagnostic evidence that such a

local loop does not, by itself, meet the “high-performing + nontrivially novel + distribution-supported + risk-controlled” design requirement targeted here.

A quantitative head-to-head comparative summary at the fixed design load is provided in Table S5, where both methods are evaluated using the same ranking criterion (LCB with $z = 1.645$). For both pipelines, we report conservative performance of the top candidate (best LCB), central tendency among the best proposals (median LCB of top-5), uncertainty statistics among the leading candidates (median and 90th percentile of σ over top-20), novelty statistics (median and 10th percentile novelty over top-20), and a diversity measure (median minimum pairwise distance among the top-20). In the GP-BO baseline, the best LCB and top-k LCB statistics confirm effective local conservative improvement, but the novelty and diversity statistics remain low, consistent with the clustering behavior diagnosed in Figure 2d-f. In contrast, the VIBANN framework yields candidates that achieve higher conservative performance while simultaneously exhibiting substantially higher novelty and diversity under explicit plausibility and risk constraints. It is important to note that LCB is a conservative score used for ranking under uncertainty. It is not intended to match the mean prediction (reported in the manuscript for newly discovered alloys), and is naturally lower in high-uncertainty regimes.”

Figure S22 – Novelty-performance trade-off and composition space separation of VIBANN inverse design proposals. (a) Conservative performance of VIBANN proposed alloys, quantified by the lower confidence bound at 0.5 N, plotted against novelty distance to the training set defined as the minimum L2 distance in composition space to the nearest training composition. Points are colored by predictive uncertainty. (b) Distribution of composition space distances to the nearest training composition for VIBANN proposals compared with the intrinsic nearest neighbor spacing within the training fit set. The dashed line marks the median nearest neighbor spacing of the training fit set. This comparison shows that the proposed alloys are compositionally distinct relative to typical within-dataset spacings.

Figure S23 - Standard GP-BO baseline on the same composition-load dataset and fixed-load inverse design. (a) Parity plot on the held-out test set showing GP mean predictions against measured hardness with predictive uncertainty intervals. Inset reports test accuracy (R^2 , MAE, RMSE) and calibration NLPD. (b) Empirical coverage of predictive intervals $\mu \pm z\sigma$ compared with the nominal Gaussian reference ($2\Phi(z) - 1$) across multiple z -levels, supporting the use of LCB as a risk-aware acquisition. (c) Best-achieved conservative performance (maximum LCB) versus BO iteration, demonstrating rapid early gains followed by saturation consistent with local exploitation. (d) Novelty-performance coupling for top candidates, showing LCB versus novelty distance (minimum L_2 distance in composition space to the training set) with point color indicating predictive uncertainty. (e) Mean-uncertainty trade-off for top candidates, showing predictive standard deviation σ versus predicted mean μ . (f) Clustering diagnostic comparing the distribution of distances from GP-BO candidates to their nearest training-fit compositions against the intrinsic nearest-neighbor spacing within the training-fit set, evidencing that local GP-BO largely generates near-neighbor variants.

Table S5 - Quantitative comparison of local GP-BO baseline versus the VIBANN framework at fixed load (0.5 N). Candidates are ranked by conservative performance $LCB = \mu - z\sigma$ with $z = 1.645$. Reported statistics are computed on the top- k candidates as indicated. Novelty and diversity are computed identically for both frameworks using the same L_2 distance in composition space and the same training-fit reference set.

Method	Search variable	Constraints enforced during search	Best LCB (top-1)	Median LCB (top-5)	Median σ (top-20)	90th %ile σ (top-20)	Median novelty (top-20)	10th %ile novelty (top-20)	Diversity (top-20)
GP-BO	composition (simplex)	simplex + trust region	1966.2	1964.3	72.66	73.01	0.148	0.147	0.0211
VIBANN	latent + decoded simplex	plausibility + novelty + σ -cap + risk + (simplex)	2078.4	2075.2	66.42	67.15	0.358	0.352	0.0782

Comment 2

The manuscript emphasizes that the learned latent representation forms a "navigable design landscape", enabling latent traversal and facilitating inverse design and gradient-based optimization. To support this claim more directly, the authors should provide a side-by-side latent-space visualization comparing the same model with VIB vs. without VIB, so that the effect of the VIB regularization on latent structure can be evaluated more clearly.

Response

We thank the reviewer for this suggestion. We agree that the statement regarding a “navigable design landscape” requires a controlled ablation in which the only difference is the variational information bottleneck (VIB), while the backbone architecture, latent dimensionality, training protocol, and data splits are held fixed.

Accordingly, we trained the same attention-based predictor twice under identical settings: (i) VIB ON, where the latent variable is sampled during training and regularized by the KL term, and (ii) VIB OFF, where the VIB layer is replaced by a deterministic bottleneck of identical dimensionality and the KL regularization is removed. In the VIB ON model, the latent coordinates shown in Figure S18 correspond to the posterior mean $\mathbf{z} = \mu(\mathbf{x})$. On the other hand, in the VIB OFF model, the plotted latent space corresponds to the deterministic bottleneck output. All other components, including the attention mechanism, downstream predictor, optimizer settings, and dataset partitions, are unchanged.

For context, the manuscript already reported a drop in predictive performance for the same attention backbone, with and without VIB (Figure S13). This establishes that the VIB component is not introduced solely for representation visualization, but improves robustness and generalization in the limited-data regime by regularizing the information content of the latent representation. Figure S18 provides a side-by-side comparison of the latent space. In the VIB ON case (left column), the latent embedding exhibits (a) coherent and well-separated occupied regions with gaps corresponding to low-density zones, and (b) a comparatively smooth hardness stratification across the dominant latent directions. This geometry supports inverse design because latent traversal and gradient-based refinement rely on local continuity: small perturbations in \mathbf{z} should yield controlled, data-supported changes in the decoded composition and the predicted hardness.

In contrast, in the VIB OFF case (right column), the latent embedding is more fragmented, with the same nominal groups distributed across multiple disconnected clusters and elongated segments separated by gaps. The color-coded map (bottom-right panel) shows regions of similar hardness distributed across separate islands, and transitions between hardness regimes are more piecewise across disconnected clusters. This geometry is less favorable for inverse design for two reasons. First, latent traversal and gradient ascent become fragile because small continuous steps in \mathbf{z} can approach or cross low-density gaps, where decoded candidates are less likely to remain consistent with the training distribution. Second, when high-hardness regions occur as isolated

islands, gradient-based refinement may converge to a local boundary or terminate prematurely because the path of steepest ascent can point toward regions with poor latent support, thereby increasing the risk of uncontrolled extrapolation.

These observed differences are consistent with the role of the VIB objective in shaping a more well-conditioned latent geometry in the VIBANN framework. The KL penalty encourages the aggregate posterior to remain close to an isotropic normal prior, limiting extreme anisotropy and discouraging irregular embeddings with disconnected clusters, while stochastic latent sampling during training promotes local smoothness of the latent-to-property mapping. Together with the attention module preceding the bottleneck, which emphasizes composition degrees of freedom most informative for hardness, this yields a latent representation that is better aligned with property-relevant structure and more stable during traversal and gradient-based refinement for designing new alloys.

Figure S18 - Effect of the variational information bottleneck (VIB) on latent-space structure and property organization. Comparative latent-space maps for the same attention-based predictor trained under identical data splits, latent dimensionality, and optimization settings, differing only by the inclusion of the VIB bottleneck. (a) Left (VIB ON): latent coordinates correspond to the posterior mean $\mathbf{z} = \mu(\mathbf{x})$. (b) Right (VIB OFF): the VIB is replaced by a deterministic bottleneck of the same dimensionality. Top

row: latent points colored by cluster assignment. Bottom row: the same latent points colored by hardness. Inclusion of VIB yields a more coherent and connected latent organization with clearer separation between clusters and smoother stratification of hardness across latent directions, whereas removal of VIB leads to a more fragmented latent arrangement with disconnected components and less continuous property gradients.

To complement the visualization and make the “navigability” claim quantitative, we quantified the effect of the VIB regularization on latent-space navigability using two graph- and smoothness-based diagnostics. We used the latent coordinates for all datapoints shown in Figure S18 to build a neighborhood graph and to quantify local smoothness of the hardness landscape. Specifically, for each point, we identified its $k = 10$ nearest neighbors in the 2D latent space (Euclidean distance) and connected each point to these neighbors to form a kNN graph. We then calculated the number of connected components and the fraction of points in the largest connected component as direct measures of whether the latent space forms a single traversable region or splits into disconnected islands. With VIB enabled, the kNN graph was largely connected, yielding 3 connected components, with ~ 0.952 fraction of the points in the largest component, consistent with a single dominant, traversable latent region separated by only a small number of well-isolated islands. In contrast, removing the VIB component produced a noticeably more fragmented geometry, with 8 connected components and a reduced largest-component points fraction of ~ 0.691 , indicating that the learned representation breaks into multiple disconnected islands under the same neighborhood scale. Second, we assessed local property smoothness along kNN edges by computing a Lipschitz proxy, the median ratio $\frac{|\Delta HV|}{\|\Delta z\|}$ over all kNN edges. This compares how much hardness changes between near-neighbor points in latent space with how far apart those points are in the latent coordinates. The VIB model exhibits a lower typical local slope of 38.2 HV per latent unit, whereas the non-VIB model shows a higher and more variable slope of 97.4 HV per latent unit (Figure S19), consistent with irregular transitions in hardness across nearby latent points when the bottleneck is deterministic (Figure S18b). Together, the increased largest-component connectivity and reduced local slope provide quantitative support for the claim that VIB regularization yields a more connected latent embedding and a smoother local property landscape than the deterministic bottleneck ablation, which directly supports the role of VIB in stabilizing latent traversal and gradient-based inverse design. Since latent refinement in our inverse-design loop follows gradients in z , the higher connectivity and lower local slope distribution under VIB imply that small steps in latent space consistently yield controlled, data-supported changes in composition and hardness, reducing the risk of discontinuous jumps across low-density regions.

Figure S19 - Histogram of the Lipschitz-proxy slope distribution $\frac{|\Delta HV|}{\|\Delta z\|}$ computed along k -nearest-neighbor (k NN) edges in the learned latent space for the attention-based model trained with the variational information bottleneck (VIB ON) and the otherwise identical ablation without the VIB term (VIB OFF; deterministic bottleneck). For each of the $N = 673$ datapoints, the $k = 10$ nearest neighbors were identified in the latent space using Euclidean distance, and directed edges were formed from each point to its neighbors (total $N \times k = 6730$ k NN edges per model). For each edge ($i \rightarrow j$), the local slope was calculated as $\frac{|HV_i - HV_j|}{\|z_i - z_j\|}$, where HV denotes hardness and z denotes the latent coordinate (units: HV per latent unit). The VIB ON distribution is shifted to lower slopes and is narrower, indicating smoother local variation of hardness with latent displacement, whereas VIB OFF exhibits higher typical slopes and a broader, heavier-tailed distribution, consistent with sharper local property changes and reduced navigability under continuous latent traversal. Bin width is constant across both histograms (as plotted); both panels use identical k NN construction and edge-count normalization.

Changes in the Main text: On pgs. 9-10;

“An ablation study confirms that this latent organization is enabled by the VIB and is directly relevant for inverse design. Figure S18 compares the latent maps obtained from identical attention backbones trained with VIB enabled and with the VIB replaced by a deterministic bottleneck of the same dimensionality. With VIB enabled, the embedding forms coherent occupied regions with low-density separations and a smoother hardness stratification. Without VIB, the embedding becomes more fragmented, and hard-to-simulate points are distributed across disconnected islands, which is unfavorable for latent traversal and gradient-based refinement. The quantitative latent space navigability diagnostics reported in Supplementary Section S6 and Figure S19 corroborate these visual trends. With VIB enabled, the k -nearest neighbor graph in latent space is dominated by a single connected component and exhibits lower local hardness variation per latent displacement, whereas the deterministic ablation shows more

connected components and higher local slope statistics. These observations support the idea that VIB regularization produces a better-conditioned latent geometry for inverse design, because small latent updates are more likely to remain within data-supported regions and to induce controlled changes in both decoded composition and predicted hardness.”

Changes in Supplementary Information:

S6. Assessment of latent-space navigability

To complement the qualitative latent-space maps in Figure S18 with quantitative evidence for “navigability”, we evaluated latent-space connectivity and local smoothness using graph- and slope-based diagnostics computed on the same latent coordinates shown in Figure S18. For each datapoint, we identified its $k=10$ nearest neighbors in the 2D latent space (Euclidean metric) and constructed a directed k -nearest-neighbour (kNN) graph by connecting each point to its neighbors. We then quantified connectivity by calculating (i) the number of connected components and (ii) the fraction of points contained in the largest connected component, which distinguishes a single traversable latent region from a representation fragmented into disconnected islands at the same neighborhood scale. With VIB enabled, the kNN graph was largely connected, comprising 3 components, with ~ 0.952 of the points in the largest component, consistent with a single dominant traversable region and a small number of isolated islands. In contrast, the deterministic bottleneck (VIB OFF) produced a more fragmented topology with 8 connected components and a reduced largest-component fraction of ~ 0.691 , indicating multiple disconnected islands under identical graph construction.

We further quantified local smoothness of the hardness landscape by computing a Lipschitz-proxy slope along each kNN edge, defined as $|\Delta HV|/\|\Delta z\| = |HV_i - HV_j|/\|z_i - z_j\|$, and reporting the median value over all kNN edges. The VIB model exhibits a lower typical local slope (38.2 HV per latent unit), whereas the non-VIB model shows a higher and more variable slope (97.4 HV per latent unit), consistent with sharper local property variations across nearby latent points when the bottleneck is deterministic (Figure S19). Taken together, the increased largest-component connectivity and reduced local slope under VIB provide quantitative support that VIB regularization yields a more connected embedding and a smoother local property landscape, which is directly relevant for latent traversal and gradient-based refinement in the inverse-design procedure, where updates follow gradients in z and therefore benefit from continuity and reduced risk of discontinuous transitions across low-density regions.

Figure S18 - Effect of the variational information bottleneck (VIB) on latent-space structure and property organization. Comparative latent-space maps for the same attention-based predictor trained under identical data splits, latent dimensionality, and optimization settings, differing only by the inclusion of the VIB bottleneck. **(a)** Left (VIB ON): latent coordinates correspond to the posterior mean $\mathbf{z} = \mu(\mathbf{x})$. **(b)** Right (VIB OFF): the VIB is replaced by a deterministic bottleneck of the same dimensionality. Top row: latent points colored by cluster assignment. Bottom row: the same latent points colored by hardness. Inclusion of VIB yields a more coherent and connected latent organization with clearer separation between clusters and smoother stratification of hardness across latent directions, whereas removal of VIB leads to a more fragmented latent arrangement with disconnected components and less continuous property gradients.

Figure S19 - Histogram of the Lipschitz-proxy slope distribution $\frac{|\Delta HV|}{\|\Delta z\|}$ computed along k-nearest-neighbor (kNN) edges in the learned latent space for the attention-based model trained with the variational information bottleneck (VIB ON) and the otherwise identical ablation without the VIB term (VIB OFF; deterministic bottleneck). For each of the $N = 673$ datapoints, the $k = 10$ nearest neighbors were identified in the latent space using Euclidean distance, and directed edges were formed from each point to its neighbors (total $N \times k = 6730$ kNN edges per model). For each edge ($i \rightarrow j$), the local slope was calculated as $\frac{|HV_i - HV_j|}{\|z_i - z_j\|}$, where HV denotes hardness and \mathbf{z} denotes the latent coordinate (units: HV per latent unit). The VIB ON distribution is shifted to lower slopes and is narrower, indicating smoother local variation of hardness with latent displacement, whereas VIB OFF exhibits higher typical slopes and a broader, heavier-tailed distribution, consistent with sharper local property changes and reduced navigability under continuous latent traversal. Bin width is constant across both histograms (as plotted); both panels use identical kNN construction and edge-count normalization.

Comment 3

While the encoder components are described reasonably well, key details of the decoder remain insufficiently specified. To ensure reproducibility, the authors should explicitly provide the decoder architecture and the complete mathematical formulation of the training loss.

Response

We thank the reviewer for pointing this out. In our VIBANN framework, the decoder is a composition-reconstruction head branching directly from the variational information bottleneck (VIB) latent vector $\mathbf{z} \in \mathbb{R}^d$. This head is trained jointly with the hardness predictor and maps the latent representation back to a normalized composition vector on the simplex. Concretely, the decoder takes \mathbf{z} as input and applies two fully connected layers with 128 and 64 hidden units and ReLU activation, followed by batch normalization and Monte Carlo (MC) dropout, and

finally a fully connected output layer with N_{elem} units and a **softmax** activation to produce the reconstructed composition $\hat{\mathbf{x}} \in \Delta^{N_{\text{elem}}-1}$ (non-negative and summing to unity) [Dense(128, ReLU) \rightarrow BN \rightarrow MCdropout \rightarrow Dense(64, ReLU) \rightarrow BN \rightarrow MCdropout \rightarrow Dense(N_{elem} , softmax)]. The reconstruction target \mathbf{x} is the input composition after simplex normalization, ensuring a consistent probabilistic interpretation for the reconstruction loss.

The full training objective optimized in our framework is the sum of three terms: (1) a supervised hardness regression loss, (2) a composition reconstruction loss for the decoder head, and (3) the VIB regularization term (KL divergence between the approximate posterior $q_{\phi}(\mathbf{z}|\mathbf{x})$ and the standard normal prior $p(\mathbf{z}) = \mathcal{N}(\mathbf{0}, \mathbf{I})$). The hardness prediction head outputs a standardized hardness \hat{y}_s and is trained with mean squared error, $\mathcal{L}_{\text{HV}} = \text{MSE}(y_s, \hat{y}_s)$. The composition decoder head is trained using a Kullback-Leibler divergence reconstruction loss, $\mathcal{L}_{\text{recon}} = D_{\text{KL}}(\mathbf{x}||\hat{\mathbf{x}}) = \sum_i x_i \log(x_i/\hat{x}_i)$, multiplied by a fixed weight $\lambda_{\text{recon}} = 0.5$. The VIB regularizer is implemented as a KL penalty added internally by the VIB layer: the encoder parameterizes $q_{\phi}(\mathbf{z}|\mathbf{x}) = \mathcal{N}(\mu(\mathbf{x}), \text{diag}(\sigma^2(\mathbf{x})))$, and the KL divergence $D_{\text{KL}}(q_{\phi}(\mathbf{z}|\mathbf{x})||p(\mathbf{z})) = \frac{1}{2} \sum_{j=1}^d (\exp(\log \sigma_j^2) + \mu_j^2 - 1 - \log \sigma_j^2)$ is computed and normalized by the latent dimensionality d (KL-per-dimension) and averaged over the mini-batch. This KL per dimension is multiplied by a coefficient β and added to the total loss. Importantly, β is adaptively updated during training by a feedback controller to maintain a target KL per dimension (set to 0.15 in our training runs), using a multiplicative factor of 1.05, thereby stabilizing the information budget of the bottleneck across folds and hyperparameter settings.

Accordingly, the total objective minimized during training is

$$\mathcal{L}_{\text{total}} = \text{MSE}(y_s, \hat{y}_s) + \lambda_{\text{recon}} D_{\text{KL}}(\mathbf{x}||\hat{\mathbf{x}}) + \mathbb{E}_{p(\mathbf{x}, \mathbf{y})} \left[\mathbb{E}_{q(\mathbf{z}|\mathbf{x})} [\log p(\mathbf{y}|\mathbf{z})] \right] - \beta \mathbb{E} \left[\frac{1}{d} D_{\text{KL}}(q_{\phi}(\mathbf{z}|\mathbf{x})||p(\mathbf{z})) \right],$$

with $\lambda_{\text{recon}} = 0.5$.

We clarify that the attention tensors are exported as additional outputs solely for interpretability analyses. They are not supervised by labels and therefore carry zero-loss weights. This is a deliberate design choice. The attention weights are not independent free variables; they are deterministic functions of trainable parameters (e.g., projection matrices and downstream layers) and are therefore learned implicitly through backpropagation from the supervised hardness loss (and the auxiliary reconstruction and VIB regularization terms). Introducing a non-zero attention loss would require prescribing a target attention distribution or imposing heuristic priors (e.g., sparsity/entropy penalties) in the absence of ground-truth rationales, which can bias the explanation mechanism and potentially distort predictive fidelity. We therefore avoid direct regularization of attention and instead report the emergent attention maps learned under the predictive objective.

In the revised manuscript (Methods), we (i) explicitly state the decoder topology (latent-to-composition reconstruction head) and (ii) provide the complete mathematical form of the optimized loss, matching our implementation.

Changes in Main Manuscript: In Methods, on pg. 31,

“**Latent to composition decoder head:** A composition reconstruction head branches directly from the latent vector. It maps \mathbf{z} to a simplex-normalized composition $\hat{\mathbf{x}}$ using two fully connected layers with 128 and 64 hidden units, ReLU activations, each followed by batch normalization and Monte Carlo dropout, and a final **softmax** layer with 56 outputs. This head is jointly trained with the hardness predictor and used as the generator in inverse design.

Training objective and adaptive information budget

The total loss minimized during training is the sum of three terms.

1. Supervised hardness regression loss

$$\mathcal{L}_{HV} = \text{MSE}(y_s, \hat{y}_s).$$

2. Composition reconstruction loss for the decoder head

The decoder is trained with a KL divergence reconstruction objective

$$\mathcal{L}_{recon} = D_{KL}(x||\hat{x}) = \sum_i x_i \log\left(\frac{x_i}{\hat{x}_i}\right)$$

weighted by a fixed coefficient $\lambda_{recon} = 0.5$.

3. VIB regularization

$$\mathcal{L}_{VIB} = \mathbb{E}_{p(x,y)} \left[\mathbb{E}_{q(z|x)} [\log p(y|z)] \right] - \beta \mathbb{E} \left[\frac{1}{d} D_{KL} \left(q_\phi(z|x) || p(z) \right) \right]$$

where the KL per dimension is computed in the VIB layer.

Accordingly,

$$\mathcal{L}_{Total} = \mathcal{L}_{HV} + \lambda_{recon} \mathcal{L}_{recon} + \mathcal{L}_{VIB}.$$

Beta controller for KL per dimension: Rather than using a fixed β schedule, β is updated during training by a feedback controller that aims to maintain a target KL per dimension of 0.15. After each epoch, β is multiplied by 1.05 if the observed KL per dimension exceeds the target and divided by 1.05 otherwise, subject to minimum and maximum bounds. This dynamic β -annealing strategy ensures that latent compression is prioritized during the initial training epochs, gradually transitioning to fine-tuning the predictive task. This stabilizes the information budget across folds and hyperparameter settings.”

Comment 4

The framework reports uncertainty estimates (via MC dropout) and calibration, which are valuable for assessing prediction reliability. However, it remains unclear whether and how this uncertainty is incorporated into the inverse-design decision-making.

Response

We thank the reviewer for asking for this clarification. In the inverse design of new, exceptionally hard BMGs, Monte Carlo (MC) dropout predictive uncertainty is used explicitly as a decision

variable to drive risk-aware exploration and candidate selection. Specifically, during latent-space exploration, each proposed latent vector is decoded to a composition and evaluated with MC dropout to obtain predictive mean μ_{HV} and standard deviation σ_{HV} . These uncertainties are incorporated into the inverse-design decision rule at two distinct levels in our implementation:

- (i) **Risk-aware utility used to bias MCMC exploration:** In our framework, the MCMC sampler includes a utility term of the form $U(\mathbf{z}) = \mu_{\text{HV}} - \mathbf{z}_{\mathbf{u}}\sigma_{\text{HV}}$, where $\mathbf{z}_{\mathbf{u}}$ is the uncertainty multiplier used for the utility bias inside the MCMC target. We use $\mathbf{z}_{\mathbf{u}} = 1.02$ (a mild exploration bias inside the sampler), which biases sampling toward regions that are simultaneously high-mean and low-uncertainty under MC dropout, rather than chasing high mean alone.
- (ii) **A conservative bound used as a robustness constraint (penalty) in the sampling objective:** Separately from $\mathbf{z}_{\mathbf{u}}$, the inverse-design pipeline computes a lower confidence bound, $\text{LCB}(\mathbf{z}) = \mu_{\text{HV}} - \mathbf{z}\sigma_{\text{HV}}$, with $\mathbf{z} = 1.645$ (a one-sided $\sim 95\%$ confidence multiplier). It implements a soft robustness constraint by adding a penalty whenever $\text{LCB}(\mathbf{z})$ falls below a data-driven threshold, $\text{LCB}_{\text{TARGET}}$, so that candidates are prioritized only when they are simultaneously high-hardness and high-confidence.

Crucially, this uncertainty-aware LCB is integrated into inverse design at two levels. First, it is built directly into the latent-space sampling objective used by the mixture-aware multi-chain MCMC. During MCMC traversal of the GMM-supported latent manifold, each proposed latent vector is decoded into a composition and evaluated via MC-dropout to estimate μ_{HV} and σ_{HV} . The proposed latent vectors are then scored using a log-target that adds a utility term proportional to $\mu_{\text{HV}} - \mathbf{z}\sigma_{\text{HV}}$ to bias exploration toward robustly high-performing regions, and imposes an explicit penalty when the resulting LCB falls below a data-driven threshold $\text{LCB}_{\text{TARGET}}$. This construction ensures that the sampler concentrates probability mass in regions that are both plausible under the latent prior and robustly high-performing under uncertainty, rather than simply chasing high mean predictions that may be poorly supported.

Second, after pooling MCMC samples and decoding them into candidate compositions, uncertainty is used as a hard gate and as a ranking criterion for down-selection. Each candidate receives MC-dropout estimates $(\mu_{\text{HV}}, \sigma_{\text{HV}})$ and an LCB. The candidates are accepted only if they satisfy (i) a latent-plausibility threshold under the GMM prior, (ii) a novelty threshold relative to the training compositions, (iii) a maximum-uncertainty gate (to avoid highly uncertain predictions), and (iv) the near-peak $\text{LCB}_{\text{TARGET}}$ criterion to ensure high hardness. Concretely, for each candidate decoded from a latent vector \mathbf{z} , we require its latent log-likelihood under the fitted GMM to exceed a threshold set from the training latent distribution, $\log \mathbf{p}_{\text{GMM}}(\mathbf{z}) \geq \log \mathbf{p}_{\text{thr}}$, where $\log \mathbf{p}_{\text{thr}}$ is chosen as the 10th percentile of the training log-likelihoods (weighted consistently with the GMM fitting). Novelty is enforced in composition space by requiring a minimum Euclidean distance from the training set, $\mathbf{d}_{\min}(\mathbf{x}) = \min_{\mathbf{x}_i \in \mathcal{D}_{\text{train}}} \|\mathbf{x} - \mathbf{x}_i\|_2 \geq \mathbf{d}_{\text{thr}}$, where \mathbf{d}_{thr} is set as the 80th percentile of a reference ‘‘training spacing’’ distribution constructed from

training-pair distances and nearest-neighbor distances on a training subset. The threshold LCB_{TARGET} is computed directly from the training set: we first identify a “near-peak” subset as those training compositions with μ_{HV} above the 98th percentile of the training predictive mean distribution, and then set LCB_{TARGET} to the 85th percentile of the training LCB values restricted to this near-peak subset. This yields an internal robustness floor anchored to what the model considers reliably achievable within the learned data manifold, thereby preventing inverse design from selecting candidates whose apparent performance is driven primarily by predictive variance. In addition, to prevent the selection of candidates whose apparent performance is dominated by high epistemic uncertainty, we impose a hard uncertainty cap $\sigma_{HV} \leq \sigma_{max}$, where $\sigma_{max} = 2.5 \times \sigma_{ref,80}$ and $\sigma_{ref,80}$ is the 80th percentile of the training predictive σ_{HV} distribution estimated by MC dropout during the same calibration step. Thus, the final acceptance mask is the conjunction

$$(\log p_{GMM}(z) \geq \log p_{thr}) \wedge (d_{min}(x) \geq d_{thr}) \wedge (\sigma_{HV} \leq 2.5 \sigma_{ref,80}) \wedge (LCB \geq LCB_{TARGET}),$$

ensuring that the reported candidates are simultaneously (a) on-manifold in latent space, (b) compositionally non-trivial relative to training data, and (c) high-hardness under conservative, uncertainty-penalized scoring.

Accordingly, uncertainty is incorporated into inverse design as an explicit risk-control mechanism: (1) it shapes the latent-space exploration policy (via LCB-based utility and penalties during MCMC), and (2) it governs final decision-making (via LCB-based acceptance thresholds, uncertainty gates, and LCB-first ranking). This makes the inverse-design step a deliberately uncertainty-aware selection procedure that preferentially returns alloy candidates with high lower-bound performance rather than high point estimates alone.

Changes in Main Manuscript: In Methods,

“Inverse design of exceptionally hard bulk MMGs

A salient feature of this framework is the encoder-decoder inverse design loop. Inverse design was performed at a fixed design load range of 0.5-10 N. The pipeline consists of four steps: latent density modeling, mixture-aware multi-chain Markov chain Monte Carlo (MCMC) sampling in the latent space, gradient-based refinement of selected seeds, and uncertainty-gated down selection.

Latent density model: A Gaussian mixture model was fit to training latent embeddings evaluated at the design load, with the number of components selected by Bayesian information criterion over a candidate range. The fitted model provides an explicit latent density $p_{GMM}(z)$ and an operational notion of data support through the latent log likelihood.

A feasibility threshold on latent support was defined as

$$\log p_{GMM}(z) \geq \log p_{thr}$$

where $\log p_{thr}$ was set to the 10th percentile of training log likelihoods.

Uncertainty calibration and conservative performance targets: Uncertainty calibration was

performed using MC dropout on a random subset of the training data at the design load. The procedure used 300 stochastic forward passes and up to 5000 samples, then computed μ_{HV} , σ_{HV} , and the one-sided lower confidence bound

$$LCB(z) = \mu_{HV}(z) - z\sigma_{HV}(z)$$

with $z = 1.645$.

A data-anchored robustness floor was defined from training statistics at the design load. First, a near peak subset was identified as samples with μ_{HV} above the 98th percentile. Then the target lower bound was set to the 85th percentile of LCB values in this near-peak subset. A reference uncertainty scale was defined as $\sigma_{ref,q80}$, the 80th percentile of training σ_{HV} values.

Mixture-aware multi-chain MCMC in latent space: Latent exploration was performed using a mixture-aware multi-chain MCMC scheme. Proposals combined global moves based on mixture components with local random-walk perturbations, and feasibility checks were applied using the latent log-likelihood threshold. Each proposed latent vector was decoded to a composition using the trained decoder, projected to the simplex, and evaluated with a small number of MC dropout passes inside the MCMC loop.

Uncertainty enters the MCMC target in a risk-aware utility

$$U(z) = \mu_{HV}(z) - z_u\sigma_{HV}(z)$$

with $z_u = 1.02$ used inside the sampler. The sampler also enforces soft robustness through penalties tied to target LCB (80th percentile of LCB values) and includes additional terms that discourage samples from collapsing into a small region of latent space.

Gradient-based refinement: A subset of high-quality latent seeds was refined by gradient-based optimization of a composite objective that rewards high hardness while penalizing leaving the high support region of the latent GMM and violating robustness. The refinement updates were performed directly in the latent space using backpropagation through the decoder and the hardness head. The *Adam* optimizer is employed with a learning rate of $1e^{-4}$, and early stopping is applied to ensure efficient convergence.

Down selection with feasibility, novelty, and uncertainty gates: Candidates decoded from sampled and refined latent points were subjected to an acceptance mask that required simultaneous satisfaction of four criteria.

1. Latent support

$$\log p_{GMM}(z) \geq \log p_{thr}$$

2. Novelty in composition space

Novelty was quantified by the minimum Euclidean distance to the training set

$$d_{min}(x) = \min_{x_i \in \mathcal{D}_{train}} \|x - x_i\|_2$$

with the novelty threshold set to the 80th percentile of a reference training spacing distribution.

3. Uncertainty cap

$$\sigma_{HV}(z) \leq 2.5\sigma_{ref,q80}$$

4. Conservative hardness requirement

$$LCB(z) \geq LCB_TARGET$$

with LCB_TARGET calibrated from the training distribution at the design load.

Candidates that pass this acceptance mask were ranked and reported for experimental validation.

Comment 5

The code did not provide a README file, so I did not try to install or run the code.

Response 5

We have addressed the code availability concern by adding a comprehensive README.md file to the repository, along with clear installation instructions, a description of the repository structure, the dataset file naming, dependencies (requirements.txt), and the primary entry points for training and inverse design.

In addition, to ensure full methodological reproducibility independent of software installation, we now include in the Supplementary Information three explicit pseudocode listings: Algorithm S1 (prediction model training, including stratified splitting, load scaling and hardness standardization, VIBANN architecture definition, loss terms and hyperparameter selection), Algorithm S2 (BatchNorm-safe MC-dropout uncertainty estimation, including deterministic BatchNorm evaluation with stochasticity confined to custom MCDropout layers, reproducible seed streaming, and computation of μ_{HV}, σ_{HV} , and LCB), and Algorithm S3 (end-to-end inverse design, including latent GMM prior fitting, training-based calibration of LCB_TARGET and uncertainty reference levels, mixture-aware multi-chain MCMC, candidate decoding and simplex projection, novelty/realism/uncertainty/LCB gating, and gradient-based latent refinement with subsequent uncertainty-aware re-screening).

Changes in Supplementary Information:

Algorithm S1. Prediction model training

Inputs:

- Training dataset $\mathcal{D} = \left\{ \left(\mathbf{x}_{\text{comp}}^{(i)}, \mathbf{x}_{\text{load}}^{(i)}, y^{(i)}, \mathbf{w}^{(i)} \right) \right\}_{i=1}^N$, where $\mathbf{x}_{\text{comp}} \in \mathbb{R}^d$ (composition features), $\mathbf{x}_{\text{load}} \in \mathbb{R}$ (load), y (hardness), and \mathbf{w} (sample weights).
- Pre-processing/scalers for \mathbf{x}_{load} and y ; hyperparameter search space.

Outputs:

Trained surrogate final_model (includes attention, VIB latent, Monte-Carlo Dropout), fitted scaler_load_final, output standardization constants (μ_y, σ_y) stored as y_mu_final, y_sd_final and attention scores; Comp_Attention, Load_Attention.

Procedure:

1. Preprocessing

A. Split \mathcal{D} into training/validation/test (Stratified Splitting to preserve distribution among the subsets).

B. Fit the load scaler on training loads; transform all loads:

$$\mathbf{x}_{\text{load,scaled}} \leftarrow \text{scaler_load}(\mathbf{x}_{\text{load}}).$$

C. Standardize targets to scaled space:

$$y_{\text{scaled}} \leftarrow \frac{(y - \mu_y)}{\sigma_y}.$$

2. Define the architecture (Attention \rightarrow trunk \rightarrow VIB \rightarrow prediction head)

A. **Inputs:** (\mathbf{x}_{comp} , $\mathbf{x}_{\text{load,scaled}}$).

B. **Composition attention module:** compute attention-weighted composition representation $\tilde{\mathbf{x}}_{\text{comp}}$ using the attention subnetwork in `final_model`.

C. **Feature trunk:** apply dense transformations with BatchNorm and MCDropout layers (custom dropout that can be toggled during inference).

D. **VIB layer (`vib_layer`):**

- compute latent parameters $(\mu_z, \log\sigma_z^2)$ and sample

$$\mathbf{z} = \mu_z + \sigma_z \odot \epsilon, \quad \epsilon \sim \mathcal{N}(\mathbf{0}, \mathbf{I}),$$

- add the KL term $\text{KL}(q(\mathbf{z}|\cdot)\|p(\mathbf{z}))$ to the total loss via the layer's internal `add_loss`.

E. **Prediction head:** output scaled hardness \hat{y}_{scaled} via the layer named "Output_Layer".

F. **Attention Scores:** exported attention tensors (trained implicitly; zero-loss); `Comp_Attention`, `Load_Attention`

3. Training objective

A. Weighted regression loss in scaled space:

$$\mathcal{L}_{\text{reg}} = \sum_{i=1}^N w^{(i)} (\hat{y}_{\text{scaled}}^{(i)} - y_{\text{scaled}}^{(i)})^2.$$

B. Total loss:

$$\mathcal{L} = \mathcal{L}_{\text{reg}} + \lambda_{\text{recon}} D_{\text{KL}}(\mathbf{x}|\hat{\mathbf{x}}) + \beta(\mathbf{t}) \frac{1}{d} \text{KL}(q(\mathbf{z}|\mathbf{x})\|p(\mathbf{z})),$$

where $\beta(\mathbf{t})$ follows the schedule/controller implemented in the code.

4. Hyperparameter selection (Bayesian Optimization)

A. For each hyperparameter trial: instantiate the model, train with early stopping/checkpointing, evaluate on validation.

B. Select the best trial and retrain/refit the final model accordingly.

5. Batch Normalization-safe Monte-Carlo dropout configuration (for uncertainty quantification)

- Enforce BatchNorm freezing by evaluating with `training=False`.

- Enable epistemic uncertainty only by toggling MCDropout via the `mc_active` flag.

Algorithm S2. MC-dropout epistemic uncertainty for the trained Attention-VIB surrogate

Goal:

Approximate the predictive distribution of hardness at fixed load using Monte Carlo dropout while keeping BatchNorm frozen, and compute μ_{HV} , σ_{HV} and risk-aware LCB.

Inputs:

- Trained model final_model with output Output_Layer (or first element of list/tuple output).
- Candidate compositions $X_{\text{comp}} \in \mathbb{R}^{N \times d}$.
- Design load L (raw, in N) and fitted load scaler scaler_load_final.
- Output scaling constants (μ_y, σ_y) used during training.
- MC sample count S , confidence parameter z , batch size B .
- Base random seed seed_base.

Outputs:

- $\mu_{\text{HV}} \in \mathbb{R}^N$, $\sigma_{\text{HV}} \in \mathbb{R}^N$, $\text{LCB}_{\text{HV}} \in \mathbb{R}^N$.
- Quantiles q_p (e.g., $p = 0.10, 0.80$).
- The full MC sample matrix $\text{HV}^{(s)} \in \mathbb{R}^N$ for $s = 1 \dots S$.

Procedure:

1. Pre-process inputs (conditioning variables)

A. Project compositions onto the simplex (nonnegative, sum-to-one):

$$X_{\text{comp}} \leftarrow \Pi_{\Delta}(X_{\text{comp}}).$$

B. Form a raw load vector $X_{\text{load,raw}} \leftarrow L \cdot \mathbf{1}_{N \times 1}$.

C. Scale load using the training scaler:

$$X_{\text{load,raw}} \leftarrow \text{scaler}_{\text{load}_{\text{final}}} \cdot \text{transform}(X_{\text{load,raw}}).$$

2. Configure BN-safe MC-dropout (epistemic only)

A. Enforce BatchNorm freezing by always evaluating the model with training=False.

B. Enable stochasticity only in custom dropout layers:

- Traverse all sublayers of final_model.
- For each layer m : if m is of class "MCDropout" and has attribute mc_active, set: $m.\text{mc_active} \leftarrow \text{True}$.
- All other layers (including BatchNorm) remain deterministic under training=False.

3. Verify MC-dropout is active under training=False (required diagnostic)

A. For a small test batch $(X_{\text{comp}}^{\text{test}}, X_{\text{load,scaled}}^{\text{test}})$, run S_{check} stochastic forward passes with distinct seeds.

B. Compute the mean per-point standard deviation:

$$\bar{s} \leftarrow \text{mean}_i \left(\text{std}_s(\hat{y}_{\text{scaled},i}^{(s)}) \right).$$

C. If $\bar{s} \leq \epsilon$, abort: MC-dropout is not active (uncertainty estimates would be invalid).

4. Generate an independent seed stream (reproducible, uncorrelated draws)

A. Create a seed sequence with `SeedSequence(seed_base)` and generate S integer seeds:

$$\{\text{seed}_s\}_{s=1}^S.$$

This prevents correlated MC samples while preserving full reproducibility.

5. Monte Carlo dropout sampling (BN frozen)

A. Initialize `samples_scaled` with shape (S, N) .

B. For each draw $s = 1, \dots, S$:

A. Set the global RNG state using `seed_s`.

B. Iterate over mini-batches of $(X_{\text{comp}}, X_{\text{load, scaled}})$ of size B .

C. Evaluate:

$$\hat{y}_{\text{scaled}}^{(s)} \leftarrow \text{final_model}([X_{\text{comp}}, X_{\text{load, scaled}}], \text{training} = \text{False}).$$

D. Extract the scalar prediction vector robustly:

- If output is a dict: use `Output_Layer`.
- If output is a list/tuple: use the first element.
- Else: use the output tensor directly.

E. Store $\hat{y}_{\text{scaled}}^{(s)}$ in `samples_scaled[s, :]`.

C. Disable MC-dropout (set `mc_active=False`) after sampling.

6. Convert to hardness and summarize epistemic uncertainty

A. Convert scaled predictions to hardness: $\text{HV}^{(s)} \leftarrow \hat{y}_{\text{scaled}}^{(s)} \sigma_y + \mu_y$.

B. Compute predictive mean: $\mu_{\text{HV}} \leftarrow \frac{1}{S} \sum_{s=1}^S \text{HV}^{(s)}$.

C. Compute epistemic standard deviation: $\sigma_{\text{HV}} \leftarrow \sqrt{\frac{1}{S} \sum_{s=1}^S (\text{HV}^{(s)} - \mu_{\text{HV}})^2}$.

D. Compute risk-aware lower confidence bound (LCB): $\text{LCB}_{\text{HV}} \leftarrow \mu_{\text{HV}} - z \sigma_{\text{HV}}$.

E. Compute quantiles q_p over $\{\text{HV}^{(s)}\}$ (e.g., $p = 0.10, 0.80$).

Algorithm S3. Inverse design pipeline

Inputs:

- Trained model `final_model` from Algorithm S1
 - Training compositions `X_comp_train_fit`
 - Sample weights `w_train_fit`; scaler `scaler_load_final`
 - Output normalization (μ_y, σ_y)
 - Design load x_{load} (= 0.5-10.0 N)
 - MC-dropout toggling function `set_mc_dropout_inv`;
 - Inverse-design and optimization hyperparameters.
-

Outputs:

- Latent samples and diagnostics (`mcmc_latent_diagnostics_cheap.csv`)
 - Ranked candidate table (`inverse_design_candidates_FINAL.csv`)
 - Optimized candidates (`optimized_candidates_with_uncertainty.csv`).
-

Procedure:

1. Latent prior construction and “near-peak @ design_load” calibration

- A. Fix the design load $x_{\text{load}} = \text{DESIGN_LOAD_VALUE}$ (0.5-10.0 N) and compute $x_{\text{load,scaled}} \leftarrow \text{scaler_load}(x_{\text{load}})$.
- B. Construct an encoder reusing the trained VIB layer
 - A. Define `encoder_model` with
$$z \leftarrow \text{final_model.get_layer}(\text{"vib_layer"}).\text{output}.$$
 - B. Encode all training compositions at the fixed design load:
$$Z_{\text{train}} \leftarrow \text{encoder_model}([X_{\text{comp}}^{\text{train}}, x_{\text{load,scaled}}]).$$
- C. Fit a weighted GMM prior in latent space
 - a. Fit $p_{\text{gmm}}(z)$ with K components to Z_{train} using weights `w_train_fit`.
 - b. Define realism score $\log p_{\text{gmm}}(z)$ using `gmm.score_samples`.
 - c. Set the realism gate `logp_thr` as a weighted quantile of training $\log p$.
- D. Calibrate near-peak behavior (training-based)
 - a. Run BN-safe MC-dropout on a subset of training data at 0.5-10.0 N to compute $\mu_{\text{train}}^{1N}, \sigma_{\text{train}}^{1N}$.
 - b. Define training LCB:
$$\text{LCB}_{\text{train}}^{1N} = \mu_{\text{train}}^{1N} - Z_{\text{conf}} \sigma_{\text{train}}^{1N}.$$
 - c. Define near-peak mean threshold μ_{peak} as a high quantile of μ_{train}^{1N} and define `LCB_TARGET` as a conservative high quantile of $\text{LCB}_{\text{train}}^{1N}$ restricted to the near-peak set.
 - d. Store `SIGMA_REF_Q85` as the 85th percentile of $\sigma_{\text{train}}^{1N}$.

2. Mixture-aware multi-chain MCMC in latent space

- A. Rebuild `decoder_model` by wiring the trained layers from `final_model`:
$$x_{\text{comp}} \leftarrow \text{Comp_Dec_} * \rightarrow \text{Comp_Recon}.$$
 - B. No re-training is performed: weights are shared with `final_model`.
-

-
- C. Define BN-safe MC evaluation for a latent point z (cheap in-MCMC)
 - a. Decode $\hat{\mathbf{x}}_{\text{comp}} = \text{decoder_model}(z)$, project to simplex (nonnegativity + normalization).
 - b. Evaluate $\text{final_model}([\hat{\mathbf{x}}_{\text{comp}}, \mathbf{x}_{\text{load,scaled}}])$ under MC-dropout with `training=False` (BatchNorm frozen) to estimate $\mu(z)$, $\sigma(z)$, and $\text{LCB}(z)$.
 - D. Define the MCMC log-target
 - a. Hard gate: reject if $\log p_{\text{gmm}}(z) < \log p_{\text{thr}}$.
 - b. Soft penalties: LCB shortfall below `LCB_TARGET`, uncertainty above `SIGMA_REF_Q85`, and latent non-centrality terms.
 - E. Run mixture-aware multi-chain MCMC
 - a. Initialize multiple chains from diverse GMM components.
 - b. Propose z' via a mixture of:
 - local Gaussian random walk; and
 - global proposals sampled from the fitted GMM.
 - c. Accept/reject using the Metropolis-Hastings ratio with the implemented proposal correction `log_q`.
 - d. After burn-in and thinning, collect pooled latent samples:

$$\mathbf{Z}_{\text{samp}} = \{z_j\}_{j=1}^M.$$

3. Inverse-design candidate evaluation and selection

- A. Decode latent samples to candidate compositions

$$\mathbf{X}_{\text{cand}} \leftarrow \text{decoder_model}(\mathbf{Z}_{\text{samp}}),$$
 followed by simplex projection.
- B. Define composition novelty threshold from training spacing
 - a. Subsample training compositions $\mathbf{X}_{\text{train,sub}}$.
 - b. Compute a reference distribution of pairwise distances within the subsample.
 - c. Set `novel_thr` to a high quantile of that reference distribution.
- C. Evaluate candidates at 0.5-10 N with high MC-dropout
 - a. For each candidate, compute S MC draws under BN-safe MC-dropout.
 - b. Convert scaled predictions back to hardness: $\text{HV} = y_{\text{scaled}}\sigma_y + \mu_y$.
 - c. Compute μ_{cand} , σ_{cand} , quantiles, and LCB_{cand} .
- D. Acceptance and ranking

Accept if:

 - $\log p_{\text{gmm}}(z) > \log p_{\text{thr}}$,
 - $\sigma_{\text{HV}} \leq 2.5 \times \text{SIGMA_REF_Q85}$,
 - novelty distance $\geq \text{novel_thr}$.

Ranking score (`soft_score`):

 - $\text{LCB} = \mu - z_{\text{conf_seed}} \sigma$
 - $\text{sigma_pen} = \max(0, \sigma - \text{SIGMA_REF_Q85})$
 - $\text{soft_score} = \text{LCB} + 250 * \text{novelty} + 0.03 * \log p - 0.20 * \text{sigma_pen}^2$

4. Gradient-based refinement in latent space

-
- A. Seed selection: Select a small set of high-quality latent seeds from Z_{samp} using feasibility (`logp_thr`) and near-peak constraint (`LCB_TARGET`) plus a ranking score.
 - B. Optimize latent vectors
 - a. Treat latent vectors as continuous variables and decode compositions via the same `decoder_model`.
 - b. Evaluate deterministic mean (dropout OFF) and MC uncertainty (dropout ON) at 0.5-10 N.
 - c. Optimize an LCB-based objective plus extrapolation guards and regularizers (band penalty, prior penalty, seed-pull, uncertainty cap) exactly as in the code.
 - C. Re-evaluate optimized candidates with MC-dropout, apply acceptance rules, and save optimized alloy candidates.
-

-End of the response to Reviewer #2-

References

- 1 Sarker, S. et al. Discovering exceptionally hard and wear-resistant metallic glasses by combining machine-learning with high throughput experimentation. *Applied Physics Reviews* **9**, 011403, (2022).
- 2 Li, H. X., Lu, Z. C., Wang, S. L., Wu, Y. & Lu, Z. P. Fe-based bulk metallic glasses: Glass formation, fabrication, properties and applications. *Progress in Materials Science* **103**, 235-318, (2019).
- 3 Wang, W. H., Dong, C. & Shek, C. H. Bulk metallic glasses. *Materials Science and Engineering: R: Reports* **44**, 45-89, (2004).
- 4 Kong, F., Inoue, A., Wang, F. & Chang, C. The Influence of Boron and Carbon Addition on the Glass Formation and Mechanical Properties of High Entropy (Fe, Co, Ni, Cr, Mo)-(B, C) Glassy Alloys. *Coatings* **14**, 118, (2024).
- 5 Zhai, F., Pineda, E., Duarte, M. & Crespo, D. Role of Nb in glass formation of Fe–Cr–Mo–C–B–Nb BMGs. *Journal of Alloys and Compounds* **604**, 157–163, (2014).
- 6 Wang, Z., Hutter, F., Zoghi, M., Matheson, D. & De Feitas, N. Bayesian optimization in a billion dimensions via random embeddings. *Journal of Artificial Intelligence Research* **55**, 361-387, (2016).
- 7 Eriksson, D., Pearce, M., Gardner, J., Turner, R. D. & Poloczek, M. Scalable global optimization via local Bayesian optimization. *Advances in neural information processing systems* **32**, (2019).
- 8 Liu, H., Cai, J., Ong, Y.-S. & Wang, Y. Understanding and comparing scalable Gaussian process regression for big data. *Knowledge-Based Systems* **164**, 324-335, (2019).
- 9 Tighineanu, P. et al. in *International conference on artificial intelligence and statistics*. 6152-6181 (PMLR).

Detailed response to reviewers' comments

We would like to express our sincere gratitude to all the reviewers for their kind and repeated efforts in reviewing our manuscript in its revised form, and for their constructive suggestions and great support.

In the following, we provide a detailed point-by-point response report that explains all revision items. Relevant modifications in the manuscript and supplementary information have been highlighted in **YELLOW**.

Reviewer #1

Overall comment

The authors have made substantial revisions that address all of my core concerns. I particularly appreciate their constructive response to my critique, which included not only additional computational analysis but also new experimental work—specifically, the synthesis and characterization of the nominal $B_{72}Nb_{23}Fe_3$ ternary baseline alloy to directly address the relationship between their designed compositions and prior literature. The clarifications regarding terminology (replacing "ultrahard" with "exceptionally hard"), the explicit domain-bounded framing of interpretability claims, and the transparent quantification of model performance across hardness regimes all strengthen the manuscript considerably. I recommend publication.

Overall response

We sincerely thank the reviewer for the careful re-evaluation of our manuscript and for this very positive assessment. We are grateful that the reviewer found the revised manuscript to have addressed the core concerns raised in the previous round. We particularly appreciate the recognition of the additional computational analyses and the new experimental synthesis and characterization results added to the manuscript. We sincerely thank the reviewer for the constructive and rigorous comments, which have significantly improved the scientific precision and clarity of the study.

Comment 1

I offer one minor suggestion that could enhance accessibility for a broader readership. The authors provide an excellent and thorough analysis of the model's progressive underprediction at high hardness in the Supplementary Information. This analysis offers important mechanistic insights into both the model's behavior and the potentially distinct physical mechanisms governing ultra-high hardness in metallic glasses. However, since many readers do not carefully examine supplementary materials, a brief, explicit statement in the main text would make this

important insight more visible. Perhaps a short addition in the section on VIBANN performance or in the Discussion section, similar to something below?

"The observed systematic underprediction in the high-hardness regime (detailed in Supplementary Fig. S7) suggests that achieving hardness substantially above 2500 HV may involve mechanisms not fully captured by the current training distribution, where such extreme values are sparsely represented. This may reflect either qualitatively distinct deformation physics at the highest hardness levels or insufficient sampling of the relevant compositional space."

Response

We thank the reviewer for this helpful suggestion. We agree that the upper-tail underprediction is an important point and that its broader significance should be stated explicitly in the main text rather than remaining only in the Supplementary Information. Accordingly, we have now added a brief statement in the main-text discussion of model performance and calibration to clarify that the systematic underprediction observed in the highest-hardness regime likely reflects the sparse representation of such extreme values in the present training distribution and may also indicate distinct deformation behavior or insufficient sampling of the relevant compositional space at the upper tail.

Changes in the main text: In Results, on pgs. 7-8;

"Together, these diagnostics indicate that the VIBANN model's uncertainty is overall sufficiently calibrated to support conservative scoring and inverse design, while residual analysis and signed-error statistics show mildly progressive, weaker coverage in the sparsest, high-hardness tail. This systematic underprediction in the highest-hardness regime suggests that hardness substantially above 2500 HV may involve features that are only incompletely captured by the present training distribution, where such extreme values remain sparsely represented (Supplementary Fig. S7). This may reflect either distinct deformation behavior at the extreme upper tail or limited sampling of the relevant compositional space."

-End of the response to Reviewer #1-

Reviewer #2

Overall comment

I have examined the authors' detailed responses to the reviewers' comments as well as the corresponding modifications to the manuscript. In my assessment, the authors properly addressed each concern, and revised the manuscript accordingly. The manuscript has been significantly improved. Therefore, I recommend acceptance of this manuscript for publication in Nature Communications.

Overall response

We sincerely thank the reviewer for the careful re-evaluation of our revised manuscript and for this positive assessment. We are grateful that the reviewer found that the concerns raised in the previous round were properly addressed and that the corresponding revisions have substantially improved the manuscript. We thank the reviewer for the constructive comments throughout the review process, which have helped strengthen the scientific rigor, clarity, and overall presentation of the work.

-End of the response to Reviewer #2-